# Efficient Distributed Optimization under Heavy-Tailed Noise

**Su Hyeong Lee** [1]  **Manzil Zaheer** [2]  **Tian Li** [3]

## Abstract

Distributed optimization has become the default training paradigm in modern machine learning due to the growing scale of models and datasets. To mitigate communication overhead, local updates are often applied before global aggregation, resulting in a nested optimization approach with inner and outer steps. However, heavy-tailed stochastic gradient noise remains a significant challenge, particularly in attention-based models, hindering effective training. In this work, we propose TailOPT, an efficient framework designed to address heavy-tailed noise by leveraging adaptive optimization and novel clipping techniques. We establish convergence guarantees for the TailOPT framework under heavy-tailed noise with local updates and potentially unbounded gradient variance. Among its variants, we propose a memory- and communication-efficient instantiation (named $Bi^2Clip$) that performs coordinate-wise clipping from both above and below at both the inner and outer optimizers. $Bi^2Clip$ brings about benefits of adaptive optimization (e.g., Adam) without the cost of maintaining or transmitting additional gradient statistics. Empirically, TailOPT, including $Bi^2Clip$, demonstrates superior performance on various tasks and models compared with state-of-the-art methods, while being more efficient.

## 1. Introduction

The training of deep learning models including large language models (LLMs) has become increasingly resource-intensive, driven by expansive datasets and models with billions of parameters (Rosa et al., 2022; Liu et al., 2024b; Sriram et al., 2022; Dehghani et al., 2023). As the computational demands escalate, distributed learning has emerged as the default approach, enabling the parallel activation of training processes across multiple compute nodes such as GPUs or datacenters. However, this paradigm introduces a new bottleneck of communication overhead, especially as the progress in compute power has outpaced that of network infrastructure (Wu et al., 2023; DeepSeek-AI, 2024).

To mitigate these communication challenges, one promising strategy is the utilization of local updates. By allowing each compute node to perform multiple gradient updates locally before aggregation, the frequency and volume of inter-node communication can be significantly reduced (Smith et al., 2018; Stich, 2018; McMahan et al., 2017; Lee et al., 2024; Liu et al., 2024a; Jaghouar et al., 2024). For instance, the recent DiLoCo algorithm for training LLMs in datacenter environments can apply several hundred local gradient updates prior to aggregation to reduce communication costs (Douillard et al., 2024). This approach naturally formulates a **nested** optimization problem, where *inner* optimization occurs within each compute node, and *outer* optimization is orchestrated by the coordinating node(s).

However, training attention-based models like LLMs introduce an additional challenge due to the properties of their stochastic gradient distributions. Empirical and theoretical investigations have consistently demonstrated that the gradient noise in these models follows a heavy-tailed distribution (Ahn et al., 2024; Nguyen et al., 2019; Simsekli et al., 2019; 2020; Kunstner et al., 2024; Gorbunov et al., 2020). This heavy-tailed behavior, characterized by high or infinite variance and potentially very large deviations, poses significant challenges to the stability and convergence of existing optimization algorithms (Zhang et al., 2020b; Lee et al., 2024). Addressing these challenges necessitates the development of novel optimization strategies and a more principled understanding of their theoretical underpinnings.

In this work, we propose TailOPT, an efficient and theoretically principled nested training framework, designed to address the challenges posed by heavy-tailed gradient noise in distributed training with local updates. TailOPT introduces several key strategies, including clipping mechanisms (such as coordinate-wise or $L_2$-clipping) and adaptivity, applied at both inner and outer optimizers, to mitigate the adverse effects of heavy-tailed noise. We note that the preconditioning step in adaptive optimizers (e.g., Streeter & McMahan, 2010) may be viewed as a form of soft clipping.

---

[1]Department of Statistics, University of Chicago [2]Meta [3]Department of Computer Science, University of Chicago. Correspondence to: Su Hyeong Lee <sulee@uchicago.edu>.

*Proceedings of the 42$^{nd}$ International Conference on Machine Learning*, Vancouver, Canada. PMLR 267, 2025. Copyright 2025 by the author(s).

We analyze the convergence of TailOPT while incorporating such adaptive methods, while allowing for heavy-tailed noise with unbounded variance. Among the various instantiations of the TailOPT framework, we highlight $Bi^2Clip$, a particularly scalable method that applies coordinate-wise clipping to gradients during inner iterations, and to model parameter updates at outer communication rounds, enforcing thresholding from both above and below on a per-coordinate basis. Our empirical and theoretical results demonstrate that TailOPT is strongly effective in mollifying heavy-tailed noise, enhancing the stability and convergence of the training dynamics across several language benchmarks as well as synthetic data.

Our contributions may be summarized as follows.

- We introduce TailOPT, a general distributed training framework for large-scale models under communication-efficient local updates and heavy-tailed gradient distributions. Among its instantiations, we highlight $Bi^2Clip$, which adjusts to gradient geometry similar to adaptive optimizers (e.g., Adam (Kingma & Ba, 2015)) while avoiding additional memory and communication overhead for maintaining or transmitting preconditioners.

- We provide convergence guarantees for a class of TailOPT algorithms that leverage adaptive optimizers and various clipping strategies, effectively addressing heavy-tailed noise with potentially infinite gradient variance. This is achieved using a nested optimization framework, where the inner optimizer employs clipping operations to mitigate heavy-tailed gradient noise, while the outer optimizer utilizes either fully adaptive or efficient approximations of adaptive updates to guide the optimization process.

- We validate the practicality and effectiveness of TailOPT through extensive experiments on synthetic and real-world datasets in large-scale settings. Our experiments demonstrate that TailOPT produces several algorithmic instantiations that consistently outperform state-of-the-art baselines while being more efficient.

## 2. Related Works

We cite the most related work in this section, and provide an extended literature review in Appendix A.

**Heavy-Tailed Gradient Noise.** Training transformers and LLMs is complicated by heavy-tailed stochastic gradient distributions with very large variance, often theoretically and empirically modeled as Lévy $\alpha$-stable processes (Ahn et al., 2024; Nguyen et al., 2019; Simsekli et al., 2019; 2020; Gorbunov et al., 2020; Kunstner et al., 2024; Chezhegov et al., 2024a). In such scenarios, vanilla SGD-based optimization methods have been shown to destabilize during training in both centralized as well as distributed settings (Gorbunov et al., 2020; Zhang et al., 2020b; Lee et al., 2024).

Recent advancements have explored centralized adaptive optimization techniques and robust gradient aggregation methods to mitigate the adverse effects of heavy-tailed noise, including gradient clipping (Simsekli et al., 2019; Juditsky et al., 2019a; Gorbunov et al., 2024a; Sadiev et al., 2023; Cutkosky & Mehta, 2021; Nguyen et al., 2023a) or adaptive clipping strategies (Chezhegov et al., 2024a). However, the complexities of handling heavy-tailed noise in nested distributed optimization environments often prevent these algorithms and their convergence bounds from extending to scenarios with multiple nodes training in parallel. Additionally, algorithms utilizing adaptive updates generally require preconditioner maintenance that incurs substantial memory costs. To our knowledge, developing an efficient distributed algorithm that provably converges under heavy-tailed stochastic gradient noise has remained an open challenge. For example, although DiLoCo (Douillard et al., 2024; Jaghouar et al., 2024; Liu et al., 2024a) is a recent algorithmic development with local updates for communication efficiency that demonstrates competitive empirical performance, it noticeably lacks theoretical guarantees, and incurs non-trivial overheads when using adaptive optimizers. Our method addresses these gaps by introducing a nested and principled optimization framework, where a particular instantiation ($Bi^2Clip$) brings about benefits of adaptivity without the added overhead of maintaining preconditioners, which also outperforms DiLoCo empirically (Section 6).

**Clipping for Stabilizing Training Dynamics.** Due to its success in stabilizing model updates, gradient clipping is a common technique that has been extensively studied empirically (Gehring et al., 2017; Merity et al., 2018; Peters et al., 2018; Mikolov, 2012) and theoretically (Gorbunov et al., 2020; Zhang et al., 2020b; Chezhegov et al., 2024a; Cutkosky & Mehta, 2021; Menon et al., 2020; Zhang et al., 2020a; Chen et al., 2020; Koloskova et al., 2023; Das et al., 2024). The majority of results study the centralized setting (e.g., Gorbunov et al., 2024a; Puchkin et al., 2024; Zhang & Cutkosky, 2022; Liu et al., 2023; Nguyen et al., 2023b; Parletta et al., 2024; Li & Liu, 2022), as moving to the distributed setting with local updates for communication efficiency provides significant added analytical challenges such as multiple inner optimizer updates prior to outer optimizer synchronization. Additionally, it was shown that using a constant clipping threshold can induce gradient bias, preventing the algorithm from ever converging (Chen et al., 2020; Koloskova et al., 2023). Therefore, some works have attempted to circumvent this issue by debiasing via error feedback (Khirirat et al., 2023; Zhang et al., 2024). Other works in distributed optimization have imposed strong distributional stochastic gradient structures in the analysis. For instance, Qian et al. (2021) assume a well-behaved angular dependence between the stochastic and deterministic gradients throughout training, and Liu et al. (2022) assume

symmetric gradient noise, almost surely bounded stochastic gradients, as well as homogeneous data.

Our proposed clipping mechanism, realized as an instantiation of TailOPT (i.e., $BiClip$), fundamentally differs from prior approaches by integrating per-coordinate clipping from both above and below. The inner optimization steps employ clipping operations to adapt to the gradient geometry, complemented by the outer optimizers which enhance rare signals through adaptivity or adaptive approximations. In addition, in the analysis of TailOPT (Section 5), we do not impose any conditions on the noise nor data distributions except for finite noise $\alpha$-moment for some $\alpha \in (1, 2)$. Our algorithm and analysis also accommodate local updates and allow for potentially unbounded stochastic gradient variance. We provide an extended review of distributed algorithms under heavy-tailed noise in Appendix A.

## 3. Problem Formulation

In distributed optimization, the global objective is constructed by taking a weighted average over the local node objectives $F_i(x)$ for model parameters $x \in \mathbb{R}^d$ and node $i$. In scenarios where data sizes at each node are unbalanced or sampling probabilities vary, the objective becomes:

$$F(x) = \sum_{i=0}^{N-1} p_i F_i(x), \qquad (1)$$

where $p_i$ is proportional to the local data size of node $i$. Here, $F_i(x)$ is defined as $\mathbb{E}_{\xi \sim \mathcal{D}_i} [F_i(x, \xi)]$, where $F_i(x, \xi) = F_i(x) + \langle \xi, x \rangle$ represents the stochastic local objective, and $\mathcal{D}_i$ is the noise distribution of node $i$. This term comes from integrating the gradient noise model $\nabla F_i(x_i^t, \xi_i^t) = \nabla F(x_i^t) + \xi_i^t$, where $x_i^t, \xi_i^t$ are the parameter weights and stochastic gradient noise of node $i$ at timestep $t$. In our formulation and theoretical analysis (Section 5), we allow for both independent and identically distributed (IID) data across $N$ nodes, as commonly observed in datacenter environments, and more challenging *non-IID* data distributions. We now present the assumptions used in the convergence analysis.

**Assumption 1** ($L$-smoothness). *For all $x, y \in \mathcal{X}$ and $i \in [N]$, the local objectives $F_i(x)$ satisfy $F_i(x) \leq F_i(y) + \langle x - y, \nabla F_i(y) \rangle + L_i \|x - y\|^2 / 2$.*

**Assumption 2** (Bounded $\alpha$-moment). *For all nodes $i \in [N]$ with noise distribution $\mathcal{D}_i$, there exists $\alpha_i \in (1, 2)$, $B_i > 0$ such that $\mathbb{E}[\|\xi_i\|^{\alpha_i}] < B_i^{\alpha_i}$.*

Assumption 2 expresses that the noise distribution can be heavy-tailed. In particular, we note that the variance of the noise can be infinite ($\alpha_i = 2$), a setting in which distributed SGD was shown to fail to converge, both empirically and theoretically (Yang et al., 2022; Lee et al., 2024) This condition on the $\alpha_i$ is 'optimally weakest', in that sending $\alpha_i \to 1^+$

recovers the integrability condition of the noise, the minimal assumption necessary to form expectations. Furthermore, we note that $\mathbb{E}\|\xi\|^\alpha < \infty \implies \mathbb{E}\|\xi\|^\beta < \infty$ for $\forall \beta < \alpha$, $\alpha \in \mathbb{R}$. Therefore, we let $\alpha := \min_{i \in [N]} \alpha_i \in (1, 2)$ in the proceeding analysis for notational convenience.

We note that this is strictly weaker than a conventional heavy-tailed assumption on the stochastic gradients, which is commonly given (e.g., Zhang et al., 2020b)) as

$$\mathbb{E}[\|\nabla F_i(x_i^t, \xi_i^t)\|^{\alpha_i}] < B_i^{\alpha_i},$$

which implies that $\nabla F_i(x_i^t)$ is bounded. By contrast, this cannot be implied by Assumption 2. We also note that some works in the literature also define heavy-tailed distributions with *bounded* variance when establishing algorithm convergence bounds (e.g., Gorbunov et al., 2020; Parletta et al., 2024; Li & Liu, 2022; Das et al., 2024), which differs from our definition. We carry out our convergence proofs which subsumes the more general infinite variance setting, which naturally implies convergence under bounded stochastic gradients or variance.

## 4. TailOPT: An Efficient Heavy-Tailed Optimization Framework

In this section, we begin by motivating the heavy-tailed optimization framework (TailOPT), a scalable training framework for heavy-tailed noise. SGD is a strong candidate given its simplicity and efficiency, but it has been shown to diverge under heavy-tailed noise in both centralized (Zhang et al., 2020b) and distributed settings (Lee et al., 2024). Therefore, modifications are necessary to stabilize the noisy updates.

Gradient clipping is a widely adopted technique to mitigate the impact of large gradients (Menon et al., 2020; Zhang et al., 2020a; Chen et al., 2020; Koloskova et al., 2023; Yang et al., 2022). Typically, the clipping operator rescales the gradient uniformly to ensure its $L_2$ norm remains below a predefined threshold. This procedure is mathematically equivalent to applying a dynamically adjusted, lower learning rate when large stochastic gradients are encountered. Therefore, we first include and analyze the usage of $L_2$ clipping ($L_2Clip$) in TailOPT to stabilize heavy-tailed noisy updates. More specifically, we use $L_2$ clipping on the gradients prior to standard gradient descent updates on each node, while a global model weight projection strategy is utilized on the outer optimizer after synchronizing all the collected updates. For additional clarity, the precise pseudocode (Algorithm 2) and analysis are given in Appendix C.1.

**Interpolating Adaptivity: $BiClip$.** However, previous works on $L_2$ clipping of gradients or model updates (e.g., Yang et al., 2022) do not adapt to gradient geometry, as

they proportionally and uniformly downscale each gradient coordinate. Therefore, smaller signals become even more difficult to detect and propagate. Adaptive optimizers have consistently demonstrated superior performance for training modern architectures (Zhang et al., 2020b; Reddi et al., 2021; Lee et al., 2024). Key among adaptive methods such as Adam (Kingma & Ba, 2015) and Adagrad (Duchi et al., 2011; Streeter & McMahan, 2010) is the use of pre-conditioning, where preconditioners that are estimated from historical gradients effectively result in per-coordinate learning rates. This process amplifies rare yet important gradient coordinates, while scales down uninformative gradients, speeding up the convergence. The trade-off, however, lies in the increased systems requirements to compute and maintain preconditioners. For instance, deploying Adam can triple the memory demand to host model parameters during minibatch backpropagation, due to the maintenance of first and second moment exponentially decaying moving average statistics compared to vanilla SGD.

To take advantage of adaptivity without incurring additional memory or communication overhead, we propose a new clipping mechanism, $BiClip$, that performs coordinate-wise clipping from both above and below. $BiClip$ is motivated by an interpolation between clipped-SGD and adaptive methods. It relies on two clipping thresholds (scalars) to dynamically rescale gradients in a per-coordinate fashion, while eliminating the overhead of preconditioner maintenance. For a model parameter $x \in \mathbb{R}^m$, parameter coordinate $j \in [m]$, lower clipping threshold $d$, and upper clipping threshold $u$ ($0 \leq d \leq u$), $BiClip$ is defined as[1]

$$
\begin{aligned}
BiClip(u, d, x)_j &:= \text{sign}(x_j) \left[ d \, \chi \left( |x_j| \leq d \right) \right] \\
&+ \text{sign}(x_j) \left[ u \, \chi \left( |x_j| \geq u \right) + |x_j| \chi \left( d < |x_j| < u \right) \right],
\end{aligned} \quad (2)
$$

where $\chi$ is the indicator function.

$BiClip$ draws on the intuition of adaptive methods by selectively amplifying small gradient coordinates ($|x_j| \leq d$) while clipping down large ones ($|x_j| \geq u$). In contrast to typical adaptive optimizers, $BiClip$ does not require preconditioner maintenance, with significantly reduced optimizer requirements identical to SGD. While our focus is on the distributed setting which aligns with practical applications, we note that $BiClip$ itself can also be beneficial for centralized training (empirically validated in Section 6.4). In the next paragraph, we discuss our general TailOPT framework (Algorithm 1), where one instance of TailOPT applies $BiClip$ as both the inner and outer optimizers.

**TailOPT.** In the TailOPT framework (Algorithm 1), the inner optimization strategy, denoted as $TailClip$, refers to either $BiClip$ or $L_2Clip$. In Line 10, the outer optimization strategy can be either adaptive or non-adaptive methods that incorporate clipping, adaptivity, or momentum on top of

---

[1]For clarity in notation, we define $0/0 := 0$.

$\Delta_t$ by treating them as *pseudogradients*. We present multiple instantiations of TailOPT along with their convergence bounds under heavy-tailed noise in Section 5, as well as in Appendix C.

---

**Algorithm 1** Heavy-Tailed Optimization (TailOPT)

**Require:** Initial model $x_1$, learning rate schedule $\eta_t$
    Clipping schedules $u_t \geq d_t \geq 0$,
    Synchronization timestep $z \in \mathbb{Z}_{>0}$
1: **for** $t = 1, \ldots, T$ **do**
2:     **for** each node $i \in [N]$ in parallel **do**
3:         $x_{i,0}^t \leftarrow x_t$
4:         **for** each local step $k \in [z]$ **do**
5:             Draw gradient $g_{i,k}^t = \nabla F_k(x_{i,k}^t, \xi_{i,k}^t)$
6:             $x_{i,k+1}^t \leftarrow x_{i,k}^t - \eta_t \cdot TailClip(u_t, d_t, g_{i,k}^t)$
7:         **end for**
8:     **end for**
9:     $\Delta_t = \frac{1}{N} \sum_{i \in [N]} \left( x_{i,z}^t - x_{t-1} \right)$
10:     $x_t = Outer\_Optimizer\left( x_{t-1}, \Delta_t \right)$
11: **end for**

---

Among those, we propose and highlight one efficient method that achieves superior empirical performance which utilizes the $BiClip(\cdot)$ operator (Eq. (2)) in both the inner and outer optimizers, called $Bi^2Clip$. The exact pseudocode is presented in Algorithm 4 (Appendix C.3). Intuitively, $Bi^2Clip$ mitigates the effects of heavy-tailed noise across all inner as well as outer optimizers, while mimicking adaptive updates to amplify rare gradient signals. In Section 6, we empirically demonstrate that $Bi^2Clip$ outperforms state-of-the-art baselines without transferring or maintaining preconditioners in the distributed setting.

For clarity, throughout the paper, we list the outer optimizer followed by the inner optimizer when referencing algorithms. For example, 'Adam-$BiClip$' instantiates Adam as the outer optimizer and $BiClip$ as the inner optimizer. Similarly, 'RMSProp-$TailClip$' refers to RMSProp as outer optimizer, and $TailClip$ (either $L_2Clip$ or $BiClip$) as the inner optimizer. Finally, '$Bi^2Clip$' refers to the algorithm with $BiClip$ as both inner and outer optimizers.

## 5. Convergence of the TailOPT Framework

Due to space constraints, we present convergence results for only a subset of TailOPT instantiations in the main text. For a comprehensive analysis, Appendices C.1 and C.2 provide detailed convergence bounds for Avg-$L_2Clip$, and Appendices C.3-C.6 include additional convergence analyses and precise pseudocodes for various (adaptive) algorithms of the TailOPT framework incorporating Adagrad, RMSProp, or Adam. Additionally, we note that the convergence of $Bi^2Clip$ subsumes that of Avg-$BiClip$.

While clipping offers the benefit of stabilization, it introduces a non-zero bias on the stochastic gradients, rendering them to be no longer unbiased estimators of the true gradient. Theorems 1 and 2 demonstrate that with appropriately chosen (increasing) upper clipping $u_t$ and (decreasing) learning rate $\eta_t$ and lower clipping $d_t$ schedules, convergence of Algorithm 1 is nevertheless attainable. Up to $\mathcal{O}(d)$, the presented convergence bounds hold for both gradient-wise clipping as well as coordinate-wise clipping. Generalization to layer-wise clipping with varying thresholds specific to each layer or model weight tensor slice is straightforward.

We carry out our analysis for possibly non-convex problems where the model weights $x_t \in \mathcal{X}$ are contained within a sufficiently large, compact set $\mathcal{X} \subset \mathbb{R}^d$. In such settings, finding the global minimum is known to be NP-Hard, and the standard convergence metric is the stabilization of the minimum gradient. We then obtain the following theorems, where the pseudocode for each algorithm instantiation is detailed in Appendix C.

**Theorem 1.** *Let Assumptions 1-2 hold. Instantiating the outer optimizer in Algorithm 1 with RMSProp gives Algorithm 6 (RMSProp-$TailClip$). Let the clipping and learning rate thresholds satisfy $\eta_t = \Theta(t^\omega)$, $\eta_\ell^t = \Theta(t^\nu)$, $d_t = \Theta(t^\gamma)$, and $u_t = \Theta(t^\zeta)$. Impose the rate schedule conditions $\zeta > 0$, $\gamma < -1/2$, and $\omega - \zeta - 1/2 > 0$. Then, we have that*

$$\min_{t \in [T]} \mathbb{E} \left\| \nabla F(x_t) \right\|^2 \leq \sum_{i=1}^{6} \Psi_i,$$

*where the $\Psi_i$ are upper bounded by*

$$\Psi_1 \leq \mathcal{O}(T^{-\omega+\zeta-\frac{1}{2}}), \quad \Psi_2 \leq \mathcal{O}(T^{\omega+2\nu+3\zeta+\frac{1}{2}}),$$

$$\Psi_3 \leq \mathcal{O}(T^{4\zeta+3\nu+\frac{1}{2}}), \Psi_4 \leq \mathcal{O}(T^{2\nu+2\zeta+\frac{1}{2}}),$$

$$\Psi_5 \leq \mathcal{O}(T^{\nu+\gamma+\zeta+\frac{1}{2}}), \quad \Psi_6 \leq \mathcal{O}(T^{\nu+(2-\alpha)\zeta+\frac{1}{2}}),$$

*which guarantees convergence via an inversely proportional power law decay with respect to $T$. Here, the exponential moving average parameter of the second pseudogradient moment is fixed within the range $\widetilde{\beta}_2 \in [0, 1)$.*

In particular, the proof of this result immediately implies the following summarizing corollary.

**Corollary 1.** *Algorithm 6 (RMSProp-$TailClip$) convergences under heavy-tailed stochastic gradient noise. The convergence rate can be attained for*

$$\zeta = \frac{1}{2\alpha}, \quad \nu = -\frac{\alpha+1}{2\alpha}, \quad \omega = 0$$

*as stabilizing the minimum expected gradient norm squared at rate $\mathcal{O}(T^{(1-\alpha)/2\alpha})$.*

The full proofs of all results in this section are given in Appendix C, which holds for both convex and non-convex functions. This achieves convergence in the presence of

heavy-tailed noise with local updates where the rate is dependent on the $\alpha$-moment condition, which is consistent with prior convergence results in the heavy-tailed setting (e.g., Zhang et al. (2020b)). As $\alpha \to 1^+$, we see that convergence is nullified. We also attain the identical convergence rate for an alternate instantiation (Adagrad-$TailClip$) and provide the exact algorithm in Algorithm 5 and convergence result in Theorem 6 of the appendix.

When deploying distributed optimization, adaptive optimizers such as Adam can considerably increase the memory requirements on each compute node due to preconditioner storage, which matches the model parameter tensor size. This issue gets magnified when one deploys adaptive optimizers at both local compute notes and outer coordinating nodes, as preconditioners may be transmitted from outer to inner optimizers to maximize performance, incurring additional communication overhead (Wang et al., 2021b; Sun et al., 2023). This naturally motivates us to apply $BiClip$ (Eq. (2)) at both inner and outer optimizers, resulting in $Bi^2Clip$, retaining the benefits of adaptivity with minimal overhead. In general, all instantiations of our framework do not require to transmit preconditioners or extra gradient statistics. Convergence results of $Bi^2Clip$ are given below.

**Theorem 2.** *Let the learning rate and clipping schedules satisfy $\eta_t = \Theta(t^\omega)$, $\eta_\ell^t = \Theta(t^\nu)$, $d_t = \Theta(t^\gamma)$, $u_t = \Theta(t^\zeta)$, $\widetilde{d}_t = \Theta(t^{\widetilde{\gamma}})$, and $u_t = \Theta(t^{\widetilde{\zeta}})$. For $Bi^2Clip$ (Algorithm 4), we have that the minimum gradient satisfies*

$$\min_{t \in [T]} \mathbb{E}[\|\nabla F(x_{t-1})\|^2] \lesssim \sum_{i=1}^{7} \Psi_i,$$

*where the $\Psi_i$ are given*

$$\Psi_1 = \mathcal{O}\left(T^{-\omega-\nu-1}\right), \quad \Psi_2 = \mathcal{O}\left(T^{\omega+2\widetilde{\zeta}-\nu}\right), \quad \Psi_3 = \mathcal{O}\left(T^\gamma\right),$$

$$\Psi_4 = \mathcal{O}\left(T^{\widetilde{\gamma}-\nu}\right), \quad \Psi_5 = \mathcal{O}\left(T^{(\alpha-1)\nu+(1-\alpha)\widetilde{\zeta}}\right),$$

$$\Psi_6 = \mathcal{O}\left(T^{(1-\alpha)\zeta}\right), \quad \Psi_7 = \mathcal{O}\left(T^{\nu+\zeta}\right).$$

*To attain convergence, we impose $\zeta, \widetilde{\zeta} > 0 > \gamma, \widetilde{\gamma}$, for $\omega, \nu \leq 0$, as well as $-1 < \omega + \nu$. Then, for $\gamma, \widetilde{\gamma}$ small enough and*

$$\nu = -\frac{\alpha}{4\alpha-2}, \quad \zeta = \frac{1}{4\alpha-2}, \quad \omega = -\frac{1}{2},$$

*$Bi^2Clip$ converges with rate $\mathcal{O}(T^{-\sigma})$, where $\sigma = (\alpha-1)/(4\alpha-2)$ in the limit $\widetilde{\zeta} \to 0^+$.*

This gives the following corollary.

**Corollary 2.** *Algorithm 4 ($Bi^2Clip$) converges with respect to heavy-tailed stochastic gradient noise ($\alpha > 1$). For instance, if the moment is further constrained by $\alpha \geq 1.5$, the algorithm converges with a rate of at least $\mathcal{O}(T^{-\sigma})$ for $\sigma = 1/8$.*

Similar as RMSProp-$TailClip$, the results here hold for

both convex and non-convex functions as long as the assumptions are satisfied. The convergence rate given in Corollary 2 represents a lower bound on the maximal achievable rate, obtained by a selection of hyperparameters. Interestingly, our empirical results demonstrate that $Bi^2Clip$ outperforms other methods, suggesting that the current convergence bounds could be further refined.

**Discussions.** To ensure convergence and mitigate bias in the derived bound, it is necessary for the upper clipping threshold $u_t \to \infty$ and the lower clipping threshold $d_t \to 0$ as $t \to \infty$, consistent with established counterexamples that occur due to unmitigated clipping bias (Koloskova et al., 2023; Chen et al., 2020). In cases where stochastic gradients are sampled from large-variance distributions, this necessitates a continual warm-up phase that is continuously relaxed, akin to learning rate *warm-up* schemes that conclude after a finite period (Kosson et al., 2024).

The clipping schedules prescribed by Theorems 1, 2 grow polynomially with respect to $t$, which depict the realization of model weights throughout training. This effectively deactivates gradient clipping after an initial warm-up phase that is shaped by the noise distribution's tail behavior and the clipping thresholds. This may help to explain why learning rate warm-ups are observed to significantly improve training (Kalra & Barkeshli, 2024; Ma & Yarats, 2021) in the presence of heavy-tailed stochastic gradients. Finally, as the maximal bounded moment condition $\alpha$ approaches the integrability threshold ($\alpha = 1$), or as $\gamma$ nears $0^-$, the convergence bound is mollified. Despite this, in our experiments, we set $\nu = \zeta = \gamma = 0$ (i.e., fixing learning rates and clipping thresholds), which yielded strong empirical performance. Intuitively, this setup corresponds to a continual amplification of informative coordinates and attenuation of uninformative covariates.

**Other Instantiations and Extensions.** As noted previously, we extend our analysis to support an Adagrad-based outer optimizer (Algorithm 5) and provide a convergence guarantee under heavy-tailed noise, detailed in Theorem 6 in Appendix C. In Appendix C.6, we further generalize our framework by incorporating momentum into the first-order stochastic pseudogradient statistics, resulting in an outer optimizer Adam instantiation. While we establish that the expected minimum gradient is asymptotically bounded even under restarting (Theorem 8), proving formal convergence to 0 remains an open challenge. The difficulty arises from the moving average applied to the first moment, which retains all historical gradient information and complicates the analytical proof structure. We also extend convergence results for certain instantiations to allow for *node drop or failures* at each round (Appendix C.2). Our bound further highlights that the dominating terms are influenced by the upper clipping threshold $u_r$, which we tune empirically in

Section 6 by sweeping over a hyperparameter grid defined in Appendix D.7. For this result, we extremize the noise tails such that there $\nexists \alpha$ such that the $\alpha$-moment is finite for $\forall \alpha > 1$, under which $u_t$ stabilizes the gradient dynamics.

# 6. Experiments

We assess the performance of various TailOPT instantiations across a range of empirical tasks, benchmarking them against state-of-the-art algorithms from the literature. Our experiments include synthetic tasks with heavy-tailed noise injection and real-world benchmarks, including GLUE (Wang et al., 2019) for natural language understanding, WMT (Foundation, 2019) for machine translation, Qwen2.5 (Yang et al., 2025) for question answering, and ViT (Dosovitskiy et al., 2021) for image classification. We present a brief summary of each experimental setup in the corresponding subsections. Extended details of the experimental setup, dataset descriptions, and extensive hyperparameter tuning procedures (including the best hyperparameters for each method and dataset) are provided in Appendix D. Our code are publicly available at github.com/sulee3/Heavy_Tails.

## 6.1. Convex Models

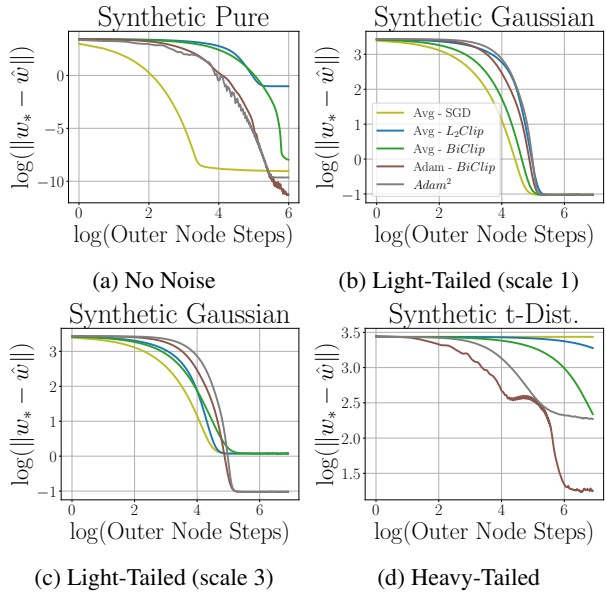

(a) No Noise     (b) Light-Tailed (scale 1)

(c) Light-Tailed (scale 3)     (d) Heavy-Tailed

Figure 1: Impact of heavy-tailed noise on synthetic datasets. Without gradient noise, Avg-SGD achieves good performance (c.f., (a)). As the noise tails grow heavier (from (a) to (d)), the performance of Avg-SGD deteriorates considerably. By contrast, both clipping mechanisms and adaptive updates demonstrate improved performance in learning the ground truth $w_*$, and effectively mitigates the adverse effects of heavy-tailed noise. $BiClip$ as the inner optimizer outperforms other options (Adam, $L_2Clip$, and SGD) with heavy-tailed nose (see (d)). See Appendix D.1 for full results.

Table 1: Test results on the GLUE Benchmark. Metric descriptions are given in Appendix D.3, and the full results are reported as Table 3 in the appendix. Entries marked with 0.0 indicate failure to learn. Top **first**, **second**, and **third** best-performing algorithms are highlighted. For $Adam^2$, preconditioners are transmitted between the inner and outer optimizers, whereas DiLoCo requires maintaining preconditioners on the inner optimizers, both of which incur significant communication or memory overhead than $Bi^2Clip$. Our experiments show that $Bi^2Clip$ achieves the best performance with the smallest overhead.

| Algorithm | MNLI | QNLI | QQP (Acc/F1) | RTE | SST-2 | MRPC (Acc/F1) | CoLA | STS-B (S/P) | Average |
|---|---|---|---|---|---|---|---|---|---|
| **Avg-SGD** (McMahan et al., 2017) | 81.13 | 83.21 | 78.71/78.69 | 57.40 | 90.94 | 67.30/80.52 | 0.0 | 26.76/28.20 | 61.17 |
| **Avg-**$L_2Clip$ (Yang et al., 2022) | 81.82 | 85.68 | 80.00/79.82 | 54.51 | 91.97 | 68.38/81.22 | 0.0 | 41.27/40.96 | 64.15 |
| **Avg-Adagrad** | 84.70 | 88.79 | 87.09/83.34 | 64.26 | 93.34 | 71.56/82.63 | 27.72 | 81.93/81.26 | 76.97 |
| **Avg-Adam** | 84.97 | 89.47 | 87.66/84.09 | 64.62 | **93.80** | 81.86/87.74 | 41.41 | **86.21/86.55** | 80.76 |
| **Avg-**$BiClip$ | 85.08 | 89.45 | 87.83/84.12 | 66.06 | **94.03** | 71.32/82.45 | 41.40 | 84.08/84.48 | 79.12 |
| **Adagrad-SGD** (Reddi et al., 2021) | 82.40 | 86.61 | 82.51/77.68 | 71.48 | 92.08 | 85.53/89.52 | 47.80 | 40.37/42.24 | 72.69 |
| **Adagrad-**$BiClip$ | **85.54** | **90.02** | 88.60/**85.05** | **73.36** | 93.23 | 85.78/89.86 | 48.87 | 84.03/85.90 | 82.75 |
| **RMSProp-SGD** (Reddi et al., 2021) | 84.20 | 88.46 | 87.12/83.30 | **72.56** | 91.85 | 85.50/89.17 | 52.39 | 45.72/41.80 | 74.73 |
| **RMSProp-**$BiClip$ | **85.56** | 89.82 | 88.50/84.44 | 70.75 | **93.69** | 84.80/88.92 | 50.99 | **87.65/87.79** | **82.99** |
| **Adam-SGD** (Reddi et al., 2021) | 82.93 | 86.98 | 85.99/80.87 | 66.78 | 90.71 | 87.01/90.09 | 49.93 | 44.48/41.26 | 73.37 |
| **Adam-**$L_2Clip$ | 82.54 | 86.69 | 85.88/80.72 | 59.92 | 89.67 | 85.29/89.90 | 48.54 | 69.19/67.16 | 76.86 |
| **Adam-**$BiClip$ | 84.26 | 89.20 | **88.64**/84.74 | 69.67 | 92.43 | 86.52/90.09 | **56.12** | 82.83/79.71 | 82.20 |
| $Adam^2$ (Wang et al., 2021b) | 85.11 | 88.87 | **89.04/85.51** | 71.48 | 92.66 | **87.50/91.03** | 52.70 | 84.47/83.82 | 82.93 |
| **DiLoCo** (Douillard et al., 2024) | **85.68** | **89.87** | **88.78/85.19** | 67.87 | 91.89 | **87.99/91.20** | **54.77** | 85.93/84.76 | **83.08** |
| $Bi^2Clip$ | 85.06 | 89.73 | 84.93/83.97 | **76.53** | **93.80** | **89.21/92.44** | **60.08** | **87.07/86.89** | **84.52** |

We designed our convex, synthetic dataset setup to explicitly control and inject heavy-tailed noise, enabling a focused study of its effects. In language tasks, the frequencies of words or tokens typically follows a heavy-tailed distribution, where a small subset of tokens occurs with high frequency, while the majority appear infrequently yet carry significant contextual information. To mirror this phenomenon, emulating a similar setup in Li et al. (2022), we partitioned the input feature space into common and rare features. Specifically, we set the first $p = 10\%$ features (or tokens) from data $X$ as common features, with each feature activated according to a Bernoulli distribution Bern(0.9). The remaining $90\%$ of the features are configured as rare, each sampled from Bern(0.1). The weight vector $w_*$ is drawn from a standard multivariate normal distribution, $w_* \sim \mathcal{N}(0, I_m)$, and the labels are generated as $\hat{y} = Xw_* + \xi_{noise}$. A neural network with model weight $\hat{w}$ is then trained to learn the ground truth $w_*$. A comprehensive explanation of the dataset construction and experimental setup is provided in Appendix D.1. We inject noise $\xi_{noise}$ sampled from a heavy-tailed distribution, which implies that the induced stochastic gradients must be heavy-tailed under the MSE loss. In Figure 1, we sample from the Gaussian and Student $t$ distributions for the non-heavy-tailed and heavy-tailed $\xi_{noise}$, respectively. By default, we multiply the noise by scale 1 unless otherwise specified.

We observe that while SGD demonstrates strong performance in non-noisy settings, its effectiveness diminishes as noise tails become heavier—a scenario where adaptive methods and $BiClip$ excel. Similarly, $L_2Clip$ shows some ability to mitigate heavy-tailed noise but exhibits a comparable decline in performance under heavy-tailed conditions.

## 6.2. Transformer Encoders

To evaluate the effectiveness of our approach, we finetune RoBERTa (Liu et al., 2019) on the General Language Understanding Evaluation (GLUE) benchmark (Wang et al., 2019), a widely-used suite of natural language understanding tasks. The GLUE benchmark includes diverse tasks such as sentiment analysis, sentence similarity, textual entailment, and natural language inference, providing a comprehensive evaluation of model performance across multiple linguistic phenomena. We follow standard finetuning protocols for RoBERTa, leveraging pretrained weights and optimizing task-specific objectives for each dataset in GLUE. Our results demonstrate that $BiClip$ ($Bi^2Clip$) attains competitive or superior performance similar to Adam ($Adam^2$), while only requiring SGD-like memory and compute.

Table 1 presents the performance of the algorithms of interest on the GLUE benchmark. Our results show that $L_2Clip$ enhances performance on real-world data. Adaptive methods further improve upon these results, consistently outperforming $L_2Clip$ (e.g., convergence curves in Figure 2). Notably, the newly proposed clipping method in TailOPT, $BiClip$, demonstrates superior performance compared to $L_2Clip$ and, in some cases, even surpasses Adam during test time (c.f., comparing $Bi^2Clip$ and $Adam^2$), highlighting its potential as an efficient and effective optimizer in real-world applications. Additionally, instantiations of TailOPT achieving $\geq 80\%$ average accuracy generally employ adaptive or adaptive-approximating optimizers across all nodes. In particular, adaptivity on the inner optimizer appears crucial for performance, as SGD-based methods perform considerably worse ($\leq 75\%$). By contrast, both $BiClip$ or Adam reach $\sim 80\%$ even when combined with a

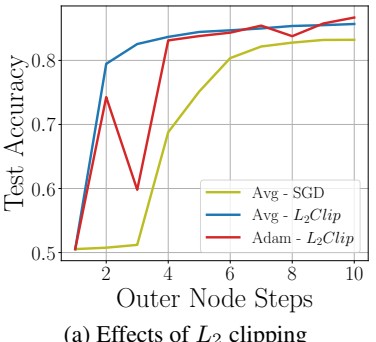 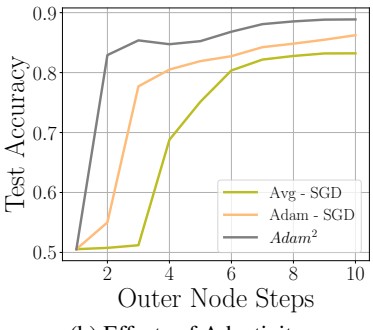 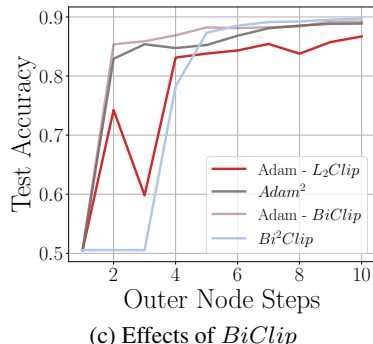

(a) Effects of $L_2$ clipping     (b) Effects of Adaptivity     (c) Effects of $BiClip$

Figure 2: Convergence curves on the QNLI dataset. In (a), we see that $L_2Clip$ (one option of $TailClip$) can help to improve performance under different outer optimizers. Middle Figure (b) demonstrates that adaptivity also helps to mitigate the negative effects of heavy-tailed noise. In all three plots (a)-(c), $L_2Clip$ performs worse than adaptive methods, but the coordinate-wise $BiClip$ optimizer performs comparably or even better than adaptive optimization frameworks, manifesting Adam-like performance. We note that the $Adam^2$ baseline, which applies Adam both in inner and outer optimization, requires transmitting preconditioners of the same size as the model weights to inner optimizers, resulting in substantial communication and memory overhead to deploy. By contrast, $Bi^2Clip$ removes the necessity of preconditioner maintenance, sidestepping this bottleneck entirely.

simple averaging outer optimizer strategy. We include full results of more baselines in Appendix D.3.

### 6.3. Generative Models

We also evaluate TailOPT on machine translation tasks utilizing the WMT datasets, a widely used benchmark for translation research (Foundation, 2019). Specifically, we finetune the T5 (Raffel et al., 2020) generative model on the TED Talks and News Commentary parallel training datasets. The TED Talks dataset, originally sourced from IWSLT 2017 (Cettolo et al., 2017), comprises multilingual translations of TED Talk transcripts, while the News Commentary dataset includes parallel text from news articles across various languages. We report both BLEU and METEOR scores across several variants of source and target language translations in Table 2. An expanded table with a more extensive evaluation is provided in Table 4 in the appendix.

Table 2: Evaluation results on machine translation benchmarks. Metrics reported are formatted as BLEU/METEOR for various language pairs across the TED and News Commentary datasets. The final column represents the average score across all metrics for each algorithm See Table 4 for expanded results.

| Algorithm | TED (en-de) | NewsComm (en-fr) | Avg |
|---|---|---|---|
| **Avg-SGD** | 28.02/58.52 | 30.07/54.13 | 42.68 |
| **Avg-**$L_2Clip$ | 28.99/58.94 | 31.02/56.73 | 43.92 |
| $Adam^2$ | 28.06/58.05 | 30.97/55.85 | 43.23 |
| $Bi^2Clip$ | **29.41/59.18** | **31.79/57.69** | **44.51** |

**Discussions.** For language reasoning benchmarks, the performance differences across algorithmic instantiations are particularly pronounced. While $L_2$ clipping is a common stabilization strategy, it exhibits limited effectiveness. In contrast, coordinate-wise $BiClip$ demonstrates significantly better stability and performance. Moreover, frameworks

aiming to utilize or mimic adaptivity in both the inner and outer optimizers generally achieve superior results, surpassing 80% average performance across all benchmarks. Notably, performance is highly sensitive to the choice of inner optimizers, with SGD and $L_2$ clipping yielding the lowest results. For machine translation, however, the performance variance across different optimizer strategies can be relatively smaller when optimal hyperparameters are selected.

We observe that $Bi^2Clip$ can outperform even $Adam^2$ in some settings. While its design aims to emulate adaptivity under heavy-tailed noise, $BiClip$ exhibits characteristics that can interpolate between non-adaptive and adaptive methods, capturing benefits from both without necessarily fully belonging to either paradigm (Figure 5 in the appendix). Further, $BiClip$ retains the same memory and compute overheads as standard SGD, which cements a highly resource-efficient adaptive approximation.

In addition, federated learning is a special distributed learning paradigm designed to train machine learning models jointly without the transmission of raw data (McMahan et al., 2017; Li et al., 2020a; Wang et al., 2021a). TailOPT can also be applied to federated learning with limited client participation, especially when the local data shards induce heavy-tailed stochastic gradients. See Appendix D.5 for experiments on effects of TailOPT in this setting.

### 6.4. $BiClip$ for Centralized Learning

While we focus on modern distributed settings, the $BiClip$ optimizer itself can be readily used in centralized learning with similar benefits—achieving Adam-like performance without maintaining any gradient statistics. We empirically demonstrate the convergence and test performance of $BiClip$ compared with SGD and Adam on both text and

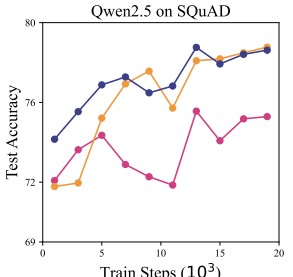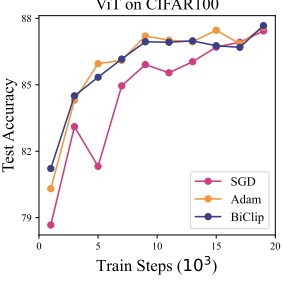

Figure 3: Effects of $BiClip$ in centralized training. We see that $BiClip$ (only relying on two clipping thresholds) achieves the same accuracies and convergence speed as Adam without maintaining any preconditioners or their compressed versions.

image datasets. We finetune Qwen2.5 (Yang et al., 2025) on the SQuAD dataset (Rajpurkar et al., 2016), and finetune a ViT-base model (Dosovitskiy et al., 2021) (pretrained on ImageNet) on CIFAR100 (Krizhevsky et al., 2009). We use the same hyperparameter tuning protocol as in other experiments, discussed in Appendix D.7. In Figure 3, we see that both $BiClip$ and Adam outperform SGD on two tasks in terms of final accuracies and/or convergence speed, whereas $BiClip$ only requires the same memory and compute as SGD, which is significantly more efficient than Adam.

## 7. Conclusion

In this work, we have introduced TailOPT, a framework for efficient heavy-tailed optimization. We have proposed the $BiClip$ optimizer based on coordinate-wise clipping from above and below, which utilizes nearly identical memory and compute resources to vanilla SGD yet manifests Adam-like performance. We have established convergence guarantees for our TailOPT under potentially unbounded variance and provide a thorough empirical evaluation with real-world as well as synthetic datasets. Our experiments indicate that $BiClip$ stabilizes training under heavy-tailed noise and achieves the benefits of efficient adaptive optimization, exceeding the state-of-the-art performance.

Future work could explore automatic selection of $u_t$ and $d_t$ based on initial statistics or bespoke estimators, which could provide more practical solutions. Alternatively, allowing the clipping thresholds to vary depending on coordinate partition subsets (e.g., across tensor slices) may further enhance performance. An extended conclusion with possible future directions is provided in Appendix B.

## Acknowledgements

We thank anonymous reviewers for their feedback. We are grateful to Xiyuan Yang for valuable discussions and contributions to Section 6.4 of this paper.

## Impact Statement

This paper presents work whose goal is to advance the field of machine learning. As our work is primarily focusing on optimization algorithms, we do not believe that there are many potential societal consequences of our work.

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

# Contents

# A. Additional Related Works

**Clipping for Stabilizing Training Dynamics.** Due to its success in stabilizing model updates, gradient clipping has been extensively studied empirically (Gehring et al., 2017; Merity et al., 2018; Peters et al., 2018; Mikolov, 2012) and theoretically (Chezhegov et al., 2024a; Zhang et al., 2020b; Menon et al., 2020; Zhang et al., 2020a; Chen et al., 2020; Koloskova et al., 2023; Gorbunov et al., 2020; Cutkosky & Mehta, 2021). The majority of results study the centralized setting (e.g., Liu et al., 2023; Zhang & Cutkosky, 2022; Parletta et al., 2024; Li & Liu, 2022; Puchkin et al., 2024; Nguyen et al., 2023b; Gorbunov et al., 2024a). Additionally, it was shown that using a constant clip threshold can induce gradient bias, preventing the algorithm from ever converging (Koloskova et al., 2023; Chen et al., 2020). Therefore, some works have attempted to circumvent this issue by debiasing via error feedback (Khirirat et al., 2023; Zhang et al., 2024). Other works in distributed optimization have imposed strong distributional stochastic gradient structures in the analysis. For instance, Qian et al. (2021) assume a well-behaved angular dependence between the stochastic and deterministic gradients throughout training, and Liu et al. (2022) assume symmetric gradient noise, almost surely bounded stochastic gradients, as well as homogeneous data. In contrast with these works, we do not impose any conditions on the noise nor data distributions except for bounded noise $\alpha$-moments for $\alpha \in (1, 2)$. This also sharpens the sensitivity of our bounds to gradient distributions, as $\alpha$ may be selected as the minimal (or close to infimum) $\alpha$-moment value such that the moment is bounded.

There are some recent results studying the dynamics of heavy-tailed clipped-SGD in the distributed setting, without local updates (Sun & Chen, 2024; Gorbunov et al., 2024b; Yu et al., 2024). In particular, Sun & Chen (2024) study the convergence of distributed clipped-sgd for strongly-convex objectives in the absence of a central server, where smaller nodes communicate with their neighbors according to a strongly connected graph. By contrast, Yu et al. (2024) propose 'smooth-clipping' the difference between a local gradient estimator and the local stochastic gradient, which is shown to converge under only the integrability condition (finite first moment) for strongly convex objectives when assuming symmetric noise distributions. The work by Yang et al. (2022) studies $L_2$ clipping with local updates (compared against in our paper as the 'Avg + $L_2 Clip$' baseline in Table 1). Our proposed clipping mechanism, $BiClip$, fundamentally differs from these approaches by incorporating coordinate-wise clipping from both above and below, bringing about benefits of adaptive optimizers. An added advantage of TailOPT is significant communication efficiency, as we do not transmit preconditioners from the inner and outer optimizers under iterative local updates. Our analysis covers both convex and non-convex functions without additional assumptions on the noise distribution except for heavy-tailedness with potentially unbounded variance. It also holds for a variety of adaptive optimizers and different clipping methods.

**Convergence Bounds under Heavy-Tailed Gradient Noise.** There are two primary types of convergence bounds: in-probability bounds (Juditsky et al., 2019b; Davis et al., 2021; Gorbunov et al., 2022; 2020; Sadiev et al., 2023; Cutkosky & Mehta, 2021; Gorbunov et al., 2024a;b) and in-expectation bounds (Wang et al., 2022; Li et al., 2020b; Reddi et al., 2021; Xie et al., 2020; Wang et al., 2020; Karimireddy et al., 2021; Li et al., 2023; 2022). In-probability bounds provide an upper limit on the number of timesteps required to achieve model parameters $x$ such that $\mathbb{P}\{\mathcal{M}(x) \leq \varepsilon\} \geq 1 - \delta$ for a given evaluation metric $\mathcal{M}(x)$ (e.g., $\min_{t \in 1,...,T} |\nabla F(x_t)|$). As $\delta \to 0^+$, the required communication complexity or number of timesteps may diverge. The key challenge is to mitigate this divergence as effectively as possible through novel algorithm designs or refined mathematical analysis, such as by deriving a polylogarithmic dependence on $\delta$ rather than a more severe inverse power-law dependence. A recent work (Sadiev et al., 2023) provides the first high-probability results under unbounded variance for clipped-SGD applied to star-convex or quasi-convex objectives in a distributed setting without local updates. Their analysis reveals an inverse logarithmic dependence on the confidence level.

By contrast, in-expectation bounds complement in-probability bounds by ensuring that convergence to an optimal point is guaranteed under expectations, without a confidence level. However, the majority of such analyses assume a bounded noise variance, typically denoted by an upper bound $G$ (Gorbunov et al., 2020; Parletta et al., 2024; Li & Liu, 2022). Due to this dependence, some works (e.g., those studying high-probability results (Davis et al., 2021; Gorbunov et al., 2020; 2024a)) argue that in-expectation bounds are insensitive to the underlying distributional structures of the stochastic gradients, due to being compressed or approximated away by $G$. Relaxing this assumption is particularly challenging because unbounded noise adds significant uncertainty to controlling model updates. Furthermore, works such as (Lee et al., 2024) have demonstrated that under stochastic gradient descent, unbounded noise is instantaneously transmitted to the model parameters in both centralized and distributed settings, leading to instability and divergence in expectation. Such results elucidate the additional difficulties induced by efforts to remove the bounded gradient condition. In this paper, we develop a more efficient and general TailOPT framework, and study the dynamics of TailOPT under heavy-tailed stochastic gradient distributions. Specifically, we provide the in-expectation convergence guarantees under infinite variance and local

updates for potentially non-convex functions, offering new bounds that are more sensitive to distributional structures of minibatch noise.

## B. Future Directions and Possible Extensions

Efficient and effective tuning of the clipping thresholds $d_t$ and $u_t$ in $BiClip$ remains an open avenue for research. One potential approach is to segment the thresholds into coordinate subsets (e.g., row-wise or column-wise), similar to the memory-efficient partitioning strategies employed in approximate optimizers such as SM3 (Anil et al., 2019). Alternatively, autonomous selection of $u_t$ and $d_t$ based on initial statistics or bespoke estimators could provide practical solutions. Our experiments indicate that coordinate-wise $BiClip$, rather than standard $L_2$ clipping, achieves the benefits of adaptive optimization without incurring any additional memory overhead compared to SGD. Notably, methods like Adam at least double memory usage, whereas $BiClip$ maintains parity with non-adaptive methods. This suggests that uniformly amplifying small updates can contribute to optimization efficiency. Furthermore, layer-wise $BiClip$ can be readily generalized, with proofs extending naturally.

Another intriguing direction for future research is the integration of Adam on top of $BiClip$ to enhance optimization stability in either centralized or distributed training. Notably, when employing the Adam optimizer, some studies apply $L_2$ clipping to gradients prior to plugging in Adam updates to improve stability of the optimization dynamics (Chezhegov et al., 2024b). A natural extension of this approach is to substitute $BiClip$ for $L_2$ clipping before passing updates to the Adam adaptive optimizer. This modification could not only enhance stability but also potentially reduce dependence on the hyperparameters of Adam, offering a more robust optimization framework.

## C. Convergence of TailOPT

In this section, we rigorously analyze the convergence of TailOPT under heavy-tailed noise, beginning with the case of Avg-$L_2Clip$ to enhance readability before progressively advancing to more sophisticated TailOPT variants incorporating $BiClip$ and other adaptive outer optimizers. We first establish the convergence proof for Avg-$L_2Clip$ in Appendix C.1, which serves as the basis for subsequent analyses. The proof for Avg-$L_2Clip$ studies a virtual history of model weights synthesized by inner optimizers, which is inaccessible in real-world settings except when the model updates are communicated to the outer optimizer. However, by analyzing the virtual history, we are able to attain convergence of a moving average of accessible model weights to the optimum, which can be materialized in practice. In Appendix C.2, we extend this proof to settings with partial participation and failing compute nodes, examining the resulting dynamics under heavy-tailed noise. In Appendix C.3, we further generalize the analysis to the $Bi^2Clip$ instantiation, where $BiClip$ is applied to both the inner and outer optimizers. Notably, $Bi^2Clip$ encompasses Avg-$BiClip$ as a special case under specific hyperparameter choices, which in turn subsumes Avg-$L_2Clip$. Finally, in Appendices C.4, C.5, and C.6, we investigate the convergence properties of TailOPT when the outer optimizer is instantiated with Adagrad, RMSProp, and Adam, respectively.

### C.1. Convergence of Avg-$L_2Clip$

We aim to model contemporary, large-scale neural network training across multiple powerful compute nodes (datacenters or GPU clusters), in which data is typically preprocessed IID to optimize for training. However, for fullest generality, we conduct our theoretical analysis in the more challenging, non-IID setting. Our setup is identical to Section 3, with some added notation. We denote $x^*$ to represent the global optimum of $F(x)$ with a minimum value $F^* = F(x^*)$, and additionally, we let $x_i^*$ be the global optimum of $F_i(x) = \mathbb{E}_\xi[F_i(x, \xi)]$, with a minimum value $F_i^* = F(x_i^*)$.

For model weight or stochastic gradient averages, we use the following notation

$$\overline{x}_t = \sum_{i=1}^N p_i x_{i,0}^t, \quad g_t = \sum_{i=1}^N p_i \cdot Clip(c_t, \nabla F_i(x_{i,0}^t, \xi_{i,0}^t)), \quad Clip(c, y) := \min\left\{1, \frac{c}{\|y\|}\right\} y.$$

The use of the notation $x_{i,0}^t$ instead of $x_i^t$ carefully reflects the flow of the proof, which studies a 'virtual synchronization' of the model weights synthesized by the inner optimizer at each time $t \in [T]$ (see Algorithm 2). In other words, we first analyze the virtual average $\overline{x}_t$ which is not materially realized except at outer optimizer synchronization steps, before modifying the proof to procure a moving average of weights which is solely dependent on those communicated to the outer optimizer, which can now be obtained.

We now present some assumptions used in the convergence analysis for this section. We take the model weight projection domain to be $\mathcal{X} = \mathcal{B}(0, B) \subset \mathbb{R}^d$, where $\mathcal{B}(0, B)$ is the closed ball centered at the origin with radius $B$. Clearly, $B > 0$ needs to be large enough to contain $x^*, x_i^* \in \mathcal{X}$ for convergence. However, we note that the convergence analysis holds for $\mathcal{X}$ any large enough compact, convex set.

**Assumption 3 ($\mu$-strong convexity).** For all $x, y \in \mathcal{X}$ and $i \in [N]$, $F_i(x)$ satisfies $F_i(x) \geq F_i(y) + \langle x - y, \nabla F_i(y) \rangle + \mu_i \|x - y\|^2/2$.

Gradient clipping is a widely adopted technique to stabilize model updates by mitigating the impact of large gradients (Menon et al., 2020; Zhang et al., 2020a; Chen et al., 2020; Koloskova et al., 2023). The $Clip(\cdot)$ operator rescales the gradient uniformly to ensure it remains below a predefined threshold. This procedure is mathematically equivalent to applying a dynamically adjusted, lower learning rate when large stochastic gradients are encountered. Another related technique is projection, which operates in the model weight space rather than the gradient space, effectively stabilizing the model parameters themselves instead of acting on the updates. These observations motivate Algorithm 2, which may be interpreted as dynamically modulating the learning rates as well as backtracking toward the model origin $\overline{0}$ when heavy-tailed stochastic gradient updates are realized.

---

**Algorithm 2** Avg-$L_2 Clip$

---

**Require:** Initial model $x_1$, learning rate schedule $\eta_t$, clipping schedule $c_t$
    Synchronization timestep $z \in \mathbb{Z}_{>0}$, projection domain $\mathcal{X}$
1: **for** $t = 1, \ldots, T$ **do**
2:    **for** each node $i \in [N]$ **do**
3:       Draw minibatch gradient $g_{i,0}^t = \nabla F_i(x_{i,0}^t, \xi_{i,0}^t)$
4:       $x_{i,0}^{t+1} \leftarrow x_{i,0}^t - \eta_t \cdot Clip(c_t, g_{i,0}^t)$
5:    **end for**
6:    **if** $t - 1 \in z \cdot \mathbb{Z}_{\geq 0}$ :
7:       $x_{i,0}^{t+1} \leftarrow \text{Proj}_{\mathcal{X}} \left( \sum_{i \in [N]} p_i x_{i,0}^{t+1} \right)$, for $\forall i \in [N]$
8: **end for**

---

Theorem 3 demonstrates that distributed Avg-$L_2 Clip$ converges in expectation under heavy-tailed noise, despite potential clipping-induced bias. We also offer the first proof demonstrating convergence under an extension of these results to accommodate failing nodes for additional utility in Appendix C.2. To proceed with the analysis, we first provide a proposition.

**Proposition 1.** *If $F_i(x)$ is $\mu$-strongly convex (or $L$-smooth), then so is $F_i(x, \xi)$ for the identical $\mu$ (or $L$).*

While clipping offers the benefit of stabilization, it introduces complexities that complicate the convergence analysis. In particular, clipping induces a non-zero bias on the stochastic gradients, rendering them to be no longer unbiased estimators of the true gradient. Furthermore, unlike in previous analyses, our work also considers scenarios involving distributions with infinite variance, where the clipping bias is exacerbated by the presence of heavy tails. Despite these challenges, Theorem 3 demonstrates that with appropriately chosen (increasing) clipping and (decreasing) learning rate schedules, convergence of Algorithm 2 is nevertheless attainable in expectation.

**Theorem 3.** *Let Assumptions 1-3 hold, and the clipping threshold in Avg-$L_2 Clip$ (Algorithm 2) satisfy $c_t = c\eta_t^\gamma$ for $c > 0$ and $1/2 > \gamma > 0$. Decay the learning rate with schedule $\eta_t = r/(t + 1)$ for $r > 2/\mu$, where $\mu = \min_{k \in [N]} \mu_k$ and $L = \max_{k \in [N]} L_k$. Then, we have for $\tilde{x}_T := \sum_{t=1}^T 2t\mathbb{E}[\overline{x}_t]/T(T + 1)$ that*

$$F(\tilde{x}_T) - F(x^*) \leq \Psi_1 + \Psi_2 + \Psi_3 + \Psi_4,$$

*where*

$$\Psi_1 = \frac{rc^2 T^{2\gamma+1}}{(2\gamma+1)T(T+1)},$$

$$\Psi_2 = \frac{(M^\alpha + B^\alpha)^2 c^{2-2\alpha}(T^{(2-2\alpha)\gamma+2}+1)}{(\mu - 2/r)((2-2\alpha)\gamma+1)T(T+1)},$$

$$\Psi_3 = \frac{c^{2-\alpha} rzu(M^\alpha + B^\alpha)LT^{(2-\alpha)\gamma+1}}{(\mu - 2/r)((2-\alpha)\gamma+1)T(T+1)},$$

$$\Psi_4 = \frac{r^2 c^2 z^2 u^2 L^2 (T^{2\gamma}+1)}{2\gamma(\mu - 2/r)T(T+1)}.$$

*Here, we have used the notation*

$$M = \sqrt{\max_{k\in[N], x\in\tilde{\mathcal{X}}} \frac{2L^2}{\mu}(F_i(x) - F_i(x_i^*))}, \quad \alpha = \min_{k\in[N]} \alpha_k, \quad B = \max_{k\in[N]} B_k, \quad u = \frac{z+1}{2},$$

*where $\tilde{\mathcal{X}}$ is a compact domain constructed by a uniformly closed extension of $\mathcal{X}$ with $L_2$ distance $\sum_{t=1}^{z} rct^{\gamma-1}$.*

*Proof.* Let us bound the distance between the averaged model weights $\overline{x}_t$ and the global optimum $x^*$. Assume that $t \in z \cdot \mathbb{Z}$. We consider the following function

$$f(t) = \|x^* - \mathrm{Proj}_{\mathcal{X}}(\overline{x}_t - \eta_t g_t) + t(-\overline{x}_t + \eta_t g_t + \mathrm{Proj}_{\mathcal{X}}(\overline{x}_t - \eta_t g_t))\|^2,$$

for which

$$f'(0) = 2\langle x^* - \mathrm{Proj}_{\mathcal{X}}(\overline{x}_t - \eta_t g_t), -\overline{x}_t + \eta_t g_t + \mathrm{Proj}_{X}(\overline{x}_t - \eta_t g_t)\rangle.$$

Now, consider the function

$$g(t) = \|(1-t)\mathrm{Proj}_{\mathcal{X}}(\overline{x}_t - \eta_t g_t) + t\mathrm{Proj}_{\mathcal{X}}(x^*) - \overline{x}_t + \eta_t g_t\|$$

By the projective property,

$$g(t) \geq \|\mathrm{Proj}_{X}(\overline{x}_t - \eta_t g_t) - (\overline{x}_t - \eta_t g_t)\|.$$

holds for $t \in [0, 1]$ via convexity of $\mathcal{X}$. Additionally, $g(t)^2$ meets its minimum at $t = 0$. Therefore, we have that $\mathrm{d}g(t)^2/\mathrm{d}t_{t=0} \geq 0$ due to $g(t)^2$ being quadratic with respect to $t$. Noting that $f'(0) = \mathrm{d}g(t)^2/\mathrm{d}t|_{t=0}$, we have that $f(t)$ is monotonically increasing for $t \geq 0$, again due to properties of a quadratic. Then, $f(1) \geq f(0)$ gives that

$$\|\mathrm{Proj}_{\mathcal{X}}(\overline{x}_t - \eta_t g_t) - x^*\|^2 \leq \|\overline{x}_t - \eta_t g_t - x^*\|^2.$$

Therefore, we may conclude

$$\|\overline{x}_{t+1} - x^*\|^2 = \left\|\sum_{i=1}^{N} p_i \mathrm{Proj}_{\mathcal{X}}(\overline{x}_t - \eta_t g_t) - x^*\right\|^2 = \|\mathrm{Proj}_{\mathcal{X}}(\overline{x}_t - \eta_t g_t) - x^*\|^2$$

$$\leq \|\overline{x}_t - \eta_t g_t - x^*\|^2 = \|\overline{x}_t - x^*\|^2 - 2\eta_t\langle\overline{x}_t - x^*, g_t\rangle + \eta_t^2\|g_t\|^2$$

$$= \|\overline{x}_t - x^*\|^2 \underbrace{-2\eta_t\langle\overline{x}_t - x^*, g_t - \nabla F(\overline{x}_t)\rangle}_{A_1} \underbrace{-2\eta_t\langle\overline{x}_t - x^*, \nabla F(\overline{x}_t)\rangle}_{A_2} + \underbrace{\eta_t^2\|g_t\|^2}_{A_3}.$$

Note that the final inequality LHS $\leq$ RHS also holds for $t \notin z \cdot \mathbb{Z}$. In bounding $A_2$, we aim to derive a term that decays $\|\overline{x}_t - x^*\|^2$ by inducing a coefficient $(1 - \tilde{c}\eta_t)\|\overline{x}_t - x^*\|^2$ for some $\tilde{c} > 0$ to be determined. By $\mu$-strong convexity of $F(x)$,

$$F(x^*) \geq F(\overline{x}_t) - \langle\overline{x}_t - x^*, \nabla F_i(\overline{x}_t)\rangle + \frac{\mu}{2}\|x^* - \overline{x}_t\|^2$$

$$\implies -(F(\overline{x}_t) - F(x^*)) - \frac{\mu}{2}\|\overline{x}_t - x^*\|^2 \geq -\langle\overline{x}_t - x^*, \nabla F(\overline{x}_t)\rangle.$$

To bound $A_1$, we consider conditional expectations

$$-2\eta_t \langle \overline{x}_t - x^*, \mathbb{E}_t[g_t] - \nabla F(\overline{x}_t) \rangle \leq 2\eta_t \|\overline{x}_t - x^*\| \|\mathbb{E}_t[g_t] - \nabla F(\overline{x}_t)\|,$$

where $\mathbb{E}_t[\cdot]$ conditions on all realizations up to time $t$. Unraveling definitions gives

$$
\begin{aligned}
\|\mathbb{E}_t[g_t] - \nabla F(\overline{x}_t)\| &= \| \sum_{i \in [N]} p_i (\mathbb{E}_t[Clip(c_t, \nabla F_i(x_{i,0}^t, \xi_{i,0}^t))] - \nabla F_i(x_{i,0}^t) + \nabla F_i(x_{i,0}^t) - \nabla F_i(\overline{x}_t)) \| \\
&\leq \sum_{i \in [N]} p_i \|\mathbb{E}_t[Clip(c_t, \nabla F_i(x_{i,0}^t, \xi_{i,0}^t)) - \nabla F_i(x_{i,0}^t, \xi_{i,0}^t))]\| + \sum_{i \in [N]} p_i \|\nabla F_i(x_{i,0}^t) - \nabla F_i(\overline{x}_t)\| \quad (3) \\
&\leq \sum_{i \in [N]} p_i \underbrace{\mathbb{E}_t[\|Clip(c_t, \nabla F_i(x_{i,0}^t, \xi_{i,0}^t)) - \nabla F_i(x_{i,0}^t, \xi_{i,0}^t))\|]}_{A_4} + \sum_{i \in [N]} p_i L \|x_{i,0}^t - \overline{x}_t\|,
\end{aligned}
$$

where the second line used Jensen and triangle inequality, and the third line used $L$-smoothness as well as Jensen. Now, we note that clipping biases the expectation in $A_4$, and we seek to ease out a measure of the clipping bias. For this purpose, we quantify the $\alpha$-moment of the stochastic gradient:

$$2^\alpha \mathbb{E}_t \left\| \frac{\nabla F_i(x) + \xi_{i,0}^t}{2} \right\|^\alpha \leq 2^{\alpha-1} \left( \mathbb{E}_t \|\nabla F_i(x)\|^\alpha + \mathbb{E}_t \|\xi_{i,0}^t\|^\alpha \right) \leq 2^{\alpha-1} \left( \|\nabla F_i(x)\|^\alpha + B_i^\alpha \right).$$

Here, we have used the notation $B_i < \infty$ for readability, but strictly speaking this is not identical to the $B_i$ given in Assumption 2 as $\alpha := \min_{i \in [N]} \alpha_i$. Finally, the projection in each outer optimizer synchronization step ensures that the $x_{i,0}^t$ remain in a compact set $\tilde{\mathcal{X}}$. Therefore, to bound gradients, we use $L$-smoothness and $\mu$-strong convexity of $F_i(x)$ as follows:

$$\|\nabla F_i(x)\|^2 \leq L^2 \|x - x_i^*\|^2,$$

where $x_i^*$ is the optimum of $F_i(x)$. Then, convexity gives that

$$F_i(x) \geq F_i(x_i^*) + \frac{\mu}{2}\|x - x_i^*\|^2,$$

from which we conclude

$$\|\nabla F_i(x)\|^2 \leq \frac{2L^2}{\mu}(F_i(x) - F_i(x_i^*)) \leq M^2 := \max_{k \in [N], x \in \tilde{\mathcal{X}}} \frac{2L^2}{\mu}(F_i(x) - F_i(x_i^*)). \quad (4)$$

Piecewise continuity of $F_i(x)$ is clear due to the existence of $\nabla F_i(x)$. Therefore,

$$\mathbb{E}_t \left\| \nabla F_i(x_{i,0}^t) + \xi_{i,0}^t \right\|^\alpha \leq \frac{(M^\alpha + B^\alpha)}{2}.$$

Now, note that if $\|\nabla F_i(x_{i,0}^t, \xi_{i,0}^t)\| \leq c_t$, clipping has no effect in $A_4$. Thus, we focus on the case $\|\nabla F_i(x_{i,0}^t, \xi_{i,0}^t)\| > c_t$. Additionally, clipping only downscales each stochastic gradient by a scalar, which preserves direction. Therefore,

$$
\begin{aligned}
A_4 &= \mathbb{E}_t \left[ \|Clip(c_t, \nabla F_i(x_{i,0}^t, \xi_{i,0}^t)) - \nabla F_i(x_{i,0}^t, \xi_{i,0}^t))\| \cdot \chi \left( \|\nabla F_i(x_{i,0}^t, \xi_{i,0}^t)\| > c_t \right) \right] \\
&\leq \mathbb{E}_t \left[ \|\nabla F_i(x_{i,0}^t, \xi_{i,0}^t)\| \cdot \chi \left( \|\nabla F_i(x_{i,0}^t, \xi_{i,0}^t)\| > c_t \right) \right] \quad (5) \\
&\leq \mathbb{E}_t \left[ \|\nabla F_i(x_{i,0}^t, \xi_{i,0}^t)\|^\alpha \cdot \|\nabla F_i(x_{i,0}^t, \xi_{i,0}^t))\|^{1-\alpha} \cdot \chi \left( \|\nabla F_i(x_{i,0}^t, \xi_{i,0}^t)\| > c_t \right) \right] \leq (M^\alpha + B^\alpha)c_t^{1-\alpha}.
\end{aligned}
$$

Putting these inequalities together, we obtain as an intermediary step for $a > 0$:

$$
\begin{aligned}
A_1 &\leq 2\eta_t \|\overline{x}_t - x^*\| ((M^\alpha + B^\alpha)c_t^{1-\alpha} + \sum_{i \in [N]} p_i L \|x_{i,0}^t - \overline{x}_t\|) \\
&\leq \mu a \eta_t \|\overline{x}_t - x^*\|^2 + \frac{\eta_t}{\mu a} ((M^\alpha + B^\alpha)c_t^{1-\alpha} + L \sum_{i \in [N]} p_i \|x_{i,0}^t - \overline{x}_t\|)^2.
\end{aligned}
$$

Thus, our next step is to ease out $\|x_{i,0}^t - \overline{x}_t\| = \mathcal{O}(\eta_t)$. For this purpose, our intuition is that the drift in model weights from local updates are bounded by the update size, as well as by taking a maximum of $z$ local steps after global synchronization.

Therefore, we naturally consider the timestep $t_s(t)$ of the latest synchronization round up to $t$, and observe that if the random variable $X := x_{i,0}^t - \overline{x}_{t_s}$, then $\mathbb{E}_k[X] = \overline{x}_t - \overline{x}_{t_s}$. Noting that the variance of $X$ is no greater than its second moment, we proceed as follows via telescoping:

$$
\begin{aligned}
\mathbb{E}_k[\|x_{i,0}^t - \overline{x}_t\|^2] = \sum_{i=1}^N p_i \|x_{i,0}^t - \overline{x}_t\|^2 &= \mathbb{E}_k[\|X - \mathbb{E}_k[X]\|^2] \\
&\leq \mathbb{E}_k[\|X\|^2] = \sum_{i=1}^N p_i \|x_{i,0}^t - \overline{x}_{t_s}\|^2 \\
&= \sum_{i=1}^N p_i \left\| x_{i,0}^t + \sum_{\tilde{t}=t_s+1}^{t-1} (-x_{\tilde{t}}^k + x_{\tilde{t}}^k) - \overline{x}_{t_s} \right\|^2 \\
&\leq \sum_{i=1}^N p_i (t - t_s - 1)^2 \max_{t' \in [t_s, t]} \eta_{t'}^2 \|Clip(c_t', \nabla F_i(x_{i,0}^t, \xi_{i,0}^t))\|^2 \\
&\leq \sum_{i=1}^N p_i z^2 \eta_{t_s}^2 c_t^2 = z^2 \eta_{t_s}^2 c_t^2 \leq z^2 u^2 \eta_t^2 c_t^2.
\end{aligned}
\tag{6}
$$

The final inequality was obtained by noting that $\eta_t \to 0^+$ monotonically from above and that $c_t \geq c_{t-1}$. The above holds for all $t \in \mathbb{Z}_{\geq 0}$, as if $t$ is a synchronization step, $\mathbb{E}_k \|x_{i,0}^t - \overline{x}_t\|^2 = 0$. The final inequality used that the monotonic near-harmonic decay of $\eta_t$ allows $\eta_{t_s} \leq u \eta_t$ for $u = (z+1)/2$. Finally, by Cauchy-Schwartz,

$$
\left( \sum_{i=1}^N p_i \|\overline{x}_t - x_{i,0}^t\| \right)^2 \leq \left( \sum_{i=1}^N p_i \right) \left( \sum_{i=1}^N p_i \|\overline{x}_t - x_{i,0}^t\|^2 \right),
$$

from which we conclude

$$
A_1 \leq \mu a \eta_t \|\overline{x}_t - x^*\|^2 + \frac{\eta_t}{\mu a}((M^\alpha + B^\alpha)c_t^{1-\alpha} + \eta_t c_t z u L)^2
\tag{7}
$$

It now remains to bound $A_3$, which can be done straightforwardly via Jensen:

$$
A_3 = \eta_t^2 \|g_t\|^2 \leq \eta_t^2 \sum_{i=1}^N p_i \left\| Clip(c_t, \nabla F_i(x_{i,0}^t, \xi_{i,0}^t)) \right\|^2 \leq \eta_t^2 c_t^2.
$$

Collecting all inequalities gathered thus far gives the simple form

$$
\mathbb{E}_t[\|\overline{x}_{t+1} - x^*\|^2] \leq (1 - (1-a)\mu\eta_t) \|\overline{x}_t - x^*\|^2 - 2\eta_t (F(\overline{x}_t) - F(x^*)) + \eta_t^2 c_t^2 + \frac{\eta_t}{\mu a}((M^\alpha + B^\alpha)c_t^{1-\alpha} + \eta_t c_t z u L)^2,
$$

which under tower law of expectations is amenable to telescoping. Intuitively, we want to control the learning rate and form a quadratically decaying average on the LHS, which by Jensen and convexity will give a desired near-optimal point. The rest is a matter of carefully easing out a rate schedule that enables averaging, which also converges. Rearranging gives

$$
\begin{aligned}
\mathbb{E}[F(\overline{x}_t)] - F(x^*) \leq{} & \frac{(\eta_t^{-1} - (1-a)\mu)}{2} \mathbb{E}[\|\overline{x}_t - x^*\|^2] - \frac{1}{2\eta_t} \mathbb{E}[\|\overline{x}_{t+1} - x^*\|^2] + \frac{\eta_t c_t^2}{2} \\
& + \frac{1}{2\mu a}((M^\alpha + B^\alpha)^2 c_t^{2-2\alpha} + 2(M^\alpha + B^\alpha)c_t^{2-\alpha}\eta_t z u L + \eta_t^2 c_t^2 z^2 u^2 L^2).
\end{aligned}
\tag{8}
$$

Letting $\eta_t = r/(t+1)$, $a = 1 - 2/(r\mu)$ for $r > 2/\mu$, we have

$$
\begin{aligned}
t\mathbb{E}[F(\overline{x}_t)] - tF(x^*) \leq{} & \frac{t(t-1)}{2} \mathbb{E}[\|\overline{x}_t - x^*\|^2] - \frac{(t+1)t}{2} \mathbb{E}[\|\overline{x}_{t+1} - x^*\|^2] + \frac{t\eta_t c_t^2}{2} \\
& + \frac{t}{2\mu a}((M^\alpha + B^\alpha)^2 c_t^{2-2\alpha} + 2(M^\alpha + B^\alpha)c_t^{2-\alpha}\eta_t z u L + \eta_t^2 c_t^2 z^2 u^2 L^2)
\end{aligned}
\tag{9}
$$

Setting $c_t = ct^\gamma$ for $1/2 > \gamma > 0$, $c > 0$ gives after telescoping

$$\frac{\sum_{t=1}^{T} 2t\mathbb{E}[F(\overline{x}_t)]}{T(T+1)} - F(x^*) \leq \frac{rc^2 \sum_{t=1}^{T} t^{2\gamma}}{T(T+1)} + \frac{(M^\alpha + B^\alpha)^2 c^{2-2\alpha} \sum_{t=1}^{T} t^{(2-2\alpha)\gamma+1}}{(\mu - 2/r)T(T+1)}$$
$$+ \frac{c^{2-\alpha} rzu(M^\alpha + B^\alpha)L \sum_{t=1}^{T} t^{(2-\alpha)\gamma}}{(\mu - 2/r)T(T+1)} + \frac{r^2 c^2 z^2 u^2 L^2 \sum_{t=1}^{T} t^{2\gamma-1}}{(\mu - 2/r)T(T+1)}.$$

Standard integral bounds give

$$\frac{\sum_{t=1}^{T} 2t\mathbb{E}[F(\overline{x}_t)]}{T(T+1)} - F(x^*) \leq \frac{rc^2 T^{2\gamma+1}}{(2\gamma+1)T(T+1)} + \frac{(M^\alpha + B^\alpha)^2 c^{2-2\alpha}(T^{(2-2\alpha)\gamma+2} + 1)}{(\mu - 2/r)((2-2\alpha)\gamma+1)T(T+1)}$$
$$+ \frac{c^{2-\alpha} rzu(M^\alpha + B^\alpha)LT^{(2-\alpha)\gamma+1}}{(\mu - 2/r)((2-\alpha)\gamma+1)T(T+1)} + \frac{r^2 c^2 z^2 u^2 L^2(T^{2\gamma}+1)}{2\gamma(\mu - 2/r)T(T+1)}.$$

Finally, note that by Jensen and convexity, the left hand side is lower bounded by

$$0 \leq F(\tilde{x}_T) - F(x^*) \leq \frac{\sum_{t=1}^{T} 2t\mathbb{E}[F(\overline{x}_t)]}{T(T+1)} - F(x^*)$$

where $\tilde{x}_T := \sum_{t=1}^{T} 2t\mathbb{E}[\overline{x}_t]/T(T+1)$ is a quadratically decaying average. This concludes the proof. It is straightforward to extend to the case in which the learning rate is scheduled to decay in each outer optimizer synchronization step instead of at each local step, by letting $\eta_t = r/(\lceil t/z \rceil + 1)$ in equation (8). □

The value of the moment $\alpha$ has a significant impact on the convergence behavior. When $\alpha$ is close to 1, the convergence becomes substantially slower due to the heavy-tailed nature of the induced stochastic gradients and the increased variance they introduce. Conversely, when $\alpha$ approaches 2, the variance is more controlled, leading to faster convergence rates. Importantly, our results demonstrate that even in the presence of infinite variance (i.e., $1 < \alpha < 2$), convergence can still be achieved, showcasing the robustness of the clipping approach under extreme heavy-tailed conditions.

The averages $\overline{x}_t$ are virtual constructs used for theoretical analysis of Algorithm 2, which are not accumulated during the execution phase. That is, these quantities are only available at the outer optimizer synchronization steps, $t \in z \cdot \mathbb{Z}_{\geq 0}$, and are not collected otherwise (as models are not saved for every local timestep prior to synchronization). As a result, the application of Avg-$L_2Clip$ creates a virtual history on the compute node models, where the aggregation of ephemeral model weights can theoretically induce convergence. However, in practice, this conflicts with the use of local epochs for communication efficiency, necessitating adjustments to the convergence theorem. This leads to the development of Corollary 3.

**Corollary 3.** *Let the conditions of Theorem 3 hold. Then, we have that*

$$\mathbb{E}\left[F\left(\frac{\sum_{t\in Z}(t-1)\overline{x}_t}{\sum_{t\in Z}(t-1)}\right)\right] - F(x^*) \leq \frac{(T+1)z}{(T-z)}(\psi_1 + \psi_2 + \psi_3 + \psi_4),$$

*where the $\psi_i$ are defined as in the statement of Theorem 3 and $Z$ is the set of all outer optimizer synchronization steps.*

*Proof.* We may start with equation (9), where we use the same notation as the proof of Theorem 3. Recall that $0 \leq F(x) - F(x^*)$ for all $x$. Therefore, we have for $Z = \{1, z+1, \ldots, z\lfloor T/z \rfloor + 1\}$ for $T \notin z \cdot \mathbb{Z}$ and $Z = \{1, z+1, \ldots, z(\lfloor T/z \rfloor - 1) + 1\}$ otherwise,

$$\sum_{t\in Z} t\left(\mathbb{E}[F(\overline{x}_t)] - F(x^*)\right) \leq \sum_{t\in[T]} \left(\frac{t(t-1)}{2}\mathbb{E}[\|\overline{x}_t - x^*\|^2] - \frac{(t+1)t}{2}\mathbb{E}[\|\overline{x}_{t+1} - x^*\|^2]\right)$$
$$+ \sum_{t\in[T]} \frac{t\eta_t c_t^2}{2} + \sum_{t\in[T]} \frac{t}{2\mu a}\left((M^\alpha + B^\alpha)^2 c_t^{2-2\alpha} + 2(M^\alpha + B^\alpha)c_t^{2-\alpha}\eta_t zuL + \eta_t^2 c_t^2 z^2 u^2 L^2\right).$$

Noting that

$$\sum_{t\in Z}(t-1)\left(\mathbb{E}[F(\overline{x}_t)] - F(x^*)\right) \leq \sum_{t\in Z} t\left(\mathbb{E}[F(\overline{x}_t)] - F(x^*)\right),$$

$$\frac{(T-z)T}{2z} \leq \frac{z(\lceil T/z\rceil - 1)\lceil T/z\rceil}{2} \leq \frac{z(\lfloor T/z\rfloor + 1)\lfloor T/z\rfloor}{2},$$

we obtain

$$\mathbb{E}\left[F\left(\frac{\sum_{t\in Z}(t-1)\overline{x}_t}{\sum_{t\in Z}(t-1)}\right)\right] - F(x^*) \leq \frac{(T+1)z}{(T-z)}\left(\psi_1 + \psi_2 + \psi_3 + \psi_4\right).$$

As before, extension to the case where the learning rate decays at each outer optimizer synchronization step is straightforward. Therefore, the asymptotic convergence rate is identical that give in Theorem 3. $\qquad\square$

In particular, we immediately deduce the following corollary.

**Corollary 4.** *Let the conditions of Theorem 3 hold. Then, Avg-$L_2Clip$ converges under heavy-tailed noise with rate $\mathcal{O}(T^{-(2\alpha-2)\gamma})$ for $\gamma \in (0, 1/2)$. That is, the algorithm recovers a point $\widetilde{x}_T$ which is materialized during training such that*

$$\mathbb{E}[F(\widetilde{x}_T)] - F(x^*) \lesssim \mathcal{O}(T^{-(2\alpha-2)\gamma}).$$

*Proof.* The maximal rate of convergence is attained due to the dominating term $\Psi_2$. We note that in the limit $\alpha \to 1^+$, which revokes the integrability of the heavy-tailed noise, convergence is nullified. $\qquad\square$

### C.2. Dynamics of Avg-$L_2Clip$ under Failing Compute Nodes

Node failures can happen in both datacenter training (Yu et al., 2019) or federated learning (Li et al., 2020a). Consequently, it is crucial to conduct a theoretical performance analysis of Avg-$L_2Clip$ within environments to accommodate the presence of failing compute nodes or partial participation.

In this setting, we modify Line 2 of Avg-$L_2Clip$ to sample a subset of participating nodes, $\mathcal{S} \subset [N]$, rather than selecting $S = [N]$. Additionally, normalized averaging is performed across only the participating compute nodes in Line 7. However, we assume all compute nodes remain active, as described in Algorithm 3 below. We refer to this algorithm as $SludgeClip$ to emphasize its impracticality, despite being functionally equivalent to a variant of Avg-$L_2Clip$ aggregating over a subset of nodes. By analyzing $SludgeClip$, we are able to establish convergence of Avg-$L_2Clip$ when several datacenters or compute nodes fail to partake in training.

---

**Algorithm 3** $SludgeClip$

---

**Require:** Initial model $x_1$, learning rate schedule $\eta_t$, clipping schedule $c_t$
    Synchronization timestep $z \in \mathbb{Z}_{>0}$, projection domain $\mathcal{X}$
1: **for** $t = 1, \ldots, T$ **do**
2:    Sample participating compute nodes $S \subset [N]$ according to $p_i$
3:    **for** each node $i \in [N]$ **do**
4:       Draw minibatch gradient $g_{i,0}^t = \nabla F_i(x_{i,0}^t, \xi_{i,0}^t)$
5:       $x_{t+1}^k \leftarrow x_t^k - \eta_t \cdot L_2Clip(c_t, g_{i,0}^t)$
6:    **end for**
7:    **if** $t - 1 \in z \cdot \mathbb{Z}_{\geq 0}$ :
8:       $x_{t+1}^k \leftarrow \text{Proj}_{\mathcal{X}}\left((\sum_{i'\in S} p_{i'})^{-1}\sum_{i'\in S} p_{i'} x_{t+1}^{i'}\right)$, for $\forall k \in [N]$
9: **end for**

---

**Theorem 4.** *Let the clipping threshold in $SludgeClip$ (Algorithm 3) satisfy $c_t = c\eta_t^\gamma$ for $c > 0$ and $1/2 > \gamma > 0$. Decay the learning rate with schedule $\eta_t = r/(t+1)$ for $r > 2/\mu$. If the sampling scheme preserves the global objective[2], that is,*

$$\mathbb{E}_S\left[\sum_{i\in[S]} p_i F_i(x)\right] = \sum_{i\in[N]} p_i F_i(x) = F(x),$$

*then we have for $Z$ the set of synchronization steps up to $T$ that*

$$\mathbb{E}[F(\widetilde{x}_T')] - F(x^*) := \mathbb{E}\left[F\left(\frac{\sum_{t\in Z}(t-1)\overline{x}_t}{\sum_{t\in Z}(t-1)}\right)\right] - F(x^*) \leq z \cdot \mathcal{O}\left(t^{-\omega}\right),$$

---

[2]For example, $p_i = 1/N$ satisfies this condition. That is, given any selection of $p_i$ and $F_i(x)$, we may rescale the local objectives $F_i(x)$ such that $p_i = 1/N$ by controlling the influence of each local gradient update.

*where now $\omega$ satisfies*

$$\omega = \min\{1 - 2\gamma, 1 - (2 - 2\alpha)\gamma, 1 - (2 - \alpha)\gamma, 2 - 2\gamma, 2\gamma(\alpha - 1)\}.$$

*If the subsampling scheme fails to preserve the global objective (e.g., by sampling only a strict subset of avaliable nodes repeatedly), then Algorithm 3 asymptotes toward biased minimizer points within an increasing region determined by the clipping threshold $\mathbb{E}\left[F\left(\tilde{x}_T'\right)\right] - F(x^*) \lesssim \mathcal{O}(t^{2\gamma}).$*

We note that convergence is not clearly guaranteed when subsampling procedures violate the global objective in expectation. Specifically, we evaluate the algorithm's output relative to $x^*$, the global optimum of the true objective $F(x)$. However, when subsampling alters the objective, the algorithm no longer optimizes for $F(x)$, thereby clearly undermining convergence toward $x^*$. We then measure the propensity of the algorithm output to $x^*$, the global optimum of the true objective $F(x)$ which is no longer the objective of the subsampled algorithm.

*Proof.* We first analyze the case in which the subsampling strategy preserves the correct global objective, which allows for convergence to $x^*$. Recall that $SludgeClip$-SGD was constructed to allow the analysis for non-synchronization steps to be analogous to full-participation Avg-$L_2Clip$. Therefore, we focus on outer optimizer synchronization steps while incorporating the elements of the previous analysis for Theorem 3. We now use the following notation for subsampled averages of participating compute node devices:

$$\tilde{x}_t = \frac{\sum_{i \in S} p_i x_{i,0}^t}{\sum_{i \in S} p_i}, \quad \tilde{g}_t = \frac{\sum_{i \in S} p_i \cdot Clip(c_t, \nabla F_i\left(x_{i,0}^t, \xi_{i,0}^t\right))}{\sum_{i \in S} p_i}.$$

For added clarity, we denote $g_t$ as $\overline{g}_t$ to indicate that normalized averages are taken over all inner compute nodes, and not solely participating nodes as in $\tilde{g}_t$. Then for $t + 1$ a synchronization step, we have that

$$\|\tilde{x}_{t+1} - x^*\|^2 \le \|\tilde{x}_t - x^* - \eta_t \tilde{g}_t\|^2 = \|\overline{x}_t + (\tilde{x}_t - \overline{x}_t) - x^* - \eta_t \tilde{g}_t + (\eta_t \overline{g}_t - \eta_t \overline{g}_t)\|^2$$

$$= \|\overline{x}_t - x^*\|^2 + 2\langle \overline{x}_t - x^*, \underbrace{\tilde{x}_t - \overline{x}_t - \eta_t \tilde{g}_t + (\eta_t \overline{g}_t - \eta_t \overline{g}_t)}_{B_1}\rangle + B_1^2$$

$$\le \|\overline{x}_t - x^*\|^2 \underbrace{-2\eta_t \langle \overline{x}_t - x^*, \overline{g}_t - \nabla F(\overline{x}_t)\rangle}_{A_1} \underbrace{-2\eta_t \langle \overline{x}_t - x^*, \nabla F(\overline{x}_t)\rangle}_{A_2}$$

$$\underbrace{+2\langle \overline{x}_t - x^*, \tilde{x}_t - \overline{x}_t\rangle}_{B_2} \underbrace{+2\eta_t \langle \overline{x}_t - x^*, \overline{g}_t - \tilde{g}_t\rangle}_{B_3} \underbrace{+\|\tilde{x}_t - \overline{x}_t - \eta_t \tilde{g}_t\|^2}_{B_4}.$$

In this form, the $A_i$ terms are therefore shared with the previous analysis, and $A_2$ may be bounded by $\mu$-strong convexity as before. This gives that

$$A_2 \le -\mu\eta_t \|\overline{x}_t - x^*\|^2 - 2\eta_t \left(F(\overline{x}_t) - F(x^*)\right).$$

$A_1$ is once again bounded under conditional expectations $\mathbb{E}_t[\cdot]$ by equation (7), though with a different value of $a' > 0$ than in the previous proof,

$$A_1 \le \mu a' \eta_t \|\overline{x}_t - x^*\|^2 + \frac{\eta_t}{\mu a'}((M^\alpha + B^\alpha)c_t^{1-\alpha} + \eta_t c_t zuL)^2. \tag{7}$$

Now, as $B_2$ is eliminated under expectations under subsampling, we focus on the remaining terms. It is clear that we must bound and $\|\overline{g}_t - \tilde{g}_t\|$ to proceed. Intuitively, this is controlled by normalized averages and model drift across participating nodes. Therefore, we consider the nearest or most recent synchronization timestep $t_s(t)$ as before and rearrange to incorporate elements of our previous analysis. Assuming interchangeability between the integrals $\mathbb{E}_S$ (integrating over the randomness of node subsampling) and $\mathbb{E}_t$ (integrating over randomness of $\xi_{i,0}^t$),

$$\|\mathbb{E}_t\left[\mathbb{E}_S[\tilde{g}_t] - \overline{g}_t\right]\| = \left\|\mathbb{E}_t\left[\mathbb{E}_S\left[\sum_{i \in S} \frac{p_i}{\sum_{i' \in S} p_{i'}}(Clip(c_t, \nabla F_i\left(x_{i,0}^t, \xi_{i,0}^t\right)) - \nabla F_i(\overline{x}_t))\right] - (\overline{g}_t - \nabla F(\overline{x}_t))\right]\right\|$$

$$= \left\|\mathbb{E}_S\left[\mathbb{E}_t\left[\sum_{i \in S} \frac{p_i}{\sum_{i' \in S} p_{i'}}(Clip(c_t, \nabla F_i\left(x_{i,0}^t, \xi_{i,0}^t\right)) - \nabla F_i(\overline{x}_t, \xi_{i,0}^t))\right]\right] - \mathbb{E}_t\left[\overline{g}_t - \nabla F(\overline{x}_t)\right]\right\|$$

$$\le \mathbb{E}_S\left[\sum_{i \in S} \frac{p_i}{\sum_{i' \in S} p_{i'}}\mathbb{E}_t[\|Clip(c_t, \nabla F_i\left(x_{i,0}^t, \xi_{i,0}^t\right)) - \nabla F_i(\overline{x}_t, \xi_{i,0}^t)\|]\right] + \mathbb{E}_t[\|\overline{g}_t - \nabla F(\overline{x}_t)\|] \le 2(M^\alpha + B^\alpha)c_t^{1-\alpha}$$

where to obtain the final line we used Jensen and an analogous reasoning as in equation (5).

Therefore, we have for $b > 0$ that

$$B_3 \le b\eta_t\|\overline{x}_t - x^*\|^2 + 4\eta_t(M^\alpha + B^\alpha)^2 c_t^{2(1-\alpha)}.$$

It now remains to bound $B_4$, which can be done straightforwardly:

$$B_4 \le 2\|\tilde{x}_t - \overline{x}_t\|^2 + 2\eta_t^2\|\tilde{g}_t\|^2 \le 4z^2u^2\eta_t^2 c_t^2 + 2\eta_t^2 c_t^2.$$

Collecting all inequalities gathered under the tower law of expectation, we have

$$\mathbb{E}[\|\tilde{x}_{t+1} - x^*\|^2] \le (1 - ((1-a)\mu + b)\eta_t)\mathbb{E}[\|\overline{x}_t - x^*\|^2] - 2\eta_t\mathbb{E}\left[F(\overline{x}_t) - F(x^*)\right]$$
$$+ \frac{\eta_t}{\mu a}((M^\alpha + B^\alpha)c_t^{1-\alpha} + \eta_t c_t zuL)^2 + 4z^2u^2\eta_t^2 c_t^2 + 2\eta_t^2 c_t^2 + 4\eta_t(M^\alpha + B^\alpha)^2 c_t^{2(1-\alpha)}.$$

Recall the learning rate schedule $\eta_t = r/(t+1)$, while setting $a', b$ such that $r((1-a')\mu + b) = 2$. Then, we have for $Z$ the set of all synchronization steps,

$$\sum_{t+1\in Z} t(\mathbb{E}[F(\overline{x}_t)] - F(x^*)) \le \sum_{t+1\in Z} \left[\frac{t(t-1)}{2}\mathbb{E}[\|\overline{x}_t - x^*\|^2] - \frac{(t+1)t}{2}\mathbb{E}[\|\tilde{x}_{t+1} - x^*\|^2]\right]$$
$$+ \underbrace{\sum_{t+1\in Z} 2(M^\alpha + B^\alpha)^2 t c_t^{2(1-\alpha)}}_{B_5} + \underbrace{\sum_{t+1\in Z} \frac{1}{2\mu a}((M^\alpha + B^\alpha)c_t^{1-\alpha} + \eta_t c_t zuL)^2}_{\sim\Psi_2+\Psi_3+\Psi_4} + \underbrace{\sum_{t+1\in Z} t\eta_t c_t^2(2z^2u^2 + 1)}_{\sim\Psi_1}.$$

For $t+1 \notin Z$, we use the standard telescoping sum in equation (9) while noting that $\tilde{x}_{t+1} = \overline{x}_{t+1}$ due to the synchronization step. We do not repeat mechanical calculation steps here to not obscure the intuitions behind the proof, and instead indicate asymptotically equivalent terms to $\Psi_i$ under $1/(T^2 + T)$ averaging on the right hand side. It remains to bound the residual term $B_5$ under the averaging step, which gives

$$\frac{B_5}{T(T+1)} \lesssim \mathcal{O}(t^{2\gamma(1-\alpha)}),$$

which concludes the proof for the first case.

In the setting in which the subsampling procedure fails to preserve the global objective, we bound $\|\tilde{x}_t - \overline{x}_t\|$ as follows:

$$\|\tilde{x}_t - \overline{x}_t\| = \left\|\sum_{i\in[S]}\left(\frac{\sum_{\tilde{k}\notin[S]} p_{\tilde{k}}}{\sum_{i'\in[S]} p_{i'}}\right)p_i x_{i,0}^t - \sum_{i\notin[S]} p_i x_{i,0}^t\right\|$$
$$\le \sum_{i\in[S]}\left(\frac{\sum_{\tilde{k}\notin[S]} p_{\tilde{k}}}{\sum_{i'\in[S]} p_{i'}}\right)p_i\|x_{i,0}^t - \overline{x}_{t_s}\| + \sum_{i\notin[S]} p_i\|x_{i,0}^t - \overline{x}_{t_s}\| \le 2zu\eta_t c_t,$$

due to triangle inequality and Jensen. That is, by the synchronization step, we have $x_{t_s}^k = \overline{x}_{t_s}, \forall k \in [N]$ via to full available node activation in $SludgeClip$. This gives

$$\|x_{i,0}^t - \overline{x}_{t_s}\| = \left\|x_{i,0}^t + \sum_{t'=t_s+1}^{t-1}(-x_{t'}^k + x_{t'}^k) - \overline{x}_{t_s}\right\| \le \sum_{t'=t_s+1}^{t-1}\|x_{t'}^k - x_{t'-1}^k\| \le zu\eta_t c_t$$

as in equation (6). Similarly, we have by Jensen and convexity of the norm that

$$\|\tilde{g}_t - \overline{g}_t\| \le 2c_t.$$

Therefore, we obtain for $b_1, b_2 > 0$

$$B_2 \le b_1\eta_t\|\overline{x}_t - x^*\|^2 + \frac{1}{b_1\eta_t}\|\tilde{x}_t - \overline{x}_t\|^2 \le b_1\eta_t\|\overline{x}_t - x^*\|^2 + \frac{2z^2u^2c_t^2\eta_t}{b_1},$$
$$B_3 \le b_2\eta_t\|\overline{x}_t - x^*\|^2 + 4\eta_t c_t^2.$$

Following analogous calculations as in the case where the subsampling does not violate the global objective, we arrive at a new residual term

$$\frac{B_6}{T(T+1)} \lesssim \mathcal{O}(t^{2\gamma}),$$

which controls the expansion of the bias due to the incorrect sampling strategy. □

## C.3. Convergence of $Bi^2Clip$

In this section, we analyze the convergence of $Bi^2Clip$ under heavy-tailed noise. By employing $BiClip$ at both the inner and outer optimizers, $Bi^2Clip$ is an efficient algorithm realized by TailOPT that brings about benefits of adaptivity without maintaining gradient (or model update) statistics. Unlike $Adam^2$, which applies Adam at both the inner and outer optimizers, $Bi^2Clip$ achieves comparable or even superior empirical performance while requiring no additional memory or computational overhead (Table 1). This highlights its efficiency and practicality, particularly in resource-constrained settings. We begin with the pseudocode for $Bi^2Clip$ in Algorithm 4.

---

**Algorithm 4** $Bi^2Clip$

---

**Require:** Initial model $x_1$, learning rate schedule $\eta_t$, clipping schedules $u_t, d_t, \widetilde{u}_t, \widetilde{d}_t$
    Synchronization timestep $z \in \mathbb{Z}_{>0}$
1: **for** $t = 1, \ldots, T$ **do**
2:    **for** each node $i \in [N]$ in parallel **do**
3:        $x_{i,0}^t \leftarrow x_t$
4:        **for** each local step $k \in [z]$ **do**
5:            Draw minibatch gradient $g_{i,k}^t = \nabla F_i(x_{i,k}^t, \xi_{i,k}^t)$
6:            $x_{i,k+1}^t \leftarrow x_{i,k}^t - \eta_t \cdot BiClip(u_t, d_t, g_{i,k}^t)$
7:        **end for**
8:    **end for**
9:    $\Delta_t = \frac{1}{N} \sum_{i\in[N]} \left(x_{i,z}^t - x_{t-1}\right), \quad \widetilde{m}_t \leftarrow \Delta_t$
10:    $x_t = x_{t-1} + \eta BiClip(\widetilde{u}_t, \widetilde{d}_t, \widetilde{m}_t)$
11: **end for**

---

We carry out the analysis over a sufficiently large, compact domain $\mathcal{X}$. Let $\nabla F(x)$ be the deterministic gradient, obtained by integrating over $\nabla F(x, \xi)$, the stochastic gradient with a heavy-tailed distribution. The existence of $\nabla F(x)$ implies $F(x)$ is continuous, which gives boundedness via the extremal value theorem. Therefore, from now onward, we formally assume $\nabla F(x)$ is coordinatewise bounded by $G$ in absolute value. We have the following theorem.

**Theorem 5.** *Let Assumptions 1-2 hold, and the learning rate and clipping schedules satisfy $\eta_t = \Theta(t^\omega)$, $\eta_\ell^t = \Theta(t^\nu)$, $d_t = \Theta(t^\gamma)$, $u_t = \Theta(t^\zeta)$, $\widetilde{d}_t = \Theta(t^{\widetilde{\gamma}})$, and $u_t = \Theta(t^{\widetilde{\zeta}})$. Imposing $\zeta, \widetilde{\zeta} > 0 > \gamma, \widetilde{\gamma}$, for $\omega, \nu \leq 0$, as well as $-1 < \omega + \nu$, for $Bi^2Clip$ (Algorithm 4), we have that*

$$\min_{t\in[T]} \mathbb{E}[\|\nabla F(x_{t-1})\|^2] \lesssim \Psi_1 + \Psi_2 + \Psi_3 + \Psi_4 + \Psi_5 + \Psi_6 + \Psi_7,$$

*where the $\Psi_i$ are given*

$$\Psi_1 = \mathcal{O}\left(T^{-\omega-\nu-1}\right), \quad \Psi_2 = \mathcal{O}\left(T^{\omega+2\widetilde{\zeta}-\nu}\right), \quad \Psi_3 = \mathcal{O}\left(T^{\widetilde{\gamma}-\nu}\right), \quad \Psi_4 = \mathcal{O}\left(T^\gamma\right),$$

$$\Psi_5 = \mathcal{O}\left(T^{(\alpha-1)\nu+(1-\alpha)\widetilde{\zeta}}\right), \quad \Psi_6 = \mathcal{O}\left(T^{(1-\alpha)\zeta}\right), \quad \Psi_7 = \mathcal{O}\left(T^{\nu+\zeta}\right).$$

*Proof.* We provide the proof for $L_2$-wise $BiClip(\cdot)$ for illustrative purposes and notational convenience. The extension to coordinate-wise $BiClip(\cdot)$ is straightforward as described in the comments following the proof of Theorem 6, Remark 2. For completeness and readability, we formally provide the definition of $L_2$-wise $BiClip(\cdot)$ as

$$BiClip(u_t, d_t, x) = x \cdot \frac{d_t}{\|x\|} \chi\left(\|x\| \leq d_t\right) + x \cdot \frac{u_t}{\|x\|} \chi\left(\|x\| \geq u_t\right) + x \cdot \chi\left(d_t < \|x\| < u_t\right).$$

Here, $\chi$ is the indicator function, and $u_t \geq d_t \geq 0$ are the clipping thresholds. By default, we take $a/0 := 0$ for $\forall a \in \mathbb{R}$. Now, we begin by noting that due to $L$-smoothness, we have where $\mathbb{E}_t[\,\cdot\,]$ takes expectation up to $x_{t-1}$ that

$$
\mathbb{E}_t[F(x_t)] - F(x_{t-1}) \leq \langle \nabla F(x_{t-1}), \mathbb{E}_t[x_t - x_{t-1}] \rangle + \frac{L}{2}\mathbb{E}_t[\|x_t - x_{t-1}\|^2]
$$

$$
\leq \eta_t \underbrace{\left\langle \nabla F(x_{t-1}), -\mathbb{E}_t[BiClip(\widetilde{u}_t, \widetilde{d}_t, -\Delta_t)] \right\rangle}_{A_1} + \frac{L\eta_t^2}{2}\mathbb{E}_t\left[\left\|BiClip(\widetilde{u}_t, \widetilde{d}_t, \Delta_t)\right\|^2\right].
$$

Now, we expand to obtain the following form

$$
A_1 = -\left\langle \nabla F(x_{t-1}), \mathbb{E}_t[BiClip(\widetilde{u}_t, \widetilde{d}_t, -\Delta_t) \pm \Delta_t] \mp \eta_\ell^t \sum_{i\in[N]}\sum_{\nu\in[K]-1} p_i\mathbb{E}_t[\nabla F_i(x_{i,\nu}^t)] \mp K\eta_\ell^t \nabla F(x_{t-1}) \right\rangle
$$

$$
= -\underbrace{\left\langle \nabla F(x_{t-1}), \mathbb{E}_t[BiClip(\widetilde{u}_t, \widetilde{d}_t, -\Delta_t) + \Delta_t] \right\rangle}_{B_1} - \underbrace{\left\langle \nabla F(x_{t-1}), -\eta_\ell^t \sum_{i\in[N]}\sum_{\nu\in[K]-1} p_i\mathbb{E}_t[\nabla F_i(x_{i,\nu}^t)] - \mathbb{E}_t[\Delta_t] \right\rangle}_{B_2}
$$

$$
-\underbrace{\left\langle \nabla F(x_{t-1}), \eta_\ell^t \sum_{i\in[N]}\sum_{\nu\in[K]-1} p_i\mathbb{E}_t[\nabla F_i(x_{i,\nu}^t)] - K\eta_\ell^t \nabla F(x_{t-1}) \right\rangle}_{B_3} - K\eta_\ell^t\|\nabla F(x_{t-1})\|^2.
$$

Using the convexity of compositions (via $\alpha \geq 1$) and Jensen, we deduce

$$
\mathbb{E}_t[\|\Delta_t\|^\alpha] = \mathbb{E}_t[\|\eta_\ell^t \sum_{i\in[N]}\sum_{\nu\in[K]-1} p_i \cdot BiClip(u_t, d_t, \nabla F_i(x_{i,\nu}^t, \xi_{i,\nu}^t))\|^\alpha]
$$

$$
\leq (\eta_\ell^t)^\alpha K^\alpha \mathbb{E}_t\left[\left\|\frac{1}{K}\cdot\sum_{i\in[N]}\sum_{\nu\in[K]-1} p_i \cdot BiClip(u_t, d_t, \nabla F_i(x_{i,\nu}^t, \xi_{i,\nu}^t))\right\|^\alpha\right]
$$

$$
\leq (\eta_\ell^t)^\alpha K^{\alpha-1} \sum_{i\in[N]}\sum_{\nu\in[K]-1} p_i\mathbb{E}_t[\|BiClip(u_t, d_t, \nabla F_i(x_{i,\nu}^t, \xi_{i,\nu}^t))\|^\alpha]
$$

$$
\leq (\eta_\ell^t)^\alpha K^{\alpha-1} \sum_{i\in[N]}\sum_{\nu\in[K]-1} p_i(d_t^\alpha + \mathbb{E}_t[\|\nabla F_i(x_{i,\nu}^t, \xi_{i,\nu}^t)\|^\alpha])
$$

$$
\leq (\eta_\ell^t)^\alpha K^{\alpha-1} \sum_{i\in[N]}\sum_{\nu\in[K]-1} p_i d_t^\alpha + \underbrace{(\eta_\ell^t)^\alpha K^{\alpha-1} \sum_{i\in[N]}\sum_{\nu\in[K]-1} p_i\mathbb{E}_t[\|\nabla F_i(x_{i,\nu}^t, \xi_{i,\nu}^t)\|^\alpha]}_{C}.
$$

Note that the term $C$ can be bounded as

$$
C \leq (\eta_\ell^t)^\alpha K^{\alpha-1} \sum_{i\in[N]}\sum_{\nu\in[K]-1} p_i 2^\alpha \mathbb{E}_t\left[\frac{\|\nabla F_i(x_{i,v}^t)\|^\alpha}{2} + \frac{\|\xi_{i,v}^t\|^\alpha}{2}\right]
$$

$$
\leq (\eta_\ell^t)^\alpha K^{\alpha-1} \sum_{i\in[N]}\sum_{\nu\in[K]-1} p_i 2^{\alpha-1}(M^\alpha + B^\alpha) = (\eta_\ell^t)^\alpha K^{\alpha-1} \sum_{\nu\in[K]-1} 2^{\alpha-1}(M^\alpha + B^\alpha),
$$

where $M := \max_{x\in\mathcal{X}, i\in[N]} \|\nabla F_i(x)\|$ and $B^\alpha := \max_{i\in[N],\, \nu\in[K]-1} \mathbb{E}_t[\|\xi_{i,v}^t\|^\alpha] \leq \sup_{i\in[N]}(B_i)^{\alpha_i}$. We note that this results holds also under distribution shift for the stochastic noise $\xi_i^t$, where $t \in [T]$ and $i \in [N]$, as long as the $\alpha$-moment remains universally bounded. Therefore, we conclude

$$
\mathbb{E}_t[\|\Delta_t\|^\alpha] \leq (\eta_\ell^t)^\alpha K^{\alpha-1} \sum_{\nu\in[K]-1} d_t^\alpha + (\eta_\ell^t)^\alpha K^{\alpha-1} 2^{\alpha-1} \sum_{\nu\in[K]-1} (M^\alpha + B^\alpha) =: (\eta_\ell^t)^\alpha \widetilde{M}.
$$

This gives by the Cauchy-Schwartz inequality that

$$\begin{aligned}
B_1 &\leq \|\nabla F(x_{t-1})\| \|\mathbb{E}_t[BiClip(\widetilde{u}_t, \widetilde{d}_t, -\Delta_t)] + \Delta_t\| \\
&\leq G \cdot \mathbb{E}_t[\chi(\|\Delta_t\| \leq \widetilde{d}_t)\, \widetilde{d}_t + \chi(\widetilde{u}_t \leq \|\Delta_t\|)\, \|\Delta_t\|^{\alpha} \|\Delta_t\|^{1-\alpha}] \\
&\leq G\left[\mathbb{P}(\|\Delta_t\| \leq \widetilde{d}_t)\, \widetilde{d}_t + \mathbb{P}(\widetilde{u}_t \leq \|\Delta_t\|)\, (\eta_\ell^t)^{\alpha} \widetilde{u}_t^{1-\alpha} \widetilde{M}\right].
\end{aligned}$$

Now, $B_2$ may be bounded as follows:

$$\begin{aligned}
B_2 &\leq G\left\|\eta_\ell^t \sum_{i \in [N]} \sum_{\nu \in [K]-1} p_i \mathbb{E}_t[\nabla F_i(x_{i,\nu}^t)] + \mathbb{E}_t[\Delta_t]\right\| \\
&= G\left\|\mathbb{E}_t[\eta_\ell^t \sum_{i \in [N]} \sum_{\nu \in [K]-1} p_i \nabla F_i(x_{i,\nu}^t, \xi_{i,\nu}^t) + \Delta_t]\right\| \\
&\leq G\mathbb{E}_t\left[\left\|\eta_\ell^t \sum_{i \in [N]} \sum_{\nu \in [K]-1} p_i \nabla F_i(x_{i,\nu}^t, \xi_{i,\nu}^t) + \Delta_t\right\|\right],
\end{aligned}$$

where we used convexity, Jensen, and that the stochastic gradient noise is unbiased. Unraveling the definition of the pseudogradient $\Delta_t$ gives

$$\begin{aligned}
B_2 &\leq G\eta_\ell^t \mathbb{E}_t\left[\left\|\sum_{i \in [N]} \sum_{\nu \in [K]-1} p_i \nabla F_i(x_{i,\nu}^t, \xi_{i,\nu}^t) - \sum_{i \in [N]} \sum_{\nu \in [K]-1} p_i BiClip(u_t, d_t, \nabla F_i(x_{i,\nu}^t, \xi_{i,\nu}^t))\right\|\right] \\
&\leq G\eta_\ell^t \sum_{i \in [N]} \sum_{\nu \in [K]-1} p_i \mathbb{E}_t\left[\|\nabla F_i(x_{i,\nu}^t, \xi_{i,\nu}^t) - BiClip(u_t, d_t, \nabla F_i(x_{i,\nu}^t, \xi_{i,\nu}^t))\|\right] \\
&\leq G\eta_\ell^t \sum_{i \in [N]} \sum_{\nu \in [K]-1} p_i\left[d_t \mathbb{P}(\|\nabla F_i(x_{i,\nu}^t, \xi_{i,\nu}^t)\| \leq d_t) + \mathbb{P}(\|\nabla F_i(x_{i,\nu}^t, \xi_{i,\nu}^t)\| \geq u_t)u_t^{1-\alpha}2^{\alpha-1}(M^{\alpha} + B^{\alpha})\right] \\
&\leq G\eta_\ell^t \sum_{\nu \in [K]-1}\left[d_t + u_t^{1-\alpha}2^{\alpha-1}(M^{\alpha} + B^{\alpha})\right].
\end{aligned}$$

Additionally, $B_3$ may be bounded via $L$-smoothness and telescoping:

$$\begin{aligned}
B_3 &\leq \eta_\ell^t G\left\|\sum_{i \in [N]} \sum_{\nu \in [K]-1} p_i \nabla F_i(x_{i,\nu}^t) - K\nabla F(x_{t-1})\right\| \\
&\leq \eta_\ell^t G\left\|\sum_{i \in [N]} \sum_{\nu \in [K]-1} p_i \nabla F_i(x_{i,\nu}^t) - \sum_{i \in [N]} \sum_{\nu \in [K]-1} p_i \nabla F_i(x_{i,0}^t)\right\| \\
&\leq \eta_\ell^t G \sum_{i \in [N]} \sum_{\nu \in [K]-1} p_i L\|x_{i,\nu}^t - x_{i,0}^t\| \\
&\leq \eta_\ell^t G \sum_{i \in [N]} \sum_{\nu \in [K]-1} p_i L\left\|x_{i,\nu}^t + \sum_{r=1}^{v-1}(x_{i,r}^t - x_{i,r}^t) - x_{i,0}^t\right\| \\
&\leq \eta_\ell^t GL \sum_{i \in [N]} p_i \cdot \left(\sum_{\nu \in [K]-1} \sum_{r=1}^{v-1} \|x_{i,r}^t - x_{i,r-1}^t\|\right) \leq \frac{(\eta_\ell^t)^2 GLK^2 u_t}{2}.
\end{aligned}$$

Collecting all inequalities gathered thus far, we have

$$\begin{aligned}
\mathbb{E}_t[F(x_t)] - F(x_{t-1}) &\leq \frac{L\eta_t^2 \widetilde{u}_t^2}{2} - K\eta_\ell^t \eta_t \|\nabla F(x_{t-1})\|^2 + G\eta_t \widetilde{d}_t + G\eta_t(\eta_\ell^t)^{\alpha} \widetilde{u}_t^{1-\alpha} \widetilde{M} \\
&\quad + G\eta_\ell^t \eta_t \sum_{\nu \in [K]-1}\left[d_t + u_t^{1-\alpha}2^{\alpha-1}(M^{\alpha} + B^{\alpha})\right] + \frac{\eta_t(\eta_\ell^t)^2 GLK^2 u_t}{2}.
\end{aligned}$$

Telescoping under the law of iterated expectations gives

$$\sum_{t\in[T]} K\eta_\ell^t \eta_t \mathbb{E}[\|\nabla F(x_{t-1})\|^2] \leq F(x_0) - \mathbb{E}[F(x_T)] + \sum_{t\in[T]} \left( \frac{L\eta_t^2 \widetilde{u}_t^2}{2} + G\eta_t \widetilde{d}_t + G\eta_t (\eta_\ell^t)^\alpha \widetilde{u}_t^{1-\alpha} \widetilde{M} \right)$$

$$+ G \sum_{t\in[T]} \eta_\ell^t \eta_t \sum_{\nu\in[K]-1} \left[ d_t + u_t^{1-\alpha} 2^{\alpha-1} (M^\alpha + B^\alpha) \right] + \sum_{t\in[T]} \frac{\eta_t (\eta_\ell^t)^2 GLK^2 u_t}{2}.$$

Now, we move to the asymptotic regime. Let $\eta_t = \Theta(t^\omega)$, $\eta_\ell^t = \Theta(t^\nu)$, $d_t = \Theta(t^\gamma)$, $u_t = \Theta(t^\zeta)$, $\widetilde{d}_t = \Theta(t^{\widetilde{\gamma}})$, and $\widetilde{u}_t = \Theta(t^{\widetilde{\zeta}})$. This gives after routine calculations that

$$\min_{t\in[T]} \mathbb{E}[\|\nabla F(x_{t-1})\|^2] \lesssim \mathcal{O}\left( T^{-\omega-\nu-1} + T^{\omega+2\widetilde{\zeta}-\nu} + T^{\widetilde{\gamma}-\nu} + T^{(\alpha-1)\nu+(1-\alpha)\widetilde{\zeta}} + T^\gamma + T^{(1-\alpha)\zeta} + T^{\nu+\zeta} \right).$$

To attain convergence of the RHS, it is clear that we must impose $\zeta, \widetilde{\zeta} > 0 > \gamma, \widetilde{\gamma}$, for $\omega, \nu \leq 0$. Additionally, we have further constrained $-1 < \omega + \nu$, which ensures that the LHS diverges at a scale faster than logarithmic, validating the asymptotic regime and concluding the proof. To obtain the rate of convergence, we may let

$$\begin{aligned} r_1 &= \omega + \nu + 1, & r_2 &= -\omega - 2\tilde{\zeta} + \nu, \\ r_3 &= \nu - \widetilde{\gamma}, & r_4 &= (\alpha-1)(\tilde{\zeta} - \nu), \\ r_5 &= -\gamma, & r_6 &= (\alpha-1)\zeta, \quad r_7 = -\nu - \zeta. \end{aligned} \tag{10}$$

for $\sigma = \min\{r_1, \ldots, r_7\}$ and

$$\min_{t\in[T]} \mathbb{E}\|\nabla F(x_{t-1})\|^2 = \mathcal{O}\left(T^{-\sigma}\right).$$

Equalizing $r_6$ and $r_7$ gives that $\zeta = -\nu/\alpha$ as $\alpha > 1, \nu < 0, \zeta > 0$. Similarly, equalizing $r_1$ and $r_2$ gives $\omega = -(1 + 2\tilde{\zeta})/2$, yielding $r_1 = r_2 = 1/2 + \nu - \tilde{\zeta}$. By letting $\widetilde{\zeta} \to 0^+$,

$$\frac{1}{2} + \nu - \tilde{\zeta} = (\alpha-1)\frac{-\nu}{\alpha} \xrightarrow{\tilde{\zeta}\to 0^+} \nu = -\frac{\alpha}{4\alpha-2}, \quad \zeta = \frac{1}{4\alpha-2}, \quad \omega = -\frac{1}{2}.$$

For suitably small $\gamma, \widetilde{\gamma}$, we therefore have

$$\sigma = \frac{\alpha-1}{4\alpha-2},$$

concluding the proof. $\qquad\square$

**Remark 1.** *We note that setting $\widetilde{d}_t = 0$, $\widetilde{u}_t = \infty$, and $\eta_t = 1$ at the outer optimizer recovers a special case of $Bi^2Clip$, i.e., Avg-BiClip. Similarly, for specific hyperparameter choices, $Bi^2Clip$ collapses into $BiClip$-SGD, with upper and lower thresholding applied by the outer optimizers only to accumulated model updates from the inner compute nodes.*

Now, in the following subsections, we further analyze the convergence behavior of TailOPT under additional varying adaptive optimizer instantiations. The Adagrad instantiation (Algorithm 5) collects pseudogradients and sums their squares, effectively implementing a form of implicit clipping. However, it aggressively decays coordinate-wise learning rates, which can limit performance. To address this, we introduce RMSProp-$TailClip$ (Algorithm 6), which relaxes the preconditioning by employing an exponentially decaying moving average of the second moment. In both cases, we prove that the minimum expected gradient converges to 0. Additionally, by incorporating a moving average of the first pseudogradient moment as a form of momentum, we derive Algorithm 7. For this variant, we show that the expected minimal gradient does not diverge even under restarting of the algorithm, which in practice translates to the update of any singular step not diverging in expectation. As in the main paper, $TailClip$ refers to either $BiClip$ or $L_2Clip$, and we provide our proofs for $BiClip$ for added generality over $L_2Clip$.

### C.4. Convergence of Adagrad-$TailClip$

We begin by providing the pseudocode of Adagrad-$TailClip$ (Algorithm 5). Then, we have the following result.

---

**Algorithm 5** Adagrad-$TailClip$

---

**Require:** Initial model $x_1$, learning rate schedule $\eta_t$, clipping schedules $u_t$, $d_t$

  Synchronization timestep $z \in \mathbb{Z}_{>0}$, adaptivity parameter $\tau > 0$

1: **for** $t = 1, \ldots, T$ **do**
2:   **for** each node $i \in [N]$ in parallel **do**
3:     $x_{i,0}^t \leftarrow x_t$
4:     **for** each local step $k \in [z]$ **do**
5:       Draw minibatch gradient $g_{i,k}^t = \nabla F_i(x_{i,k}^t, \xi_{i,k}^t)$
6:       $x_{i,k+1}^t \leftarrow x_{i,k}^t - \eta_t \cdot TailClip(u_t, d_t, g_{i,k}^t)$
7:     **end for**
8:   **end for**
9:   $\Delta_t = \frac{1}{N} \sum_{i \in [N]} \left( x_{i,z}^t - x_{t-1} \right), \quad \widetilde{m}_t \leftarrow \Delta_t$
10:   $\widetilde{v}_t = \widetilde{v}_{t-1} + \Delta_t^2$
11:   $x_t = x_{t-1} + \eta \frac{\widetilde{m}_t}{\sqrt{\widetilde{v}_t} + \tau}$
12: **end for**

---

**Theorem 6.** *Let the clipping and learning rate thresholds satisfy $\eta_t = \Theta(t^\omega)$, $\eta_\ell^t = \Theta(t^\nu)$, $d_t = \Theta(t^\gamma)$, and $u_t = \Theta(t^\zeta)$ for the conditions $\zeta > 0$, $\gamma < -1/2$, and $\omega - \zeta - 1/2 > 0$. Then, we have that*

$$\min_{t \in [T]} \mathbb{E} \left\| \nabla F(x_t) \right\|^2 \leq \Psi_1 + \Psi_2 + \Psi_3 + \Psi_4 + \Psi_5 + \Psi_6,$$

*where the $\Psi_i$ are upper bounded by*

$$\Psi_1 \leq \mathcal{O}(T^{-\omega+\zeta-\frac{1}{2}}), \quad \Psi_2 \leq \mathcal{O}(T^{\omega+2\nu+3\zeta+\frac{1}{2}}), \quad \Psi_3 \leq \mathcal{O}(T^{4\zeta+3\nu+\frac{1}{2}}),$$

$$\Psi_4 \leq \mathcal{O}(T^{2\nu+2\zeta+\frac{1}{2}}), \quad \Psi_5 \leq \mathcal{O}(T^{\nu+\gamma+\zeta+\frac{1}{2}}), \quad \Psi_6 \leq \mathcal{O}(T^{\nu+(2-\alpha)\zeta+\frac{1}{2}}),$$

*which guarantees convergence via an inversely proportional power law decay with respect to $T$. The maximal convergence rate is given by $\mathcal{O}(T^{(1-\alpha)/2\alpha})$.*

*Proof.* We analyze the convergence of the global objective, where model weights are updated in a distributed fashion via local $BiClip$ under heavy-tailed noise. By $L$-smoothness, we have

$$F(x_t) \leq F(x_{t-1}) + \langle \nabla F(x_{t-1}), x_t - x_{t-1} \rangle + \frac{L}{2} \left\| x_t - x_{t-1} \right\|^2$$

$$= F(x_{t-1}) + \eta_t \underbrace{\left\langle \nabla F(x_{t-1}), \frac{\Delta_t}{\sqrt{\widetilde{v}_t} + \tau} \right\rangle}_{A_1} + \frac{\eta_t^2 L}{2} \left\| \frac{\Delta_t}{\sqrt{\widetilde{v}_t} + \tau} \right\|^2,$$

which we further decompose via noting that

$$A_1 = \eta_t \left\langle \nabla F(x_{t-1}), \frac{\Delta_t(\sqrt{\widetilde{v}_{t-1}} - \sqrt{\widetilde{v}_t})}{(\sqrt{\widetilde{v}_t} + \tau)(\sqrt{\widetilde{v}_{t-1}} + \tau)} \right\rangle + \eta_t \left\langle \nabla F(x_{t-1}), \frac{\Delta_t}{\sqrt{\widetilde{v}_{t-1}} + \tau} \right\rangle$$

$$= \eta_t \left\langle \nabla F(x_{t-1}), \frac{-\Delta_t^3}{(\sqrt{\widetilde{v}_t} + \tau)(\sqrt{\widetilde{v}_{t-1}} + \tau)(\sqrt{\widetilde{v}_{t-1}} + \sqrt{\widetilde{v}_t})} \right\rangle + \eta_t \left\langle \nabla F(x_{t-1}), \frac{\Delta_t}{\sqrt{\widetilde{v}_{t-1}} + \tau} \right\rangle$$

$$\leq \eta_t \left\langle |\nabla F(x_{t-1})|, \frac{|\Delta_t|^3}{(\sqrt{\widetilde{v}_t} + \tau)(\sqrt{\widetilde{v}_{t-1}} + \tau)(\sqrt{\widetilde{v}_{t-1}} + \sqrt{\widetilde{v}_t})} \right\rangle + \eta_t \underbrace{\left\langle \nabla F(x_{t-1}), \frac{\Delta_t}{\sqrt{\widetilde{v}_{t-1}} + \tau} \right\rangle}_{B_1}.$$

To bound $B_1$, we extract a negative gradient norm

$$B_1 = \eta_t \underbrace{\left\langle \nabla F(x_{t-1}), \frac{\Delta_t}{\sqrt{\widetilde{v}_{t-1}} + \tau} + \frac{K\eta_\ell^t \nabla F(x_{t-1})}{\sqrt{\widetilde{v}_{t-1}} + \tau} \right\rangle}_{B_2} - K\eta_t\eta_\ell^t \left\| \frac{\nabla F(x_{t-1})}{\sqrt{\sqrt{\widetilde{v}_{t-1}} + \tau}} \right\|^2,$$

where $B_2$ decomposes further into

$$B_2 = \eta_t \left\langle \nabla F(x_{t-1}), \frac{\Delta_t}{\sqrt{\widetilde{v}_{t-1}} + \tau} + \frac{\sum_{i \in [N]} \sum_{v \in [K]-1} p_i \eta_\ell^t (\nabla F_i(x_{i,v}^t) - \nabla F_i(x_{i,v}^t))}{\sqrt{\widetilde{v}_{t-1}} + \tau} + \frac{K\eta_\ell^t \nabla F(x_{t-1})}{\sqrt{\widetilde{v}_{t-1}} + \tau} \right\rangle$$

Here, we use the convention $[K] - 1 = \{0, \ldots, K-1\}$, and that summation over null indices are zero (e.g. $\sum_{j=K}^{K-1} [\,\cdot\,] = 0$). Now, recall

$$\Delta_t := \sum_{i \in [N]} p_i \Delta_i^t = \sum_{i \in [N]} p_i (x_{i,K}^t - x_{i,0}^t) = -\sum_{i \in [N]} \sum_{v \in [K]-1} p_i \eta_\ell^t \cdot \hat{g}_{i,v}^t$$

$$= -\sum_{i \in [N]} \sum_{v \in [K]-1} p_i \eta_\ell^t \cdot BiClip(u_t, d_t, \nabla F_i(x_{i,v}^t) + \xi_{i,v}^t),$$

which implies $B_2 = C_1 + C_2$ for

$$C_1 = \eta_t \left\langle \nabla F(x_{t-1}), \frac{\sum_{i \in [N]} \sum_{v \in [K]-1} p_i \eta_\ell^t (\nabla F_i(x_{i,v}^t) - BiClip(u_t, d_t, \nabla F_i(x_{i,v}^t) + \xi_{i,v}^t))}{\sqrt{\widetilde{v}_{t-1}} + \tau} \right\rangle$$

$$C_2 = \eta_t \left\langle \nabla F(x_{t-1}), \frac{\sum_{i \in [N]} \sum_{v \in [K]-1} p_i \eta_\ell^t (\nabla F_i(x_{i,0}^t) - \nabla F_i(x_{i,v}^t))}{\sqrt{\widetilde{v}_{t-1}} + \tau} \right\rangle.$$

Letting $\mathbb{E}_t[\,\cdot\,]$ condition over all stochasticity up to global step $t$, we have that $\mathbb{E}_t[C_1]$ is equal to

$$\eta_t \left\langle \nabla F(x_{t-1}), \frac{\sum_{i \in [N]} \sum_{v \in [K]-1} p_i \eta_\ell^t (\mathbb{E}_t[\nabla F_i(x_{i,v}^t) + \xi_{i,v}^t - BiClip(u_t, d_t, \nabla F_i(x_{i,v}^t) + \xi_{i,v}^t)])}{\sqrt{\widetilde{v}_{t-1}} + \tau} \right\rangle.$$

For $D_1 := \mathbb{E}_t[\nabla F_i(x_{i,v}^t) + \xi_{i,v}^t - BiClip(u_t, d_t, \nabla F_i(x_{i,v}^t) + \xi_{i,v}^t)]$, we have by convexity and Jensen that

$$\|D_1\| \leq \mathbb{E}_t[\|\nabla F_i(x_{i,v}^t) + \xi_{i,v}^t - BiClip(u_t, d_t, \nabla F_i(x_{i,v}^t) + \xi_{i,v}^t)\|]$$

$$\leq d_t \mathbb{P}(\|\nabla F_i(x_{i,v}^t) + \xi_{i,v}^t\| \leq d_t)$$

$$+ \underbrace{\mathbb{E}_t[\|\nabla F_i(x_{i,v}^t) + \xi_{i,v}^t - BiClip(u_t, d_t, \nabla F_i(x_{i,v}^t) + \xi_{i,v}^t)\|\chi\left(\|\nabla F_i(x_{i,v}^t) + \xi_{i,v}^t\| \geq u_t\right)]}_{D_2}.$$

Piecewise continuity of $F_i(x)$ is clear via the existence of $\nabla F_i(x)$. This gives that

$$\mathbb{E}_t[\|\nabla F_i(x_{i,v}^t) + \xi_{i,v}^t\|^\alpha \chi\left(\|\nabla F_i(x_{i,v}^t) + \xi_{i,v}^t)\| \geq u_t\right)] \leq \mathbb{E}_t[\|\nabla F_i(x_{i,v}^t) + \xi_{i,v}^t\|^\alpha]$$

$$\leq 2^\alpha \mathbb{E}_t\left[\left\|\frac{\nabla F_i(x_{i,v}^t) + \xi_{i,v}^t}{2}\right\|^\alpha\right] \leq 2^\alpha \mathbb{E}_t\left[\frac{\|\nabla F_i(x_{i,v}^t)\|^\alpha}{2} + \frac{\|\xi_{i,v}^t\|^\alpha}{2}\right] = 2^{\alpha-1}(M^\alpha + B^\alpha),$$

where now, $M := \max_{x \in \mathcal{X}, i \in [N]} \|\nabla F_i(x)\|$. Thus, we may bound $D_2$ via reduction to the $\alpha$-moment:

$$D_2 \leq 2^{\alpha-1}(M^\alpha + B^\alpha)\mathbb{E}_t[\|\nabla F_i(x_{i,v}^t) + \xi_{i,v}^t\|^{1-\alpha}\chi\left(\|\nabla F_i(x_{i,v}^t) + \xi_{i,v}^t)\| \geq u_t\right)]$$

$$\leq 2^{\alpha-1}(M^\alpha + B^\alpha)u_t^{1-\alpha}\mathbb{P}\left(\|\nabla F_i(x_{i,v}^t; \xi_{i,v}^t))\| \geq u_t\right).$$

Collecting inequalities gives

$$\|D_1\| \leq d_t \mathbb{P}(\|\nabla F_i(x_{i,v}^t; \xi_{i,v}^t))\| \leq d_t) + 2^{\alpha-1}(M^\alpha + B^\alpha)u_t^{1-\alpha}\mathbb{P}\left(\|\nabla F_i(x_{i,v}^t; \xi_{i,v}^t))\| \geq u_t\right).$$

Therefore,

$$\mathbb{E}_t[C_1] \leq \frac{\eta_t G d}{\tau} \sum_{i \in [N]} \sum_{v \in [K]-1} p_i \eta_\ell^t d_t \mathbb{P}(\|\nabla F_i(x_{i,v}^t; \xi_{i,v}^t))\| \leq d_t)$$

$$+ \frac{2^{\alpha-1}\eta_t G d}{\tau} \sum_{i \in [N]} \sum_{v \in [K]-1} p_i \eta_\ell^t (M^\alpha + B^\alpha)u_t^{1-\alpha}\mathbb{P}\left(\|\nabla F_i(x_{i,v}^t; \xi_{i,v}^t))\| \geq u_t\right).$$

To bound $C_2$, we note that via $L$-smoothness, we have

$$C_2 \leq \frac{\eta_t GLd}{\tau} \sum_{i \in [N]} \sum_{v \in [K]-1} p_i \eta_\ell^t \|x_{i,0}^t - x_{i,v}^t\|$$

$$\leq \frac{\eta_t GLd}{\tau} \sum_{i \in [N]} \sum_{v \in [K]-1} p_i \eta_\ell^t \|x_{i,0}^t + \sum_{r=1}^{v-1}(x_{i,r}^t - x_{i,r}^t) - x_{i,v}^t\|$$

$$\leq \frac{\eta_t GLd}{\tau} \sum_{i \in [N]} \sum_{v \in [K]-1} \sum_{r \in [v]} p_i \eta_\ell^t \|x_{i,r}^t - x_{i,r-1}^t\| \leq \frac{\eta_t GLK^2d}{2\tau}(\eta_\ell^t)^2 u_t.$$

Noting that $\|\Delta_t\| \leq \eta_\ell^t u_t K$, we thus obtain

$$\mathbb{E}_t[F(x_t)] \leq F(x_{t-1}) + \frac{\eta_t^2(\eta_\ell^t)^2 u_t^2 K^2 L}{2\tau^2} + \frac{\eta_t GdK^3 u_t^3(\eta_\ell^t)^3}{\tau^3} - K\eta_t\eta_\ell^t \left\| \frac{\nabla F(x_{t-1})}{\sqrt{\sqrt{\widetilde{v}_{t-1}} + \tau}} \right\|^2$$

$$+ \frac{\eta_t GLK^2d}{2\tau}(\eta_\ell^t)^2 u_t + \frac{\eta_t Gd}{\tau} \sum_{i \in [N]} \sum_{v \in [K]-1} p_i \eta_\ell^t d_t \mathbb{P}(\|\nabla F_i(x_{i,v}^t; \xi_{i,v}^t)\| \leq d_t)$$

$$+ \frac{2^{\alpha-1} \eta_t Gd}{\tau} \sum_{i \in [N]} \sum_{v \in [K]-1} p_i \eta_\ell^t (M^\alpha + B^\alpha) u_t^{1-\alpha} \mathbb{P}\left(\|\nabla F_i(x_{i,v}^t; \xi_{i,v}^t)\| \geq u_t\right).$$

Taking expectations on both sides and telescoping gives via the tower law of expectation,

$$\underbrace{\sum_{t \in [T]} K\eta_t\eta_\ell^t \mathbb{E}\left[\left\| \frac{\nabla F(x_{t-1})}{\sqrt{\sqrt{\widetilde{v}_{t-1}} + \tau}} \right\|^2\right]}_{E_1} \leq \underbrace{\mathbb{E}[F(x_T) - F(x_0)]}_{E_2} + \underbrace{\sum_{t \in [T]} \frac{\eta_t^2(\eta_\ell^t)^2 u_t^2 K^2 L}{2\tau^2}}_{E_3} + \underbrace{\sum_{t \in [T]} \frac{\eta_t GdK^3 u_t^3(\eta_\ell^t)^3}{\tau^3}}_{E_4}$$

$$+ \underbrace{\sum_{t \in [T]} \frac{\eta_t GLK^2d}{2\tau}(\eta_\ell^t)^2 u_t}_{E_5} + \underbrace{\sum_{t \in [T]} \frac{\eta_t Gd}{\tau} \sum_{i \in [N]} \sum_{v \in [K]-1} p_i \eta_\ell^t d_t \mathbb{P}(\|\nabla F_i(x_{i,v}^t; \xi_{i,v}^t)\| \leq d_t)}_{E_6}$$

$$+ \underbrace{\sum_{t \in [T]} \frac{2^{\alpha-1} \eta_t Gd}{\tau} \sum_{i \in [N]} \sum_{v \in [K]-1} p_i \eta_\ell^t (M^\alpha + B^\alpha) u_t^{1-\alpha} \mathbb{P}\left(\|\nabla F_i(x_{i,v}^t; \xi_{i,v}^t)\| \geq u_t\right)}_{E_7},$$

where we have enumerated each term from $E_1$ to $E_7$ for clarity. To simplify notation, we now move to the asymptotic regime. Letting $\eta_t = \Theta(t^\omega)$, $\eta_\ell^t = \Theta(t^\nu)$, $d_t = \Theta(t^\gamma)$, and $u_t = \Theta(t^\zeta)$, we have via standard integral bounds that

$$E_1 \geq \Omega\left(T^{\omega+\nu+1} \cdot T^{-\zeta-\nu-\frac{1}{2}} \cdot \min_{t \in [T]} \mathbb{E}[\|\nabla F(x_t)\|^2]\right) = \Omega\left(T^{\omega-\zeta+\frac{1}{2}} \cdot \min_{t \in [T]} \mathbb{E}[\|\nabla F(x_t)\|^2]\right),$$

$$E_2 \leq \max_{x \in \mathcal{X}} F(x) - \min_{y \in \mathcal{X}} F(y) = \mathcal{O}(1), \quad E_3 \leq \mathcal{O}(T^{2\omega+2\nu+2\zeta+1}), \quad E_4 \leq \mathcal{O}(T^{\omega+3\zeta+3\nu+1}),$$

$$E_5 \leq \mathcal{O}(T^{\omega+2\nu+\zeta+1}), \quad E_6 \leq \mathcal{O}(T^{\omega+\nu+\gamma+1}), \quad E_7 \leq \mathcal{O}(T^{\omega+\nu+(1-\alpha)\zeta+1})$$

where any $E_i$ residues of $\mathcal{O}(1)$ for $i \geq 2$ have been incorporated into the upper bound for $E_2$. We note that the bound may be sharpened as the probabilistic terms must necessarily decay if $d_t \to 0$, $u_t \to \infty$, which further diminishes $E_6$, $E_7$. Now, to attain convergence of the minimal gradient, we impose the conditions

$$\Lambda_1 : \zeta > 0 \quad \text{and} \quad \gamma < 0, \quad \Lambda_2 : \omega - \zeta + \frac{1}{2} > 0, \quad \Lambda_3 : \omega + 2\nu + 3\zeta + \frac{1}{2} < 0,$$

$$\Lambda_4 : 4\zeta + 3\nu + \frac{1}{2} < 0, \quad \Lambda_5 : 2\nu + 2\zeta + \frac{1}{2} < 0, \quad \Lambda_6 : \nu + \gamma + \zeta + \frac{1}{2} < 0,$$

$$\Lambda_7 : \nu + (2-\alpha)\zeta + \frac{1}{2} < 0.$$

We note that each condition $\Lambda_{i \geq 2}$ comes from $E_i / E_1 \to 0, T \to \infty$, as any residual terms are subsumed by $\mathcal{O}(1)$, which decays via $\Lambda_2$. To derive the convergence rate, we equalize the bottlenecks by letting $\Lambda_2 = \Lambda_3 = \Lambda_7 = \sigma$. This gives a system of equations:

$$\omega - \zeta + \frac{1}{2} = \sigma$$

$$-\left(\omega + 2\nu + 3\zeta + \frac{1}{2}\right) = \sigma$$

$$-\left(\nu + (2 - \alpha)\zeta + \frac{1}{2}\right) = \sigma$$

which gives that

$$\zeta = \frac{1}{2\alpha}, \quad \nu = -\frac{1}{\alpha} - \sigma, \quad \omega = \sigma - \frac{\alpha - 1}{2\alpha}. \tag{11}$$

Letting $\sigma = (\alpha - 1)/2\alpha$ gives the slackness of $\Lambda_4, \Lambda_5$, and $\Lambda_6$ for $\gamma < -1/2$. Therefore, the minimum expected gradient norm squared stabilizes at rate $\mathcal{O}(T^{-\sigma})$. $\qquad\square$

**Remark 2.** *In the case of coordinate-wise clipping, all major adjustments up to a scaling factor of $\sqrt{d}$ are made in the terms bounding $\mathbb{E}[C_1]$. In this case, the proof proceeds as follows.*

Defining $|\cdot|$ to act coordinatewise, $\mathbb{E}_t[C_1]$ is now less than or equal to

$$\eta_t \left\langle |\nabla F(x_{t-1})|, \frac{\sum_{i \in [N]} \sum_{v \in [K]-1} p_i \eta_\ell^t |\mathbb{E}_t[\nabla F_i(x_{i,v}^t) + \xi_{i,v}^t - BiClip(u_t, d_t, \nabla F_i(x_{i,v}^t) + \xi_{i,v}^t)]|}{\sqrt{\widetilde{v}_{t-1}} + \tau} \right\rangle.$$

Therefore by Jensen,

$$\mathbb{E}_t[C_1] \leq \frac{\eta_t \eta_\ell^t G}{\tau} \sum_{i \in [N]} \sum_{v \in [K]-1} \sum_{j \in [d]} p_i \mathbb{E}_t[\underbrace{|\nabla F_i(x_{i,v}^t) + \xi_{i,v}^t - BiClip(u_t, d_t, \nabla F_i(x_{i,v}^t) + \xi_{i,v}^t)|_j}_{D_{1,j}}].$$

We note that $\mathbb{E}_t[D_{1,j}]$ can be upper bounded by $D_{2,j} + D_{3,j}$ where

$$D_{2,j} = \mathbb{E}_t\left[D_{1,j} \cdot \chi\left(|\nabla F_i(x_{i,v}^t; \xi_{i,v}^t)|_j \leq d_t)\right)\right] \leq d_t \mathbb{P}\left(|\nabla F_i(x_{i,v}^t; \xi_{i,v}^t)|_j \leq d_t)\right)$$

$$D_{3,j} = \mathbb{E}_t\left[|\nabla F_i(x_{i,v}^t; \xi_{i,v}^t)|_j \chi\left(|\nabla F_i(x_{i,v}^t; \xi_{i,v}^t)|_j \geq u_t)\right)\right].$$

It follows that

$$D_{3,j} \leq \mathbb{E}_t\left[|\nabla F_i(x_{i,v}^t; \xi_{i,v}^t)|_j^\alpha |\nabla F_i(x_{i,v}^t; \xi_{i,v}^t)|_j^{1-\alpha} \chi\left(|\nabla F_i(x_{i,v}^t; \xi_{i,v}^t)|_j \geq u_t)\right)\right]$$

$$\leq 2^{\alpha-1}(M^\alpha + B^\alpha) u_t^{1-\alpha} \mathbb{P}\left(|\nabla F_i(x_{i,v}^t; \xi_{i,v}^t)|_j \geq u_t\right).$$

Note that we used coordinate-wise bounded alpha moments for some $\alpha \in (1,2)$, $\mathbb{E}[|\xi_i|_j^\alpha] \leq B_{i,j}^\alpha$. We therefore define the $M$ and $B$ to be

$$M := \max_{x \in \mathcal{X}, i \in [N], j \in [d]} |\nabla F_i(x)|_j \quad \text{and} \quad B = \max_{i \in [N], j \in [d]} B_{i,j}.$$

Comparing terms gives the identical asymptotic order of convergence to $L_2$ clipping in Theorem 6.

### C.5. Convergence of RMSProp-$TailClip$

For Algorithm 6, we have the following convergence bound.

**Theorem 7.** *For clipping and learning rate thresholds satisfying $\eta_t = \Theta(t^\omega)$, $\eta_\ell^t = \Theta(t^\nu)$, $d_t = \Theta(t^\gamma)$, and $u_t = \Theta(t^\zeta)$, let the conditions listed in Theorem 6 hold. Then, local $BiClip$ with outer optimizer RMSProp stabilizes the expected minimum gradient $\min_{t \in [T]} \mathbb{E}[\|\nabla F(x_t)\|^2] \to 0^+$ with maximal rate $\mathcal{O}(T^{(1-\alpha)/2\alpha})$. Here, the exponential moving average parameter of the second pseudogradient moment is fixed within the range $\widetilde{\beta}_2 \in [0, 1)$.*

---

**Algorithm 6** RMS-$TailClip$

---

**Require:** Initial model $x_1$, learning rate schedule $\eta_t$, clipping schedules $u_t$, $d_t$

    Synchronization timestep $z \in \mathbb{Z}_{>0}$, adaptivity/EMA parameters $\tau > 0$, $\widetilde{\beta}_2 \in [0, 1)$

  1: **for** $t = 1, \ldots, T$ **do**

  2:     **for** each node $i \in [N]$ in parallel **do**

  3:         $x_{i,0}^t \leftarrow x_t$

  4:         **for** each local step $k \in [z]$ **do**

  5:             Draw minibatch gradient $g_{i,k}^t = \nabla F_i(x_{i,k}^t, \xi_{i,k}^t)$

  6:             $x_{i,k+1}^t \leftarrow x_{i,k}^t - \eta_t \cdot TailClip(u_t, d_t, g_{i,k}^t)$

  7:         **end for**

  8:     **end for**

  9:     $\Delta_t = \frac{1}{N} \sum_{i \in [N]} \left( x_{i,z}^t - x_{t-1} \right), \quad \widetilde{m}_t \leftarrow \Delta_t$

10:     $\widetilde{v}_t = \widetilde{\beta}_2 \widetilde{v}_{t-1} + (1 - \widetilde{\beta}_2)\Delta_t^2$

11:     $x_t = x_{t-1} + \eta \frac{\widetilde{m}_t}{\sqrt{\widetilde{v}_t} + \tau}$

12: **end for**

---

*Proof.* The proof for outer optimizer RMSProp builds on the prior proof for $BiClip$ with outer optimizer Adagrad. We skip repeated details for clarity of exposition, and concisely present only the main steps and ideas central to the proof for readability. $L$-smoothness gives as before

$$F(x_t) \leq F(x_{t-1}) + \langle \nabla F(x_{t-1}), x_t - x_{t-1} \rangle + \frac{L}{2} \|x_t - x_{t-1}\|^2$$

$$= F(x_{t-1}) + \eta_t \left\langle \nabla F(x_{t-1}), \frac{\Delta_t}{\sqrt{\widetilde{v}_t} + \tau} \right\rangle + \frac{\eta_t^2 L}{2} \left\| \frac{\Delta_t}{\sqrt{\widetilde{v}_t} + \tau} \right\|^2. \tag{12}$$

We note the decomposition

$$\left\langle \nabla F(x_{t-1}), \frac{\Delta_t}{\sqrt{\widetilde{v}_t} + \tau} \right\rangle = \underbrace{\left\langle \nabla F(x_{t-1}), \frac{\Delta_t}{\sqrt{\widetilde{v}_t} + \tau} - \frac{\Delta_t}{\sqrt{\widetilde{\beta}_2 \widetilde{v}_{t-1}} + \tau} \right\rangle}_{B_1} + \underbrace{\left\langle \nabla F(x_{t-1}), \frac{\Delta_t}{\sqrt{\widetilde{\beta}_2 \widetilde{v}_{t-1}} + \tau} \right\rangle}_{B_2}.$$

To form an upper bound, we use that

$$B_2 = \underbrace{\left\langle \nabla F(x_{t-1}), \frac{\Delta_t}{\sqrt{\widetilde{\beta}_2 \widetilde{v}_{t-1}} + \tau} + \frac{K \eta_\ell^t \nabla F(x_{t-1})}{\sqrt{\widetilde{\beta}_2 \widetilde{v}_{t-1}} + \tau} \right\rangle}_{C_0} - K \eta_\ell^t \left\| \frac{\nabla F(x_{t-1})}{\sqrt{\sqrt{\widetilde{\beta}_2 \widetilde{v}_{t-1}} + \tau}} \right\|^2$$

where $C_0 = C_1 + C_2$ for

$$C_1 = \left\langle \nabla F(x_{t-1}), \frac{\sum_{i \in [N]} \sum_{v \in [K]-1} p_i \eta_\ell^t (\nabla F_i(x_{i,v}^t) - BiClip(u_t, d_t, \nabla F_i(x_{i,v}^t) + \xi_{i,v}^t))}{\sqrt{\widetilde{\beta}_2 \widetilde{v}_{t-1}} + \tau} \right\rangle$$

$$C_2 = \left\langle \nabla F(x_{t-1}), \frac{\sum_{i \in [N]} \sum_{v \in [K]-1} p_i \eta_\ell^t (\nabla F_i(x_{i,0}^t) - \nabla F_i(x_{i,v}^t))}{\sqrt{\widetilde{\beta}_2 \widetilde{v}_{t-1}} + \tau} \right\rangle.$$

By the tower law and conditioning on stochastic realizations up to $t - 1$, we have as before

$$\mathbb{E}[C_0] \leq \frac{Gd}{\tau} \sum_{i \in [N]} \sum_{v \in [K]-1} p_i \eta_\ell^t d_t \mathbb{P}(\|\nabla F_i(x_{i,v}^t; \xi_{i,v}^t)\| \leq d_t) + \frac{GLK^2 d}{2\tau} (\eta_\ell^t)^2 u_t$$

$$+ \frac{2^{\alpha-1} Gd}{\tau} \sum_{i \in [N]} \sum_{v \in [K]-1} p_i \eta_\ell^t (M^\alpha + B^\alpha) u_t^{1-\alpha} \mathbb{P}\left(\|\nabla F_i(x_{i,v}^t; \xi_{i,v}^t)\| \geq u_t\right)$$

$$\leq \frac{Gd}{\tau} K \eta_\ell^t d_t + \frac{GLK^2 d}{2\tau} (\eta_\ell^t)^2 u_t + \frac{2^{\alpha-1} Gd}{\tau} K \eta_\ell^t (M^\alpha + B^\alpha) u_t^{1-\alpha}.$$

To bound $B_1$, we have

$$B_1 = \left\langle \nabla F(x_{t-1}), \frac{\Delta_t}{\sqrt{\widetilde{v}_t} + \tau} - \frac{\Delta_t}{\sqrt{\widetilde{\beta}_2 \widetilde{v}_{t-1}} + \tau} \right\rangle = \left\langle \nabla F(x_{t-1}), \frac{(\widetilde{\beta}_2 - 1) \Delta_t^3}{\left(\sqrt{\widetilde{v}_t} + \tau\right) \left(\sqrt{\widetilde{\beta}_2 \widetilde{v}_{t-1}} + \tau\right) \left(\sqrt{\widetilde{v}_t} + \sqrt{\widetilde{\beta}_2 \widetilde{v}_{t-1}}\right)} \right\rangle.$$

We prepare the global inequality (12) for telescoping. It is straightforward to see that collecting inequalities gives

$$\mathbb{E}[F(x_t)] \leq \mathbb{E}[F(x_{t-1})] + \frac{\eta_t^2 LK^2 u_t^2 (\eta_\ell^t)^2}{2\tau^2} - K \eta_t \eta_\ell^t \left\| \frac{\nabla F(x_{t-1})}{\sqrt{\sqrt{\widetilde{\beta}_2 \widetilde{v}_{t-1}} + \tau}} \right\|^2$$

$$\frac{Gd}{\tau} K \eta_t \eta_\ell^t d_t + \frac{GLK^2 d}{2\tau} \eta_t (\eta_\ell^t)^2 u_t + \frac{2^{\alpha-1} Gd}{\tau} K \eta_t \eta_\ell^t (M^\alpha + B^\alpha) u_t^{1-\alpha} + \frac{dG(1 - \widetilde{\beta}_2)(u_t \eta_\ell^t)^3}{\tau^3}$$

Rearranging and telescoping gives

$$\sum_{t=1}^T K \eta_t \eta_\ell^t \mathbb{E} \left[ \left\| \frac{\nabla F(x_{t-1})}{\sqrt{\sqrt{\widetilde{\beta}_2 \widetilde{v}_{t-1}} + \tau}} \right\|^2 \right] \leq \mathbb{E}[F(x_0)] - \mathbb{E}[F(x_T)] + \sum_{t=1}^T \frac{\eta_t^2 LK^2 u_t^2 (\eta_\ell^t)^2}{2\tau^2}$$

$$+ \sum_{t=1}^T \left( \frac{Gd}{\tau} K \eta_t \eta_\ell^t d_t + \frac{GLK^2 d}{2\tau} \eta_t (\eta_\ell^t)^2 u_t + \frac{2^{\alpha-1} Gd}{\tau} K \eta_t \eta_\ell^t (M^\alpha + B^\alpha) u_t^{1-\alpha} + \frac{dG(1 - \widetilde{\beta}_2)(u_t \eta_\ell^t)^3}{\tau^3} \right)$$

By non-negativity of squared pseudogradients, we immediately obtain $\widetilde{\beta}_2 \widetilde{v}_{t-1} \leq \widetilde{v}_{t-1}$. Therefore up to constants, the convergence bound collapses to asymptotically equivalent bounds than that of Theorem 6, up to constant multiples from the exponentially decaying moving average of the second moment pseudogradient. The modification to coordinate-wise clipping instead of $L_2$ clipping follows analogous steps. □

Incorporating momentum into the first pseudogradient moment further complicates the analysis, and yields the results presented in Section C.6.

## C.6. Convergence of Adam-$TailClip$

By incorporating a moving average of the first pseudogradient moment as a form of momentum, we derive Algorithm 7. For this variant, we demonstrate that the expected minimal gradient does not diverge, even when the algorithm undergoes restarts. Practically, this ensures that the located gradient value update of any single step remains bounded in expectation. Investigating the conditions required to guarantee convergence to 0 under this framework presents a promising avenue for future research. Our bound highlights that the dominating terms are influenced by the upper clipping threshold $u_r$, suggesting that the algorithm's convergence behavior may be closely related the choice of this threshold and can be tuned in practice.

**Theorem 8.** *Let the exponentially decaying moving average parameters satisfy $\widetilde{\beta}_1 \in (0, 1)$, $\widetilde{\beta}_2 \in [0, 1)$ for the outer optimizer first and second order pseudogradient moments, respectively. Extremize the unbiased stochastic noise such*

---

**Algorithm 7** Adam-$TailClip$

---

**Require:** Initial model $x_1$, learning rate schedule $\eta_t$, clipping schedules $u_t, d_t$
Synchronization timestep $z \in \mathbb{Z}_{>0}$, adaptivity/EMA parameters $\tau > 0$, $\widetilde{\beta}_1, \widetilde{\beta}_2 \in [0, 1)$
1: **for** $t = 1, \ldots, T$ **do**
2:    **for** each node $i \in [N]$ in parallel **do**
3:       $x_{i,0}^t \leftarrow x_t$
4:       **for** each local step $k \in [z]$ **do**
5:          Draw minibatch gradient $g_{i,k}^t = \nabla F_i(x_{i,k}^t, \xi_{i,k}^t)$
6:          $x_{i,k+1}^t \leftarrow x_{i,k}^t - \eta_t \cdot TailClip(u_t, d_t, g_{i,k}^t)$
7:       **end for**
8:    **end for**
9:    $\Delta_t = \frac{1}{N} \sum_{i \in [N]} \left( x_{i,z}^t - x_{t-1} \right)$
10:   $\widetilde{m}_t = \widetilde{\beta}_1 \widetilde{m}_{t-1} + (1 - \widetilde{\beta}_1)\Delta_t$
11:   $\widetilde{v}_t = \widetilde{\beta}_2 \widetilde{v}_{t-1} + (1 - \widetilde{\beta}_2)\Delta_t^2$
12:   $x_t = x_{t-1} + \eta \frac{\widetilde{m}_t}{\sqrt{\widetilde{v}_t} + \tau}$
13: **end for**

---

*that $\nexists \alpha_k \in (1, 2)$ for which $\mathbb{E}[\|\xi_k\|^{\alpha_k}] < B_k^{\alpha_k}$ for integrable $\xi_k$. Then, Algorithm 7 gives under constant upper clipping threshold invariant to global timestep $t$ ($\zeta = 0$) that*

$$\min_{t \in [T]} \mathbb{E}[\|\nabla F(x_t)\|^2] \lesssim \mathcal{O}(1),$$

*where for $\eta_t = \Theta(t^\omega)$, $\eta_\ell^t = \Theta(t^\nu)$, and $d_t = \Theta(t^\gamma)$, we impose*

$$\nu \in (-1, 0), \quad -\nu - 1 < \omega \le 0, \quad -(1 + \nu + \omega) < \gamma < 0. \tag{13}$$

*Proof.* As in the case of outer optimizer Adagrad, we analyze the convergence of the global objective. By $L$-smoothness, we have

$$F(x_t) \le F(x_{t-1}) + \langle \nabla F(x_{t-1}), x_t - x_{t-1} \rangle + \frac{L}{2} \|x_t - x_{t-1}\|^2$$

$$= F(x_{t-1}) + \eta_t \left\langle \nabla F(x_{t-1}), \underbrace{\frac{\widetilde{\beta}_1^t \widetilde{m}_0 + (1 - \widetilde{\beta}_1) \sum_{r=1}^t \widetilde{\beta}_1^{t-r} \Delta_r}{\sqrt{\widetilde{v}_t} + \tau}}_{A_1} \right\rangle + \frac{\eta_t^2 L}{2} \|A_1\|^2. \tag{14}$$

To proceed with the proof, we note that

$$\langle \nabla F(x_{t-1}), A_1 \rangle = \left\langle \nabla F(x_{t-1}), \frac{\widetilde{\beta}_1^t \widetilde{m}_0}{\sqrt{\widetilde{v}_t} + \tau} \right\rangle + (1 - \widetilde{\beta}_1) \sum_{r=1}^t \widetilde{\beta}_1^{t-r} \left\langle \nabla F(x_{t-1}), \frac{\Delta_r}{\sqrt{\widetilde{v}_t} + \tau} \right\rangle,$$

which we further decompose by using

$$\left\langle \nabla F(x_{t-1}), \frac{\Delta_r}{\sqrt{\widetilde{v}_t} + \tau} \right\rangle = \sum_{q=0}^{t-r} \underbrace{\left\langle \nabla F(x_{t-1}), \frac{\Delta_r}{\sqrt{\widetilde{\beta}_2^q \widetilde{v}_{t-q}} + \tau} - \frac{\Delta_r}{\sqrt{\widetilde{\beta}_2^{q+1} \widetilde{v}_{t-q-1}} + \tau} \right\rangle}_{A_{1,q}}$$

$$+ \underbrace{\left\langle \nabla F(x_{t-1}) - \nabla F(x_{r-1}), \frac{\Delta_r}{\sqrt{\widetilde{\beta}_2^{t-r+1} \widetilde{v}_{r-1}} + \tau} \right\rangle}_{B_1} + \underbrace{\left\langle \nabla F(x_{r-1}), \frac{\Delta_r}{\sqrt{\widetilde{\beta}_2^{t-r+1} \widetilde{v}_{r-1}} + \tau} \right\rangle}_{B_2}.$$

We have that

$$A_{1,q} = \sum_{q=0}^{t-r} \left\langle \nabla F(x_{t-1}), \frac{\Delta_r \left( \sqrt{\widetilde{\beta}_2^{q+1}\widetilde{v}_{t-q-1}} - \sqrt{\widetilde{\beta}_2^q \widetilde{v}_{t-q}} \right)}{\left( \sqrt{\widetilde{\beta}_2^q \widetilde{v}_{t-q}} + \tau \right) \left( \sqrt{\widetilde{\beta}_2^{q+1}\widetilde{v}_{t-q-1}} + \tau \right)} \right\rangle = \sum_{q=0}^{t-r} B_{1,q}$$

$$:= \sum_{q=0}^{t-r} \left\langle \nabla F(x_{t-1}), \frac{-(1-\widetilde{\beta}_2)\widetilde{\beta}_2^q \Delta_{t-q}^2 \Delta_r}{\left( \sqrt{\widetilde{\beta}_2^q \widetilde{v}_{t-q}} + \tau \right) \left( \sqrt{\widetilde{\beta}_2^{q+1}\widetilde{v}_{t-q-1}} + \tau \right) \left( \sqrt{\widetilde{\beta}_2^{q+1}\widetilde{v}_{t-q-1}} + \sqrt{\widetilde{\beta}_2^q \widetilde{v}_{t-q}} \right)} \right\rangle.$$

To upper bound $B_2$, we observe

$$B_2 = \underbrace{\left\langle \nabla F(x_{r-1}), \frac{\Delta_r}{\sqrt{\widetilde{\beta}_2^{t-r+1}\widetilde{v}_{r-1}} + \tau} + \frac{K\eta_\ell^r \nabla F(x_{r-1})}{\sqrt{\widetilde{\beta}_2^{t-r+1}\widetilde{v}_{r-1}} + \tau} \right\rangle}_{C_{0,r}} - K\eta_\ell^r \left\| \frac{\nabla F(x_{r-1})}{\sqrt{\sqrt{\widetilde{\beta}_2^{t-r+1}\widetilde{v}_{r-1}} + \tau}} \right\|^2$$

where $C_{0,r} = C_{1,r} + C_{2,r}$ for

$$C_{1,r} = \left\langle \nabla F(x_{r-1}), \frac{\sum_{i\in[N]} \sum_{v\in[K]-1} p_i \eta_\ell^r (\nabla F_i(x_{i,v}^r) - BiClip(u_r, d_r, \nabla F_i(x_{i,v}^r) + \xi_{i,v}^r))}{\sqrt{\widetilde{\beta}_2^{t-r+1}\widetilde{v}_{r-1}} + \tau} \right\rangle$$

$$C_{2,r} = \left\langle \nabla F(x_{r-1}), \frac{\sum_{i\in[N]} \sum_{v\in[K]-1} p_i \eta_\ell^r (\nabla F_i(x_{i,0}^r) - \nabla F_i(x_{i,v}^r))}{\sqrt{\widetilde{\beta}_2^{t-r+1}\widetilde{v}_{r-1}} + \tau} \right\rangle.$$

Noting that $\mathbb{E}[\,\cdot\,] = \mathbb{E}[\mathbb{E}_r[\,\cdot\,]]$ by the tower law, we have as before

$$\mathbb{E}[C_{0,r}] \leq \frac{Gd}{\tau} \sum_{i\in[N]} \sum_{v\in[K]-1} p_i \eta_\ell^r d_r \mathbb{P}(\|\nabla F_i(x_{i,v}^r; \xi_{i,v}^r))\| \leq d_r) + \frac{GLK^2 d}{2\tau}(\eta_\ell^r)^2 u_r$$

$$+ \frac{2^{\alpha-1}Gd}{\tau} \sum_{i\in[N]} \sum_{v\in[K]-1} p_i \eta_\ell^r (M^\alpha + B^\alpha) u_r^{1-\alpha} \mathbb{P}\left( \|\nabla F_i(x_{i,v}^r; \xi_{i,v}^r))\| \geq u_r \right)$$

$$\leq \frac{Gd}{\tau} K\eta_\ell^r d_r + \frac{GLK^2 d}{2\tau}(\eta_\ell^r)^2 u_r + \frac{2^{\alpha-1}Gd}{\tau} K\eta_\ell^r (M^\alpha + B^\alpha) u_r^{1-\alpha}.$$

We retain the $\alpha$ for clarity and to draw comparision to previous proofs, however we note that $\alpha = 1$ as higher moments do not exist. Now, to bound $B_1$, we use $L$-smoothness:

$$\|B_1\| \leq \frac{L\eta_\ell^r u_r K}{\tau}\|x_{t-1} - x_{r-1}\| \leq \frac{L\eta_\ell^r u_r K \operatorname{diam}(\mathcal{X})}{\tau}.$$

Collecting all inequalities gathered thus far gives

$$\mathbb{E}[F(x_t)] \leq \mathbb{E}[F(x_{t-1})] + \frac{\eta_t^2 L}{2}\mathbb{E}[\|A_1\|^2] + \widetilde{\beta}_1^t \eta_t \mathbb{E}\left[ \left\langle \nabla F(x_{t-1}), \frac{\widetilde{m}_0}{\sqrt{\widetilde{v}_t} + \tau} \right\rangle \right]$$

$$+ (1-\widetilde{\beta}_1)\eta_t \sum_{r=1}^{t} \widetilde{\beta}_1^{t-r} \left( \sum_{q=0}^{t-r} \mathbb{E}[B_{1,q}] - K\eta_\ell^r \mathbb{E}\left[ \left\| \frac{\nabla F(x_{r-1})}{\sqrt{\sqrt{\widetilde{\beta}_2^{t-r+1}\widetilde{v}_{r-1}} + \tau}} \right\|^2 \right] + \frac{L\eta_\ell^r u_r K \operatorname{diam}(\mathcal{X})}{\tau} \right)$$

$$+ (1-\widetilde{\beta}_1)\eta_t \sum_{r=1}^{t} \widetilde{\beta}_1^{t-r} \left( \frac{Gd}{\tau} K\eta_\ell^r d_r + \frac{GLK^2 d}{2\tau}(\eta_\ell^r)^2 u_r + \frac{2^{\alpha-1}Gd}{\tau} K\eta_\ell^r (M^\alpha + B^\alpha) u_r^{1-\alpha} \right).$$

We note the use of Jensen and convexity to ensure $\|\mathbb{E}[B_1]\| \leq \mathbb{E}[\|B_1\|]$. We now rearrange and telescope $t \in [1, T]$:

$$
\underbrace{(1 - \widetilde{\beta}_1) \sum_{t=1}^{T} \eta_t \sum_{r=1}^{t} \widetilde{\beta}_1^{t-r} \left( K\eta_\ell^r \mathbb{E} \left[ \left\| \frac{\nabla F(x_{r-1})}{\sqrt{\sqrt{\widetilde{\beta}_2^{t-r+1} \widetilde{v}_{r-1}} + \tau}} \right\|^2 \right] \right)}_{F_1} \leq \underbrace{\mathbb{E}[F(x_0)] - \mathbb{E}[F(x_T)]}_{F_2} + \underbrace{\sum_{t=1}^{T} \frac{\eta_t^2 L}{2} \mathbb{E}[\|A_1\|^2]}_{F_3}
$$

$$
+ \underbrace{\sum_{t=1}^{T} \eta_t \widetilde{\beta}_1^t \mathbb{E}\left[ \left\langle \nabla F(x_{t-1}), \frac{\widetilde{m}_0}{\sqrt{\widetilde{v}_t} + \tau} \right\rangle \right]}_{F_4} + \underbrace{(1 - \widetilde{\beta}_1) \sum_{t=1}^{T} \eta_t \sum_{r=1}^{t} \widetilde{\beta}_1^{t-r}}_{F_5} \left( \underbrace{\sum_{q=0}^{t-r} \mathbb{E}[B_{1,q}]}_{F_6} + \underbrace{\frac{L\eta_\ell^r u_r K \operatorname{diam}(\mathcal{X})}{\tau}}_{F_7} \right)
$$

$$
+ \underbrace{(1 - \widetilde{\beta}_1) \sum_{t=1}^{T} \eta_t \sum_{r=1}^{t} \widetilde{\beta}_1^{t-r}}_{F_5} \left( \underbrace{\frac{Gd}{\tau} K\eta_\ell^r d_r}_{F_8} + \underbrace{\frac{GLK^2 d}{2\tau} (\eta_\ell^r)^2 u_r}_{F_9} + \underbrace{\frac{2^{\alpha-1} Gd}{\tau} K\eta_\ell^r (M^\alpha + B^\alpha) u_r^{1-\alpha}}_{F_{10}} \right).
$$

We now aim to bound each term in the left hand side from below, and right hand side from above. Letting $\eta_t = \Theta(t^\omega)$, $\eta_\ell^t = \Theta(t^\nu)$, $d_t = \Theta(t^\gamma)$, and $u_t = \Theta(t^\zeta)$, we move to the asymptotic regime to simplify notation and suppress auxiliary constants for readability. We have that

$$
(1 - \widetilde{\beta}_1) \sum_{t=1}^{T} \sum_{r=1}^{t} \eta_t \widetilde{\beta}_1^{t-r} \eta_\ell^r = (1 - \widetilde{\beta}_1) \sum_{t=1}^{T} \eta_t \widetilde{\beta}_1^t \left( \sum_{r=1}^{t} \widetilde{\beta}_1^{-r} \eta_\ell^r \right) \gtrsim (1 - \widetilde{\beta}_1) \sum_{t=1}^{T} \eta_t \widetilde{\beta}_1^t \int_1^t \widetilde{\beta}_1^{-r} r^\nu \, dr. \tag{15}
$$

Then, L'Hôpital's rule allows us to derive an asymptotically sharp bound as follows:

$$
\int_1^t \widetilde{\beta}_1^{-r} r^\nu \, dr = \left[ \frac{\widetilde{\beta}_1^{-r} r^\nu}{-\log_e(\widetilde{\beta}_1)} \right]_{r=1}^{t} - \int_1^t \frac{\nu \widetilde{\beta}_1^{-r} r^{\nu-1}}{-\log_e(\widetilde{\beta}_1)} \, dr \gtrsim \frac{\widetilde{\beta}_1^{-t} t^\nu}{|\log_e(\widetilde{\beta}_1)|} \tag{16}
$$

Here, we used that $\nu \leq 0$ and $0 < \widetilde{\beta}_1 < e$. Asymptotic equivalence is verified via

$$
\lim_{t \to \infty} \frac{|\log_e(\widetilde{\beta}_1)| (\int_1^t \widetilde{\beta}_1^{-r} r^\nu \, dr)}{\widetilde{\beta}_1^{-t} t^\nu} = \lim_{t \to \infty} \frac{|\log_e(\widetilde{\beta}_1)| \widetilde{\beta}_1^{-t} t^\nu}{-\log_e(\widetilde{\beta}_1) \widetilde{\beta}_1^{-t} t^\nu + \nu \widetilde{\beta}_1^{-t} t^{\nu-1}} = 1.
$$

Therefore, the rightmost side of (16) is an asymptotically sharp approximation, relieving the condition $\nu \leq 0$ for validity of the approximation. Within $\widetilde{\beta}_1 \in (0, 1)$, the approximation diverges as expected, validating the asymptotic analysis. Recall that $|\Delta_r| \leq K\eta_\ell^r u_r$, which now gives via (16)

$$
\widetilde{\beta}_2^{t-r+1} \widetilde{v}_{r-1} \lesssim \sum_{z=1}^{r-1} \widetilde{\beta}_2^{r-1-z} \Delta_z^2 \lesssim \widetilde{\beta}_2^{r-1} \sum_{z=1}^{r-1} \widetilde{\beta}_2^{-z} (\eta_\ell^z)^2 u_z^2 \lesssim \max \left\{ \mathcal{O}(1), T^{2(\nu+\zeta)} \right\}. \tag{17}
$$

Here, we used $\widetilde{\beta}_2 \leq 1$ and $r \leq T$. We thus obtain

$$
(1 - \widetilde{\beta}_1) \sum_{t=1}^{T} \sum_{r=1}^{t} \eta_t \widetilde{\beta}_1^{t-r} \eta_\ell^r \gtrsim (1 - \widetilde{\beta}_1) \sum_{t=1}^{T} \eta_t \frac{t^\nu}{|\log_e(\widetilde{\beta}_1)|} \gtrsim (1 - \widetilde{\beta}_1) \int_1^T \frac{t^{\omega+\nu}}{\log_e(\widetilde{\beta}_1)} dt \approx \frac{(1 - \widetilde{\beta}_1) T^{\omega+\nu+1}}{(\omega + \nu + 1) |\log_e(\widetilde{\beta}_1)|}.
$$

Therefore as $\nu + \zeta < 0$, we conclude that

$$
F_1 \gtrsim \Omega \left( \frac{(1 - \widetilde{\beta}_1)}{(\omega + \nu + 1) \log_e(\widetilde{\beta}_1)} \cdot T^{\omega+\nu+1} \cdot \min_{t \in [T]} \mathbb{E}[\|\nabla F(x_t)\|^2] \right).
$$

Clearly, $F_2 \lesssim \mathcal{O}(1)$. To bound $F_3$, we have

$$F_3 = \sum_{t=1}^{T} \frac{\eta_t^2 L}{2} \left\| \frac{\widetilde{\beta}_1^t \widetilde{m}_0 + (1-\widetilde{\beta}_1) \sum_{r=1}^t \widetilde{\beta}_1^{t-r} \Delta_r}{\sqrt{\widetilde{v}_t} + \tau} \right\|^2 \lesssim \sum_{t=1}^{T} \frac{t^{2\omega}}{\tau^2} \left( \widetilde{\beta}_1^{2t} \|\widetilde{m}_0\|^2 + (1-\widetilde{\beta}_1)^2 \left\| \sum_{r=1}^t \widetilde{\beta}_1^{t-r} \Delta_r \right\|^2 \right)$$

$$\lesssim \frac{\mathcal{O}(1)}{\tau^2} + \frac{(1-\widetilde{\beta}_1)^2 \sum_{t=1}^T t^{2\nu+2\zeta+2\omega}}{\tau^2 (\log_e(\widetilde{\beta}_1))^2} \lesssim \frac{\mathcal{O}(1)}{\tau^2} + \frac{(1-\widetilde{\beta}_1)^2 \, T^{2(\nu+\zeta+\omega)+1}}{\tau^2 (\log_e(\widetilde{\beta}_1))^2}.$$

$F_4$ is bounded similarly after using Jensen,

$$|F_4| \leq \sum_{t=1}^{T} \eta_t \widetilde{\beta}_1^t \mathbb{E}\left[ \left\langle |\nabla F(x_{t-1})|, \frac{|\widetilde{m}_0|}{\sqrt{\widetilde{v}_t} + \tau} \right\rangle \right] \leq \sum_{t=1}^{T} \eta_t \widetilde{\beta}_1^t dG \cdot \max_{j \in [d]} \frac{|\widetilde{m}_0|_j}{\sqrt{[\widetilde{v}_t]_j} + \tau} \lesssim \mathcal{O}(1).$$

Bounding $F_5$ and $F_6$ is more complex. We begin by noting that

$$|\mathbb{E}[B_{1,q}]| \leq \sum_{j=1}^{d} \frac{G(1-\widetilde{\beta}_2)\widetilde{\beta}_2^{\frac{q}{2}}}{\tau^2} \cdot \mathbb{E}\left[ \frac{[\Delta_{t-q}^2 |\Delta_r|]_j}{\sqrt{[\widetilde{v}_{t-q}]_j}} \right]$$

$$\leq \sum_{j=1}^{d} \frac{G(1-\widetilde{\beta}_2)\widetilde{\beta}_2^{\frac{q}{2}}}{\tau^2} \cdot \mathbb{E}\left[ \frac{[\Delta_{t-q}^2 |\Delta_r|]_j}{\sqrt{\max\{[\widetilde{\beta}_2^{t-q}\widetilde{v}_0 + (1-\widetilde{\beta}_2)\sum_{o=1}^{t-q} \widetilde{\beta}_2^{t-q-o} \Delta_o^2]_j, \tau^2\}}} \right]$$

$$\lesssim \sum_{j=1}^{d} \frac{(1-\widetilde{\beta}_2)\widetilde{\beta}_2^{\frac{q}{2}}}{\tau^3} \cdot \mathbb{E}\left[ [\Delta_{t-q}^2 |\Delta_r|]_j \right].$$

Therefore,

$$F_5 F_6 \lesssim (1-\widetilde{\beta}_1) \sum_{t=1}^{T} \eta_t \sum_{r=1}^{t} \widetilde{\beta}_1^{t-r} (1-\widetilde{\beta}_2) \sum_{q=0}^{t-r} \widetilde{\beta}_2^{\frac{q}{2}} \cdot \mathbb{E}\left[ \Delta_{t-q}^2 |\Delta_r| \right]$$

$$\leq (1-\widetilde{\beta}_1) \sum_{t=1}^{T} \eta_t \sum_{r=1}^{t} \widetilde{\beta}_1^{t-r} (1-\widetilde{\beta}_2) \eta_\ell^r u_r \sum_{q=0}^{t-r} \widetilde{\beta}_2^{\frac{q}{2}} (\eta_\ell^{t-q} u_{t-q})^2.$$

Under the substitution $q \leftarrow t - \widetilde{q}$, we have that

$$F_5 F_6 \lesssim (1-\widetilde{\beta}_1) \sum_{t=1}^{T} \eta_t \sum_{r=1}^{t} \widetilde{\beta}_1^{t-r} (1-\widetilde{\beta}_2) \eta_\ell^r u_r \widetilde{\beta}_2^{\frac{t}{2}} \sum_{\widetilde{q}=r}^{t} \widetilde{\beta}_2^{\frac{-\widetilde{q}}{2}} (\eta_\ell^{\widetilde{q}} u_{\widetilde{q}})^2$$

$$\lesssim (1-\widetilde{\beta}_1) \sum_{t=1}^{T} \eta_t \sum_{r=1}^{t} \widetilde{\beta}_1^{t-r} (1-\widetilde{\beta}_2) \eta_\ell^r u_r \cdot 2^{\nu+\zeta} \frac{t^{2(\nu+\zeta)}}{|\log_e(\widetilde{\beta}_2)|}$$

$$\lesssim (1-\widetilde{\beta}_1) \sum_{t=1}^{T} \frac{t^{\omega+2(\nu+\zeta)}}{|\log_e(\widetilde{\beta}_2)|} \widetilde{\beta}_1^t (1-\widetilde{\beta}_2) \sum_{r=1}^{t} \widetilde{\beta}_1^{-r} r^{\nu+\zeta}$$

$$\lesssim (1-\widetilde{\beta}_1) \sum_{t=1}^{T} (1-\widetilde{\beta}_2) \frac{t^{\omega+3(\nu+\zeta)}}{|\log_e(\widetilde{\beta}_1)||\log_e(\widetilde{\beta}_2)|} \approx \frac{(1-\widetilde{\beta}_1)(1-\widetilde{\beta}_2)}{|\log_e(\widetilde{\beta}_1)||\log_e(\widetilde{\beta}_2)|} \cdot \max\left\{ \mathcal{O}(1), T^{\omega+3(\nu+\zeta)+1} \right\}.$$

As $\mathcal{O}(1)$ terms are subsumed by $F_4$, $F_5 F_7$ is bounded via

$$(1-\widetilde{\beta}_1) \sum_{t=1}^{T} \eta_t \sum_{r=1}^{t} \widetilde{\beta}_1^{t-r} \frac{L \eta_\ell^r u_r K \operatorname{diam}(\mathcal{X})}{\tau} \lesssim (1-\widetilde{\beta}_1) \sum_{t=1}^{T} \eta_t \sum_{r=1}^{t} \widetilde{\beta}_1^t \frac{\eta_\ell^r u_r \widetilde{\beta}_1^{-r}}{\tau}$$

$$\lesssim (1-\widetilde{\beta}_1) \sum_{t=1}^{T} \frac{t^{\nu+\zeta+\omega}}{\tau |\log_e(\widetilde{\beta}_1)|} \lesssim \frac{(1-\widetilde{\beta}_1) T^{\omega+\nu+\zeta+1}}{\tau |\log_e(\widetilde{\beta}_1)|}.$$

The remaining terms may also be bounded as follows:

$$F_5 F_8 \lesssim \frac{(1-\widetilde{\beta}_1)}{\tau} \sum_{t=1}^{T} \eta_t \sum_{r=1}^{t} \widetilde{\beta}_1^{t-r} \eta_\ell^r d_r \lesssim \frac{(1-\widetilde{\beta}_1)}{\tau} \sum_{t=1}^{T} \eta_t \sum_{r=1}^{t} \widetilde{\beta}_1^t \widetilde{\beta}_1^{-r} r^{\nu+\gamma}$$

$$\lesssim \frac{(1-\widetilde{\beta}_1)}{|\log_e(\widetilde{\beta}_1)|} \sum_{t=1}^{T} t^\omega t^{\nu+\gamma} \lesssim \frac{(1-\widetilde{\beta}_1)}{|\log_e(\widetilde{\beta}_1)|} \max\{T^{\omega+\nu+\gamma+1}, \mathcal{O}(1)\}$$

where $F_9$ and $F_{10}$ can be bounded via

$$F_5 F_9 \lesssim (1-\widetilde{\beta}_1) \sum_{t=1}^{T} \sum_{r=1}^{t} \frac{\eta_t \widetilde{\beta}_1^{t-r} (\eta_\ell^r)^2 u_r}{\tau} \lesssim \frac{(1-\widetilde{\beta}_1)}{|\log_e(\widetilde{\beta}_1)|} \sum_{t=1}^{T} \sum_{r=1}^{t} \frac{\eta_t \widetilde{\beta}_1^{t-r} r^{2\nu+\zeta}}{\tau} \lesssim \frac{T^{2\nu+\zeta+1+\omega}}{\tau},$$

$$F_5 F_{10} \lesssim (1-\widetilde{\beta}_1) \sum_{t=1}^{T} \sum_{r=1}^{t} \eta_t \widetilde{\beta}_1^{t-r} \frac{\eta_\ell^r u_r^{1-\alpha}}{\tau} \lesssim (1-\widetilde{\beta}_1) \sum_{t=1}^{T} \sum_{r=1}^{t} t^\omega \widetilde{\beta}_1^t \frac{\widetilde{\beta}_1^{-r} r^{\nu+\zeta(1-\alpha)}}{\tau}$$

$$\lesssim \sum_{t=1}^{T} t^\omega \frac{(1-\widetilde{\beta}_1)}{|\log_e(\widetilde{\beta}_1)|} t^{\nu+\zeta(1-\alpha)} \lesssim \frac{(1-\widetilde{\beta}_1)}{|\log_e(\widetilde{\beta}_1)|} T^{\omega+\nu+\zeta(1-\alpha)+1}.$$

Standard calculations imply that under the conditions (13), the dominating terms are $F_7$, $F_{10}$ with order $\mathcal{O}(1)$. Within the derived upper bound, $\zeta > 0$ destabilizes $F_7$ and decays $F_{10}$ to 0, while $\zeta < 0$ gives the analogous properties with $F_7$ and $F_{10}$ swapped. $\qquad\square$

# D. Experiment Setup & Full Results

In this section, we present the experimental setups and results across two primary domains: synthetic data and natural language processing tasks. More precisely, we evaluate the performance of TailOPT instantiations with state-of-the-art benchmarks on convex models (with synthetic data), transformer encoders, as well as generative models. For convex, synthetic experiments, we construct datasets to emulate heavy-tailed stochastic gradients, focusing on linear regression models trained under contaminated label noise. The design includes generating feature matrices and labels while injecting noise from heavy-tailed distributions to study convergence behaviors. Additionally, we introduce the SynToken dataset, which models the heavy-tailed distribution of token frequencies observed in natural language processing. For brevity, we only include the results of the SynToken dataset, denoted 'Synthetic data', in the main text (Figure 1). This allows us to evaluate learning algorithms in controlled settings, easing out and exploring the effects of both common and rare features.

For assessing the optimization of transformer encoders on natural language processing tasks, we evaluate RoBERTa (Liu et al., 2019) on the General Language Understanding Evaluation (GLUE) benchmark (Wang et al., 2019), which encompasses a diverse range of tasks such as sentiment analysis, paraphrase detection, and natural language inference. By finetuning RoBERTa on GLUE, we assess its generalization capabilities and robustness. The benchmark's inclusion of multiple datasets ensures a comprehensive evaluation of model performance across various linguistic phenomena. Additionally, we also evaluate the capabilities of the T5 (Raffel et al., 2020) generative model on WMT machine translation tasks (Foundation, 2019). These experiments provide insights into the behavior of optimization algorithms and pretrained models under realistic and challenging conditions. For RoBERTa, we optimize over GLUE across 10 simulated compute nodes, whereas for T5, we model 3 compute node finetuning on WMT benchmark datasets.

## D.1. Convex Models

### D.1.1. DATA GENERATION PROCESS

To simulate heavy-tailed stochastic gradients in a simple yet controlled linear regression setting, we generated a synthetic dataset as follows. The feature matrix $X \in \mathbb{R}^{M \times m}$ was constructed with entries drawn independently from a standard normal distribution, $X_{ij} \sim \mathcal{N}(0, 1)$. The true weight vector $w_{\text{true}} \in \mathbb{R}^m$ was sampled from $\mathcal{N}(0, I_m)$, where $I_m$ is the $m \times m$ identity matrix.

The true labels were computed using:

$$y_{\text{true}} = X w_{\text{true}}.$$

To induce heavy-tailed stochastic gradients, we injected noise into the label vector by adding a noise term $\xi$, resulting in contaminated labels:

$$\hat{y} = y_{\text{true}} + \xi,$$

where $\xi \in \mathbb{R}^M$ is a noise vector with entries drawn independently from a heavy-tailed distribution $\mathcal{D}$. For simplicity, we assume coordinate-wise independence of the noise components.

After generating the dataset, we distributed the data across $n = 10$ datacenters in an IID fashion. Notably, the heavy-tailed noise was injected once prior to distribution, and no additional data were generated afterward. This approach ensured that the same contaminated training data are used locally throughout the training process.

### D.1.2. LINEAR REGRESSION MODEL

We consider a single-layer neural network without biases, parameterized by $w \in \mathbb{R}^m$, which is equivalent to linear regression. Training is performed using the contaminated labels $(X, \hat{y})$ with the mean-squared error (MSE) loss function:

$$\mathcal{L}(w) = \frac{1}{2} \|\hat{y} - Xw\|^2.$$

The gradient of the loss with respect to $w$ is given by:

$$\nabla_w \mathcal{L}(w) = -X^\top (\hat{y} - Xw).$$

Substituting $\hat{y} = y_{\text{true}} + \xi = X w_{\text{true}} + \xi$, we have:

$$\nabla_w \mathcal{L}(w) = -X^\top (X w_{\text{true}} + \xi - Xw) = -X^\top X (w_{\text{true}} - w) - X^\top \xi.$$

Simplifying, we obtain:

$$\nabla_w \mathcal{L}(w) = X^\top X(w - w_{\text{true}}) - X^\top \xi.$$

The term $-X^\top \xi$ reflects the influence of the heavy-tailed noise on the gradient. Given that $X$ has Gaussian entries and $\xi$ follows a heavy-tailed distribution, the stochastic gradients $\nabla_w \mathcal{L}(w)$ are also heavy-tailed.

### D.1.3. THE SYNTOKEN DATASET

To model the heavy-tailed nature of token frequencies observed in natural language processing, we created the synthetic SynToken dataset. In natural language, word or token usage often follows a heavy-tailed distribution. That is, a small number of tokens appear very frequently, while a large number of tokens appear infrequently but carry significant contextual information. In our dataset, we partitioned the feature space into common and rare features to reflect this phenomenon. Specifically, we designated the first $p = 10\%$ of the columns of $X$ as common features and the remaining $90\%$ as rare features. The common features were generated by sampling from a Bernoulli distribution with a high probability of success:

$$X_{\text{common}} \sim \text{Bernoulli}(0.9),$$

resulting in features that are frequently active. The rare features were sampled from a Bernoulli distribution with a low probability of success:

$$X_{\text{rare}} \sim \text{Bernoulli}(0.1),$$

introducing sparsity and emulating infrequently occurring tokens. The complete feature matrix $X$ was formed by concatenating $X_{\text{common}}$ and $X_{\text{rare}}$:

$$X = [X_{\text{common}}, \ X_{\text{rare}}].$$

The weight vector $w$ was sampled from a standard multivariate normal distribution, $w \sim \mathcal{N}(0, I_m)$, consistent with the previous setup. Noise injection was analogously applied to the labels as before. This approach was taken to mimic the key characteristics of tokenization and word embeddings in natural language processing, via a minimal yet effective model. One benefit of synthetic datasets is that by simulating the distribution of common and rare tokens, the SynToken dataset allows us to study the effects of heavy-tailed data distributions on learning algorithms in a controlled setting. Additionally, we note that the problem being studied is $\mu$-strongly convex.

### D.2. Synthetic Data Experiments Discussion

**Does the heavy-tailed distribution of covariates matter?** Figure 4 (a) and (c) illustrate that a heavy-tailed distribution of token frequencies has significant impacts on the performance of optimization strategies. In (a), RMSProp-$BiClip$ performs competitively under standard tokenization. However, in (c), heavy-tailed tokenization applied to the feature matrix destabilizes RMSProp-$BiClip$. Interestingly, under tokenized conditions without noise, RMSProp exhibits oscillatory behavior, whereas Adam maintains relative stability. This is consistent with the interpretation of Adam as incorporating an exponentially decaying moving average of the gradient's first moment, which augments optimization stability. Upon noise injection, best performing hyperparameters for RMSProp-$BiClip$ does not show oscillatory behavior, but is larger in terms of distance $\|w_* - \hat{w}\|$ than the case without noise.

**Does noise matter?** When noise is injected into the labels, the performance dynamics shift considerably. outer optimizer adaptive or non-adaptive methods *combined* with inner optimizer SGD perform poorly, which may indicate that inner optimizers should take a focal role in addressing the challenges posed by heavy-tailed noise. While the choice of the outer optimizer may appear to a limited impact on the binary question of learnability for this specific synthetic data (i.e., "Can the algorithm decrease distance to the true $w_*$ or not?"), under tokenized conditions with heavy-tailed noise (Figure 4(d)), outer optimizer Adam demonstrates the best performance. Figure 4 reveals that heavy-tailed noise generally destabilizes all algorithms, including adaptive methods, clipped approaches, and pure SGD (c.f., minimum values in (a) and (c) to (b) and (d)). Notably, coordinate-wise $BiClip$ consistently outperforms $L_2$ clipping, aligning with the results in Table 1.

**How far should these results generalize?** A word of caution is warranted against overgeneralization. These results are derived from a simplified regression model, limiting the ability to generalize the observed trends. Nevertheless, the experiments underscore the pronounced effects of heavy-tailed noise in a controlled synthetic environment and highlight the noise-mitigating capabilities of optimizers such as Adam, RMSProp, and $BiClip$. Additionally, it is important to note that real-world transformer models often comprise tens of millions to billions of parameters.

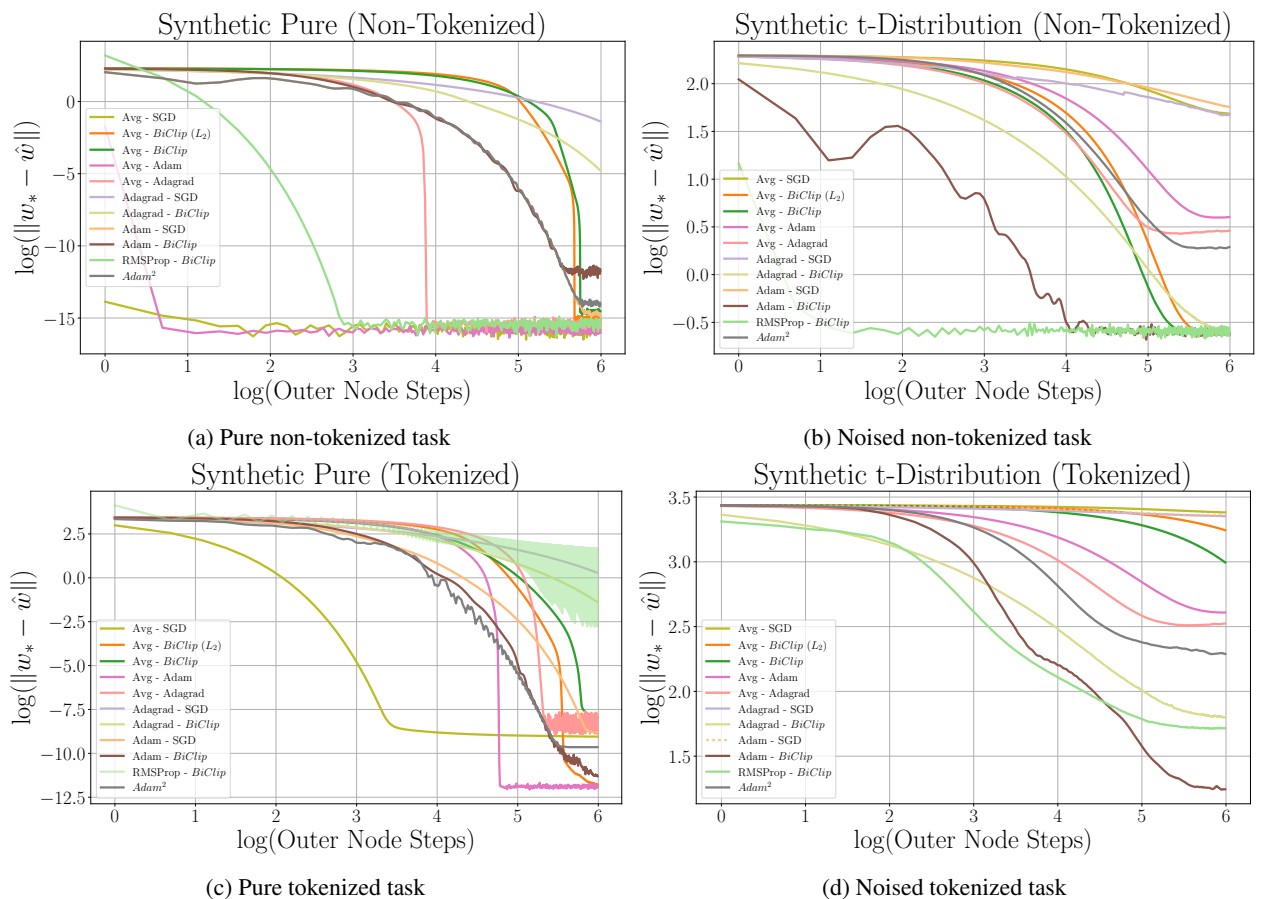

**Figure 4:** (Top) The results on the non-tokenized synthetic dataset are presented. In the absence of noise injection, Avg-Adam, Avg-SGD, and RMSProp-*BiClip* demonstrate the most competitive performance. However, under heavy-tailed noise injection, RMSProp-*BiClip* and Adam-*BiClip* achieve the highest performance, while Avg-SGD exhibits among the poorest outcomes. Notably, oscillations observed in Adam-*BiClip* may reflect the impact of amplified update learning rates in the outer optimizer, potentially enabling finer-grained exploration of the optimization landscape. (Bottom) Tokenization drastically alters algorithmic performance. Without noise, Avg-SGD decays the fastest, while Avg-Adam converges to a superior optimum. However, when synthetic, unbiased heavy-tailed noise is introduced, Avg-SGD becomes highly unstable, whereas Adam-*BiClip* and RMSProp-*BiClip* consistently deliver the best results.

## D.3. Transformer Encoders

The General Language Understanding Evaluation (GLUE) benchmark (Wang et al., 2019) serves as a comprehensive framework for evaluating natural language understanding (NLU) models across a diverse range of tasks. By incorporating datasets that span various linguistic challenges, GLUE provides a rigorous testbed for assessing the generalization capabilities of NLP models. RoBERTa is a state-of-the-art transformer-based model designed to enhance the performance of the original BERT architecture through improved pretraining strategies. Proposed by Liu et al. (2019), RoBERTa optimizes BERT by refining its training setup, enabling more robust natural language understanding (NLU) across diverse tasks.

Table 3: Evaluation results on GLUE Benchmark datasets during test time. Metrics: CoLA (Matthews Correlation Coefficient, MCC), SST-2 (Accuracy), MRPC (Accuracy/F1), STS-B (Spearman/Pearson), QQP (Accuracy/F1), MNLI (Accuracy), QNLI (Accuracy), RTE (Accuracy). Entries marked with 0.0 indicate the actual metric value (averaged across the granularity of each datapoint in the baseline dataset), which implies random guessing or failure to learn. Top first, second, and third best-performing algorithms are highlighted. We note that nested optimization algorithms utilizing adaptivity or coordinate-wise $BiClip$ on both inner and outer optimizers generally achieve greater than 80% averaged performance (out of 100%). For $Adam^2$, preconditioners are transmitted between the inner and outer optimizers, whereas DiLoCo requires maintaining preconditioners on the inner optimizers, both of which incur significant communication or memory overhead.

| Algorithm | MNLI | QNLI | QQP (Acc/F1) | RTE | SST-2 | MRPC (Acc/F1) | CoLA | STS-B (S/P) | Average |
|---|---|---|---|---|---|---|---|---|---|
| **Avg-SGD** (McMahan et al., 2017) | 81.13 | 83.21 | 78.71/78.69 | 57.40 | 90.94 | 67.30/80.52 | 0.0 | 26.76/28.20 | 61.17 |
| **Avg-**$L_2Clip$ (Yang et al., 2022) | 81.82 | 85.68 | 80.00/79.82 | 54.51 | 91.97 | 68.38/81.22 | 0.0 | 41.27/40.96 | 64.15 |
| **Avg-**$BiClip$ ($L_2$) | 81.95 | 86.16 | 84.62/79.89 | 55.59 | 92.31 | 68.38/81.23 | 0.0 | 36.93/37.22 | 64.03 |
| **Avg-Adagrad** | 84.70 | 88.79 | 87.09/83.34 | 64.26 | 93.34 | 71.56/82.63 | 27.72 | 81.93/81.26 | 76.97 |
| **Avg-Adam** | 84.97 | 89.47 | 87.66/84.09 | 64.62 | 93.80 | 81.86/87.74 | 41.41 | 86.21/86.55 | 80.76 |
| **Avg-**$BiClip$ | 85.08 | 89.45 | 87.83/84.12 | 66.06 | 94.03 | 71.32/82.45 | 41.40 | 84.08/84.48 | 79.12 |
| $Bi^2Clip$ ($L_2$) | 84.31 | 89.20 | 86.36/82.60 | 72.20 | 93.34 | 86.52/90.23 | 60.02 | 82.41/83.00 | 82.74 |
| **Adagrad-SGD** (Reddi et al., 2021) | 82.40 | 86.61 | 82.51/77.68 | 71.48 | 92.08 | 85.53/89.52 | 47.80 | 40.37/42.24 | 72.69 |
| **RMSProp-SGD** (Reddi et al., 2021) | 84.20 | 88.46 | 87.12/83.30 | 72.56 | 91.85 | 85.50/89.17 | 52.39 | 45.72/41.80 | 74.73 |
| **Adam-SGD** (Reddi et al., 2021) | 82.93 | 86.98 | 85.99/80.87 | 66.78 | 90.71 | 87.01/90.09 | 49.93 | 44.48/41.26 | 73.37 |
| **Adam-**$L_2Clip$ | 82.54 | 86.69 | 85.88/80.72 | 59.92 | 89.67 | 85.29/89.90 | 48.54 | 69.19/67.16 | 76.86 |
| **Adagrad-**$BiClip$ | 85.54 | 90.02 | 88.60/85.05 | 73.36 | 93.23 | 85.78/89.86 | 48.87 | 84.03/85.90 | 82.75 |
| **RMSProp-**$BiClip$ | 85.56 | 89.82 | 88.50/84.44 | 70.75 | 93.69 | 84.80/88.92 | 50.99 | 87.65/87.79 | 82.99 |
| **Adam-**$BiClip$ | 84.26 | 89.20 | 88.64/84.74 | 69.67 | 92.43 | 86.52/90.09 | 56.12 | 82.83/79.71 | 82.20 |
| **Adam-**$BiClip$ ($L_2$) | 83.18 | 86.47 | 85.63/80.27 | 67.50 | 89.56 | 86.02/89.65 | 53.17 | 74.73/73.48 | 79.06 |
| $Adam^2$ (Wang et al., 2021b) | 85.11 | 88.87 | 89.04/85.51 | 71.48 | 92.66 | 87.50/91.03 | 52.70 | 84.47/83.82 | 82.93 |
| **DiLoCo** (Douillard et al., 2024) | 85.68 | 89.87 | 88.78/85.19 | 67.87 | 91.89 | 87.99/91.20 | 54.77 | 85.93/84.76 | 83.08 |
| $Bi^2Clip$ | 85.06 | 89.73 | 84.93/83.97 | 76.53 | 93.80 | 89.21/92.44 | 60.08 | 87.07/86.89 | 84.52 |

## D.4. Generative Models

We additionally evaluate our method using T5 (Raffel et al., 2020), a state-of-the-art text-to-text transformer model developed by Google Research. T5 unifies natural language processing tasks under a text-to-text framework, where both inputs and outputs are text strings, making it highly versatile across tasks such as summarization, translation, and classification. The model was pretrained on the Colossal Clean Crawled Corpus (C4) using a span corruption objective and is available in multiple sizes, ranging from T5-Small (60M parameters) to T5-XXL (11B parameters). This unified framework and scalability allow T5 to excel in a wide range of tasks, making it a strong baseline for evaluating our proposed method.

To evaluate machine translation tasks, we utilize the WMT datasets, a widely recognized benchmark for translation research (Foundation, 2019). Specifically, we finetune T5 on the TED Talks and News Commentary datasets. The TED Talks dataset, originally sourced from IWSLT 2017 (Cettolo et al., 2017), provides multilingual translations of TED Talk transcripts, offering diverse linguistic and domain-specific challenges. In contrast, the News Commentary dataset contains parallel text derived from news articles in various languages, presenting a more formal and structured domain. These datasets represent distinct styles and linguistic features, providing a rigorous evaluation of algorithm agility in optimizing across various domains or tasks.

Table 4: Evaluation results on machine translation benchmarks. Metrics reported are BLEU and METEOR scores for various language pairs across the TED Talks and News Commentary datasets. The final column represents the average score across all metrics for each algorithm.

| Algorithm | TED Talks (en-de) | | TED Talks (en-fr) | | News Commentary (en-fr) | | Average |
|---|---|---|---|---|---|---|---|
| | BLEU | METEOR | BLEU | METEOR | BLEU | METEOR | |
| **Avg-SGD** | 28.02 | 58.52 | 27.48 | 54.67 | 30.07 | 54.13 | 42.15 |
| **Avg-**$L_2Clip$ | 28.99 | 58.94 | 29.66 | 57.40 | 31.02 | 56.73 | 43.79 |
| $Bi^2Clip$ | **29.41** | **59.18** | 30.70 | **58.13** | **31.79** | **57.69** | **44.48** |
| $Adam^2$ | 28.06 | 58.05 | **30.94** | 57.48 | 30.97 | 55.85 | 43.56 |

## D.5. Performance under Non-IID Data

### D.5.1. CUSTOM SHAKESPEARE DATASET

Though not the main focus of this work, in this section, we aim to briefly evaluate the performance of TailOPT and baselines under non-datacenter, distributed environments. We utilized the LEAF repository (Caldas et al., 2018), originally a benchmark suite for federated learning, which provides datasets, tools, and baselines to evaluate algorithms under real-world conditions. LEAF emphasizes non-IID data distributions, enabling the study of federated systems where data is naturally heterogeneous across smaller compute nodes. Among the datasets in LEAF, we modified the Shakespeare dataset, originally designed for next-character prediction, where each user now represented a character from Shakespeare's works. After preprocessing, the dataset contained 1144 inner compute nodes, each corresponding to a character's dialogue, with substantial variations in sample sizes, vocabulary, and syntax across compute nodes. This structure mirrors the imbalanced, domain-specific data distributions often encountered in federated learning. We modify the Shakespeare dataset by redefining the prediction task from next-character prediction to next-token predictions.

Table 5: Perplexity scores on the Federated Shakespeare Next Word Prediction Task at a 0.1% participation rate, for distillGPT-2 architecture finetuning after 3 communication rounds.

| Algorithm | Avg-SGD | Avg-$L_2Clip$ | Avg-$BiClip$ | RMSProp-$BiClip$ | $Bi^2Clip$ | $Adam^2$ |
|---|---|---|---|---|---|---|
| Perplexity Score | 1.9813 | 2.0126 | **1.7827** | 2.0054 | 1.9112 | 1.9445 |

### D.5.2. CUSTOM PHILOSOPHER DATASET

To mitigate potential data leakage, we constructed a custom dataset, termed the Philosopher Dataset, to evaluate the non-IID setting and facilitate training from scratch. The Philosopher Dataset was synthesized by allocating each literary work to one of eight compute nodes, followed by an 80-20 train-test split. These texts were open sourced from Project Gutenberg[3], an extensive online repository offering over 75,000 classic or traditional books while strictly adhering to copyright protections.

Table 6: Composition of the Philosopher Dataset.

| Title | Author | Translator |
|---|---|---|
| The Critique of Pure Reason | Immanuel Kant | J. Meiklejohn |
| The Collected Works of William Hazlitt, Volume One | William Hazlitt | - |
| The Works of Jane Austen | Jane Austen | - |
| The Republic | Plato | Benjamin Jowett |
| War and Peace | Leo Tolstoy | - |
| The Federalist Papers | Alexander Hamilton, John Jay, James Madison | - |
| The Count of Monte Cristo | Alexandre Dumas | - |
| The Brothers Karamazov | Fyodor Dostoevsky | Constance Garnett |

We instantiated a shallower GPT-2 architecture comprising 2 layers, 256 embedding dimensions, and 4 attention heads. This model was trained from scratch on the Philosopher Dataset. The training results are summarized in Table 7.

Table 7: Perplexity scores on the Philosopher Next Word Prediction Task at a 100% participation rate for the compressed GPT-2 architecture after 3 communication rounds.

| Algorithm | Avg-SGD | Avg-$L_2Clip$ | Avg-$BiClip$ | RMSProp-$BiClip$ | $Bi^2Clip$ | $Adam^2$ |
|---|---|---|---|---|---|---|
| Perplexity Score | 2.6361 | 2.1183 | **1.6266** | 1.7983 | 2.3488 | 2.5861 |

**Discussion.** In the synthesized non-IID setting, we observe that algorithmic instantiations employing joint adaptivity or adaptive approximations–i.e., incorporating adaptivity or its efficient approximations at both the inner and outer optimizers– tend to underperform slightly. This aligns with the theoretical intuition that highly sensitive, rapidly adapting optimizers are more susceptible to unmitigated client drift, effectively overfitting to the biases of local data shards at the inner optimizers. However, Avg-$BiClip$, which integrates a clipping mechanism to regulate noise variance and stabilize optimization dynamics, exhibits notably robust performance. In particular, Avg-$BiClip$ achieves the strongest results in settings with high

---

[3]https://www.gutenberg.org/

data heterogeneity across compute nodes, suggesting that $BiClip$ mitigates not only noise variance but also client drift. We further compare these findings to results on the synthetic dataset (Appendix D.1) where noise-injected data were distributed IID across nodes, contrasting with the Shakespeare and Philosopher datasets. We note that the perplexities obtained are lower compared to those achieved on larger text datasets, such as WikiText-103 or large-scale Common Crawl subsets (e.g., distillGPT reportedly achieves a perplexity of around 16 on the WikiText-103 benchmark, a long-term dependency language modeling dataset)[4]. This arises from the smaller size of the Shakespeare and Philosopher datasets in comparison to larger benchmarks. We provide the optimal hyperparameters for the non-IID experiments in Table 15.

### D.6. Gradient Distributions

Figure 5 highlights how gradient distributions can be distinctly altered by adaptive or clipping operations.

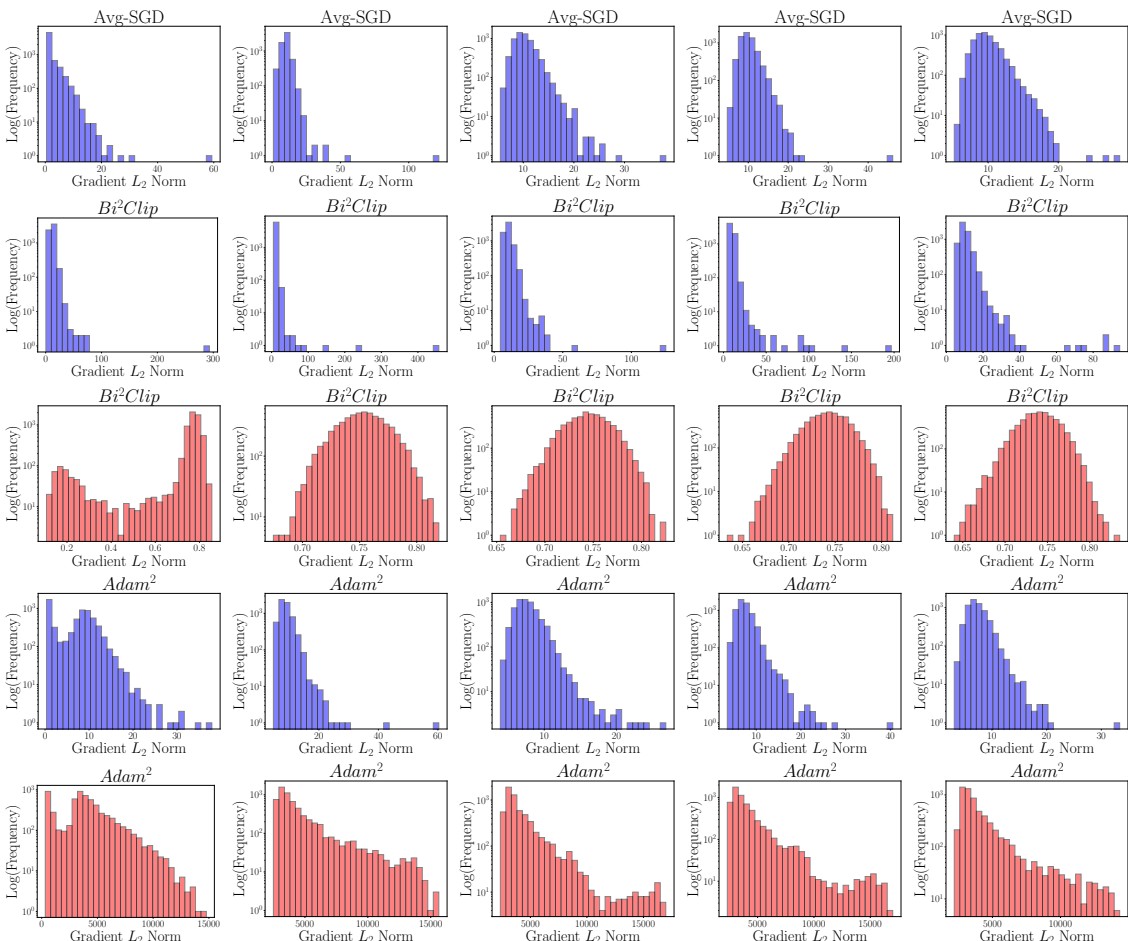

Figure 5: Gradient statistics for MNLI across different algorithms for the first 5 communication rounds, where rounds increase from left to right. (Top) We visualize local minibatch stochastic gradient distributions, where the outliers can dominate model updates upon outer aggregation. The $BiClip$ and Adam optimizers mitigate this phenomenon. (Middle) Row 2 displays the local gradients accumulated from all inner optimizers during $Bi^2Clip$ prior to clipping, which uncovers the presence of outliers akin to those visible in Avg-SGD. In Row 3, the identical gradients are plotted after applying the coordinate-wise $BiClip$ operation. We observe that $BiClip$ stabilizes updates by rescaling large and small gradient coordinates, constraining model update lengths within a defined range. (Bottom) Similar to above, Row 4 shows the accumulated gradient lengths across all inner optimizers while training via $Adam^2$. Optimal inner optimizer learning rates are 0.0059, 0.5, and 1.8$e$-5 for Avg-SGD, $Bi^2Clip$, and $Adam^2$, respectively, with corresponding outer optimizer learning rates of 1 and 3.2$e$-4 for the latter two algorithms. Test-time results show that $Bi^2Clip$ outperforms $Adam^2$, which in turn outperforms Avg-SGD (Table 1). Finally, we note that upon centering, the aggregate update gradient histograms in red depict the stochastic gradient noise distributions upon application of the optimizer strategy. $BiClip$ attenuates the pure gradient noise (in blue) by projecting the noise distribution to an almost bell-shaped curve (in red), while Adam implicitly samples gradient noise from a skewed distribution.

---

[4]https://github.com/huggingface/transformers/tree/main/examples/research_projects/distillation

## D.7. Hyperparameter Sweep Grids

The sweep grids in Tables 8, 9 were determined by first performing a coarser sweep using an approximate grid, then localizing near the discovered well-performing hyperparameters.

Table 8: Hyperparameter sweeps on gradient clipping parameters. i_u, i_d = inner optimizer $u$, $d$, o_u, o_d = outer optimizer $u$, $d$.

| Algorithm | i_u | i_d | o_u | o_d |
|---|---|---|---|---|
| **Avg-SGD** | - | - | - | - |
| **Avg-$L_2$Clip SGD** | np.linspace($10^{-4}$, 1.5, 12) | 0.0 | - | - |
| **Avg-$BiClip$** | np.linspace($10^{-4}$, 1.5, 4) | np.linspace($10^{-7}$, i_u, 4) | - | - |
| **Avg-$BiClip$ ($L_2$)** | np.linspace($10^{-4}$, 1.5, 4) | np.linspace($10^{-7}$, i_u, 4) | - | - |
| **Avg-Adagrad** | - | - | - | - |
| **Avg-Adam** | - | - | - | - |
| **Adagrad-SGD** | - | - | - | - |
| **RMSProp-SGD** | - | - | - | - |
| **Adam-SGD** | - | - | - | - |
| **Adagrad-$BiClip$** | np.linspace($10^{-4}$, 1.5, 3) | np.linspace($10^{-7}$, i_u, 3) | - | - |
| **RMSProp-$BiClip$** | np.linspace($10^{-4}$, 1.5, 3) | np.linspace($10^{-7}$, i_u, 3) | - | - |
| **Adam-$L_2$Clip** | np.linspace($10^{-4}$, 1.5, 12) | 0.0 | - | - |
| **Adam-$BiClip$** | np.logspace($-2$, 1, 5) | np.linspace($10^{-7}$, i_u, 3) | - | - |
| **Adam-$BiClip$ ($L_2$)** | np.linspace($10^{-4}$, 1.5, 3) | np.linspace($10^{-7}$, i_u, 3) | - | - |
| $Adam^2$ | - | - | - | - |
| $Bi^2Clip$ (**Coordinate-wise**) | np.linspace($10^{-4}$, 1.5, 3) | np.linspace($10^{-7}$, i_u, 3) | np.linspace($10^{-4}$, 1.5, 3) | np.linspace($10^{-7}$, o_u, 3) |
| $Bi^2Clip$ ($L_2$) | np.logspace($-1$, 0.5, 3) | np.linspace($10^{-7}$, i_u, 3) | np.logspace($-1$, 0.5, 3) | np.linspace($10^{-7}$, o_u, 3) |
| **DiLoCo** | - | - | - | - |

Table 9: Hyperparameter sweeps. ilr = inner optimizer learning rate, olr = outer optimizer learning rate, ieps = inner optimizer $\varepsilon$, oeps = outer optimizer $\varepsilon$. Additionally, DiLoCo swept over the nesterov learning rates $(0.9, 0.95)$, and inner optimizer weight decay parameters $(10^{-1}, 10^{-4})$, as reported in prior works.

| Algorithm | ilr | olr | ieps | oeps |
|---|---|---|---|---|
| **Avg-SGD** | np.logspace($-9$, 1, 100) | - | - | - |
| **Avg-$L_2$Clip SGD** | np.linspace($10^{-9}$, 1, 10) | - | - | - |
| **Avg-$BiClip$** | np.linspace($10^{-9}$, 1, 10) | - | - | - |
| **Avg-$BiClip$ ($L_2$)** | np.linspace($10^{-9}$, 1, 10) | - | - | - |
| **Avg-Adagrad** | np.linspace($10^{-9}$, 1, 30) | - | $\{10^{-8}, 10^{-6}, 10^{-4}, 10^{-3}\}$ | - |
| **Avg-Adam** | np.linspace($10^{-9}$, 1, 30) | - | $\{10^{-8}, 10^{-6}, 10^{-4}, 10^{-3}\}$ | - |
| **Adagrad-SGD** | np.linspace($10^{-5}$, 0.1, 7) | np.logspace($-5$, $-1$, 7) | - | $\{10^{-7}, 10^{-5}, 10^{-3}\}$ |
| **RMSProp-SGD** | np.linspace($10^{-5}$, 0.1, 7) | np.linspace($10^{-5}$, 0.1, 7) | - | $\{10^{-7}, 10^{-5}, 10^{-3}\}$ |
| **Adam-SGD** | np.linspace($10^{-5}$, 0.1, 7) | np.logspace($-5$, $-1$, 7) | - | $\{10^{-7}, 10^{-5}, 10^{-3}\}$ |
| **Adagrad-$BiClip$** | np.linspace($10^{-5}$, 0.1, 4) | np.logspace($-5$, $-1$, 4) | - | $\{10^{-7}, 10^{-5}, 10^{-3}\}$ |
| **RMSProp-$BiClip$** | np.linspace($10^{-5}$, 0.1, 4) | np.logspace($-5$, $-1$, 4) | - | $\{10^{-7}, 10^{-5}, 10^{-3}\}$ |
| **Adam-$L_2$Clip** | np.linspace($10^{-5}$, 0.1, 4) | np.linspace($10^{-5}$, 0.1, 4) | - | $\{10^{-7}, 10^{-5}, 10^{-3}\}$ |
| **Adam-$BiClip$** | np.logspace($-6$, $-1$, 5) | np.logspace($-6$, $-1$, 5) | - | $\{10^{-7}, 10^{-5}, 10^{-3}\}$ |
| **Adam-$BiClip$ ($L_2$)** | np.linspace($10^{-5}$, 0.1, 4) | np.linspace($10^{-5}$, 0.1, 4) | - | $\{10^{-7}, 10^{-5}, 10^{-3}\}$ |
| $Adam^2$ | np.logspace($-6$, $-1$, 5) | np.logspace($-6$, $-1$, 5) | $\{10^{-7}, 10^{-5}, 10^{-3}\}$ | $\{10^{-7}, 10^{-5}, 10^{-3}\}$ |
| $Bi^2Clip$ (**Coordinate-wise**) | np.linspace($10^{-9}$, 1, 3) | np.linspace($10^{-9}$, 1, 3) | - | - |
| $Bi^2Clip$ ($L_2$) | np.logspace($-1$, 0.5, 3) | np.logspace($-1$, 0.5, 3) | - | - |
| **DiLoCo** | np.logspace($-5$, $-1$, 5) | $\{1, 0.7, 0.5, 10^{-1}, 10^{-2}\}$ | - | $\{10^{-7}, 10^{-5}, 10^{-3}\}$ |

## D.8. Optimal Hyperparameters

In this subsection, we display the optimal hyperparameters located during our extensive sweep.

Table 10: Best hyperparameter selection over a sweep of various parameter grids. 'ilr' = inner optimizer learning rate, 'olr' = outer optimizer learning rate, 'ieps' = inner optimizer $\varepsilon$, 'oeps' = outer optimizer $\varepsilon$, 'o_u', 'o_d' = outer optimizer $u$, $d$, 'i_u', 'i_d' = inner optimizer $u$, $d$. Here, $\varepsilon$ is the adaptivity parameter in the denominator of adaptive optimizers to enhance stability of learning dynamics.

| Algorithm | Dataset | ilr | olr | ieps | oeps | o_u | o_d | i_u | i_d |
|---|---|---|---|---|---|---|---|---|---|
| **Avg-SGD** | STS-B | 0.019 | - | - | - | - | - | - | - |
| | RTE | 0.095 | - | - | - | - | - | - | - |
| | QNLI | 0.0059 | - | - | - | - | - | - | - |
| | QQP | 0.0074 | - | - | - | - | - | - | - |
| | CoLA | 0.019 | - | - | - | - | - | - | - |
| | SST-2 | 0.0074 | - | - | - | - | - | - | - |
| | MRPC | 0.038 | - | - | - | - | - | - | - |
| | MNLI | 0.0059 | - | - | - | - | - | - | - |
| **Avg-$L_2$Clip** | STS-B | 0.56 | - | - | - | - | - | 1.5 | 0.0 |
| | RTE | 1 | - | - | - | - | - | 0.14 | 0.0 |
| | QNLI | 0.33 | - | - | - | - | - | 0.14 | 0.0 |
| | QQP | 0.44 | - | - | - | - | - | 0.14 | 0.0 |
| | CoLA | 0.33 | - | - | - | - | - | 0.14 | 0.0 |
| | SST-2 | 0.11 | - | - | - | - | - | 0.27 | 0.0 |
| | MRPC | 0.22 | - | - | - | - | - | 0.41 | 0.0 |
| | MNLI | 0.11 | - | - | - | - | - | 0.41 | 0.0 |
| **Avg-$BiClip$** | STS-B | 0.44 | - | - | - | - | - | 0.0001 | 0.0001 |
| | RTE | 1 | - | - | - | - | - | 0.0001 | 6.7e-5 |
| | QNLI | 0.44 | - | - | - | - | - | 0.0001 | 6.7e-5 |
| | QQP | 0.56 | - | - | - | - | - | 0.0001 | 3.3e-5 |
| | CoLA | 0.89 | - | - | - | - | - | 0.0001 | 0.0001 |
| | SST-2 | 0.56 | - | - | - | - | - | 0.0001 | 6.7e-5 |
| | MRPC | 0.89 | - | - | - | - | - | 0.0001 | 6.7e-5 |
| | MNLI | 0.56 | - | - | - | - | - | 0.0001 | 3.3e-5 |
| **Avg-$BiClip$ ($L_2$)** | STS-B | 0.067 | - | - | - | - | - | 0.75 | 0.75 |
| | RTE | 1 | - | - | - | - | - | 0.0001 | 6.7e-5 |
| | QNLI | 0.067 | - | - | - | - | - | 0.75 | 0.75 |
| | QQP | 0.11 | - | - | - | - | - | 0.5 | 0.33 |
| | CoLA | 0.067 | - | - | - | - | - | 0.75 | 0.75 |
| | SST-2 | 0.1 | - | - | - | - | - | 0.75 | 0.38 |
| | MRPC | 0.11 | - | - | - | - | - | 1 | 1 |
| | MNLI | 0.033 | - | - | - | - | - | 1.5 | 1.5 |
| $Bi^2Clip$ | STS-B | 0.5 | 0.5 | - | - | 0.0001 | 0.0001 | 0.0001 | 1e-7 |
| | RTE | 1 | 1 | - | - | 0.0001 | 0.0001 | 0.001 | 5e-5 |
| | QNLI | 0.5 | 1 | - | - | 0.0001 | 0.0001 | 0.0001 | 5e-5 |
| | QQP | 0.5 | 1 | - | - | 1.5 | 1e-7 | 0.0001 | 5e-5 |
| | CoLA | 0.5 | 1 | - | - | 0.0001 | 0.0001 | 0.0001 | 0.0001 |
| | SST-2 | 0.5 | 1 | - | - | 0.75 | 1e-7 | 0.0001 | 1e-7 |
| | MRPC | 1 | 1 | - | - | 0.0001 | 0.0001 | 0.0001 | 1e-7 |
| | MNLI | 0.5 | 1 | - | - | 0.75 | 1e-7 | 0.0001 | 1e-7 |
| $Bi^2Clip$ ($L_2$) | STS-B | 0.56 | 3.2 | - | - | 0.1 | 0.05 | 0.1 | 0.05 |
| | RTE | 0.1 | 0.56 | - | - | 0.1 | 0.1 | 0.56 | 0.56 |
| | QNLI | 0.1 | 0.1 | - | - | 3.2 | 3.2 | 0.56 | 1e-7 |
| | QQP | 0.1 | 3.2 | - | - | 0.56 | 1e-7 | 0.56 | 0.56 |
| | CoLA | 0.1 | 3.2 | - | - | 0.1 | 0.05 | 0.56 | 1e-7 |
| | SST-2 | 0.56 | 0.1 | - | - | 3.2 | 3.2 | 0.1 | 1e-7 |
| | MRPC | 0.56 | 0.1 | - | - | 0.56 | 0.56 | 0.1 | 0.1 |
| | MNLI | 0.1 | 0.56 | - | - | 3.2 | 1.6 | 0.56 | 1e-7 |

Table 11: Best hyperparameter selection over a sweep of various parameter grids. 'ilr' = inner optimizer learning rate, 'olr' = outer optimizer learning rate, 'ieps' = inner optimizer $\varepsilon$, 'oeps' = outer optimizer $\varepsilon$, 'o_u', 'o_d' = outer optimizer $u$, $d$, 'i_u', 'i_d' = inner optimizer $u$, $d$. Here, $\varepsilon$ is the adaptivity or $\varepsilon$-smoothing parameter employed in the denominator of adaptive optimizers to enhance stability of learning dynamics.

| Algorithm | Dataset | ilr | olr | ieps | oeps | o_u | o_d | i_u | i_d |
|---|---|---|---|---|---|---|---|---|---|
| **Adam-SGD** | STS-B | 0.017 | 4.6e-5 | - | 1e-7 | - | - | - | - |
| | RTE | 0.033 | 4.6e-5 | - | 1e-7 | - | - | - | - |
| | QNLI | 0.017 | 2.2e-4 | - | 1e-7 | - | - | - | - |
| | QQP | 0.017 | 2.2e-4 | - | 1e-7 | - | - | - | - |
| | CoLA | 0.033 | 0.001 | - | 1e-5 | - | - | - | - |
| | SST-2 | 0.017 | 4.6e-5 | - | 1e-7 | - | - | - | - |
| | MRPC | 0.017 | 4.6e-5 | - | 1e-7 | - | - | - | - |
| | MNLI | 0.017 | 2.2e-4 | - | 1e-7 | - | - | - | - |
| **Adam-**$L_2Clip$ | STS-B | 0.067 | 0.033 | - | 0.001 | - | - | 0.75 | 0.0 |
| | RTE | 0.033 | 1e-5 | - | 1e-7 | - | - | 1.5 | 0.0 |
| | QNLI | 0.067 | 0.067 | - | 0.001 | - | - | 0.75 | 0.0 |
| | QQP | 0.067 | 0.033 | - | 0.001 | - | - | 1.5 | 0.0 |
| | CoLA | 0.1 | 0.033 | - | 0.001 | - | - | 0.75 | 0.0 |
| | SST-2 | 0.1 | 0.033 | - | 0.001 | - | - | 1.5 | 0.0 |
| | MRPC | 0.033 | 0.033 | - | 0.001 | - | - | 0.75 | 0.0 |
| | MNLI | 0.067 | 0.033 | - | 0.001 | - | - | 0.75 | 0.0 |
| **Adam-**$BiClip$ | STS-B | 0.0056 | 3.2e-4 | - | 1e-5 | - | - | 0.01 | 0.0067 |
| | RTE | 3.2e-4 | 1.8e-5 | - | 1e-7 | - | - | 0.01 | 0.0067 |
| | QNLI | 0.0056 | 3.2e-4 | - | 1e-7 | - | - | 0.01 | 0.0067 |
| | QQP | 0.0056 | 0.00032 | - | 1e-7 | - | - | 0.01 | 0.0033 |
| | CoLA | 0.0056 | 1.8e-5 | - | 1e-7 | - | - | 0.01 | 0.01 |
| | SST-2 | 0.0056 | 1.8e-5 | - | 1e-7 | - | - | 0.01 | 0.0067 |
| | MRPC | 0.0056 | 0.0056 | - | 0.001 | - | - | 0.056 | 0.019 |
| | MNLI | 0.0056 | 3.2e-4 | - | 1e-5 | - | - | 0.01 | 0.0033 |
| **Adam-**$BiClip$ ($L_2$) | STS-B | 0.033 | 0.033 | - | 0.001 | - | - | 1.5 | 0.75 |
| | RTE | 0.033 | 0.067 | - | 0.001 | - | - | 0.75 | 0.38 |
| | QNLI | 0.033 | 0.067 | - | 0.001 | - | - | 1.5 | 0.75 |
| | QQP | 0.067 | 0.033 | - | 0.0001 | - | - | 0.75 | 0.38 |
| | CoLA | 0.033 | 0.033 | - | 0.001 | - | - | 1.5 | 0.75 |
| | SST-2 | 0.067 | 0.033 | - | 0.001 | - | - | 1.5 | 1e-7 |
| | MRPC | 0.033 | 0.033 | - | 0.001 | - | - | 1.5 | 1e-7 |
| | MNLI | 0.067 | 0.033 | - | 0.001 | - | - | 1.5 | 0.75 |
| $Adam^2$ | STS-B | 1.8e-5 | 1.8e-5 | 1e-5 | 1e-7 | - | - | - | - |
| | RTE | 1.8e-5 | 1.8e-5 | 1e-5 | 1e-7 | - | - | - | - |
| | QNLI | 1.8e-5 | 3.2e-4 | 1e-5 | 1e-5 | - | - | - | - |
| | QQP | 1.8e-5 | 3.2e-4 | 1e-5 | 1e-7 | - | - | - | - |
| | CoLA | 1.8e-5 | 0.0056 | 1e-5 | 0.001 | - | - | - | - |
| | SST-2 | 1.8e-5 | 1.8e-5 | 0.001 | 1e-7 | - | - | - | - |
| | MRPC | 1.8e-5 | 1.8e-5 | 1e-5 | 1e-7 | - | - | - | - |
| | MNLI | 1.8e-5 | 3.2e-4 | 1e-5 | 1e-7 | - | - | - | - |

Table 12: The notational setup is analogous to Table 11. For DiLoCo*, we provide the Nesterov learning rate and weight decay parameter in the i_u, i_d entries, respectively.

| Algorithm | Dataset | ilr | olr | ieps | oeps | o_u | o_d | i_u | i_d |
|---|---|---|---|---|---|---|---|---|---|
| **Adagrad-SGD** | STS-B | 0.017 | 0.0046 | - | 0.001 | - | - | - | - |
| | RTE | 0.033 | 0.001 | - | 1e-5 | - | - | - | - |
| | QNLI | 0.017 | 0.001 | - | 1e-5 | - | - | - | - |
| | QQP | 0.017 | 0.0001 | - | 1e-5 | - | - | - | - |
| | CoLA | 0.017 | 2.2e-4 | - | 1e-7 | - | - | - | - |
| | SST-2 | 0.017 | 2.2e-4 | - | 1e-5 | - | - | - | - |
| | MRPC | 0.017 | 2.2e-4 | - | 1e-7 | - | - | - | - |
| | MNLI | 0.017 | 0.0001 | - | 1e-7 | - | - | - | - |
| **RMSProp-SGD** | STS-B | 0.017 | 1e-5 | - | 1e-7 | - | - | - | - |
| | RTE | 0.017 | 1e-5 | - | 1e-7 | - | - | - | - |
| | QNLI | 0.033 | 0.001 | - | 1e-5 | - | - | - | - |
| | QQP | 0.017 | 1e-5 | - | 1e-7 | - | - | - | - |
| | CoLA | 0.017 | 1e-5 | - | 1e-7 | - | - | - | - |
| | SST-2 | 0.017 | 1e-5 | - | 1e-7 | - | - | - | - |
| | MRPC | 0.033 | 1e-5 | - | 1e-7 | - | - | - | - |
| | MNLI | 0.017 | 1e-5 | - | 1e-7 | - | - | - | - |
| **Adagrad-**$BiClip$ | STS-B | 1e-5 | 2.2e-4 | - | 1e-7 | - | - | 1.5 | 1.5 |
| | RTE | 0.033 | 2.2e-4 | - | 1e-7 | - | - | 1.5 | 1e-7 |
| | QNLI | 1e-5 | 0.0046 | - | 0.001 | - | - | 1.5 | 1.5 |
| | QQP | 1e-5 | 0.0046 | - | 0.0001 | - | - | 1.5 | 1.5 |
| | CoLA | 0.1 | 2.2e-4 | - | 1e-7 | - | - | 0.0001 | 5e-5 |
| | SST-2 | 1e-5 | 0.0046 | - | 0.001 | - | - | 1.5 | 1.5 |
| | MRPC | 1e-5 | 2.2e-4 | - | 1e-7 | - | - | 1.5 | 0.75 |
| | MNLI | 1e-5 | 0.0046 | - | 0.001 | - | - | 1.5 | 1.5 |
| **RMSProp-**$BiClip$ | STS-B | 1e-5 | 1e-5 | - | 1e-7 | - | - | 1.5 | 1.5 |
| | RTE | 0.067 | 1e-5 | - | 1e-7 | - | - | 0.0001 | 5e-5 |
| | QNLI | 0.1 | 1e-5 | - | 1e-7 | - | - | 0.0001 | 0.0001 |
| | QQP | 0.1 | 0.0046 | - | 1e-7 | - | - | 0.0001 | 5e-5 |
| | CoLA | 0.1 | 0.0046 | - | 0.001 | - | - | 0.0001 | 1e-7 |
| | SST-2 | 0.1 | 1e-5 | - | 1e-7 | - | - | 0.0001 | 0.0001 |
| | MRPC | 1e-5 | 0.0046 | - | 0.001 | - | - | 0.75 | 0.75 |
| | MNLI | 0.1 | 0.0046 | - | 0.001 | - | - | 0.0001 | 0.0001 |
| **DiLoCo*** | STS-B | 1.8e-5 | 0.7 | 1e-5 | - | - | - | 0.9 | 0.1 |
| | RTE | 1.8e-5 | 1 | 1e-5 | - | - | - | 0.95 | 0.0001 |
| | QNLI | 1.8e-5 | 1 | 1e-5 | - | - | - | 0.9 | 0.0001 |
| | QQP | 1.8e-5 | 1 | 1e-5 | - | - | - | 0.95 | 0.0001 |
| | CoLA | 1.8e-5 | 1 | 1e-5 | - | - | - | 0.95 | 0.1 |
| | SST-2 | 1.8e-5 | 0.1 | 1e-5 | - | - | - | 0.9 | 0.0001 |
| | MRPC | 1.8e-5 | 0.7 | 1e-5 | - | - | - | 0.9 | 0.1 |
| | MNLI | 1.8e-5 | 1 | 1e-5 | - | - | - | 0.9 | 0.1 |

Table 13: Best hyperparameter selection over a sweep of various parameter grids for GLUE tasks. The notation is analogous to Table 11.

| Algorithm | Dataset | ilr | olr | ieps | oeps | o_u | o_d | i_u | i_d |
|---|---|---|---|---|---|---|---|---|---|
| **Avg-Adagrad** | STS-B | 3e-5 | - | 1e-8 | - | - | - | - | - |
| | RTE | 1.5e-4 | - | 1e-6 | - | - | - | - | - |
| | QNLI | 3.3e-4 | - | 0.001 | - | - | - | - | - |
| | QQP | 3.3e-4 | - | 0.001 | - | - | - | - | - |
| | CoLA | 6.7e-5 | - | 1e-6 | - | - | - | - | - |
| | SST-2 | 3.3e-4 | - | 0.001 | - | - | - | - | - |
| | MRPC | 1.5e-4 | - | 1e-6 | - | - | - | - | - |
| | MNLI | 3.3e-4 | - | 0.001 | - | - | - | - | - |
| **Avg-Adam** | STS-B | 1.4e-5 | - | 1e-6 | - | - | - | - | - |
| | RTE | 3e-5 | - | 1e-8 | - | - | - | - | - |
| | QNLI | 6.2e-6 | - | 1e-8 | - | - | - | - | - |
| | QQP | 1.4e-5 | - | 1e-8 | - | - | - | - | - |
| | CoLA | 6.2e-6 | - | 1e-8 | - | - | - | - | - |
| | SST-2 | 6.2e-6 | - | 1e-8 | - | - | - | - | - |
| | MRPC | 3e-5 | - | 1e-8 | - | - | - | - | - |
| | MNLI | 3e-5 | - | 0.0001 | - | - | - | - | - |

Table 14: Best hyperparameter selection over a sweep of various parameter grids for WMT. The conventions are identical with Tables 11-13.

| Algorithm | Dataset | ilr | olr | ieps | oeps | o_u | o_d | i_u | i_d |
|---|---|---|---|---|---|---|---|---|---|
| **Avg-SGD** | TED-T (en-de) | 0.03 | - | - | - | - | - | - | - |
| | TED-T (en-fr) | 0.015 | - | - | - | - | - | - | - |
| | NewsComm (en-fr) | 0.015 | - | - | - | - | - | - | - |
| **Avg-**$L_2Clip$ | TED-T (en-de) | 0.89 | - | - | - | - | - | 1.4 | 0.0 |
| | TED-T (en-fr) | 0.89 | - | - | - | - | - | 0.55 | 0.0 |
| | NewsComm (en-fr) | 0.78 | - | - | - | - | - | 0.41 | 0.0 |
| $Bi^2Clip$ | TED-T (en-de) | 1 | 1 | - | - | 0.0001 | 0.0001 | 0.75 | 1e-7 |
| | TED-T (en-fr) | 1 | 1 | - | - | 0.0001 | 0.0001 | 0.75 | 1e-7 |
| | NewsComm (en-fr) | 0.5 | 1 | - | - | 1.5 | 1e-7 | 0.0001 | 5e-5 |
| $Adam^2$ | TED-T (en-de) | 3.2e-4 | 0.0056 | 1e-7 | 0.001 | - | - | - | - |
| | TED-T (en-fr) | 1.8e-5 | 1.8e-5 | 1e-5 | 1e-7 | - | - | - | - |
| | NewsComm (en-fr) | 3.2e-4 | 0.0056 | 1e-5 | 0.001 | - | - | - | - |

Table 15: Best hyperparameter selection over a sweep of various parameter grids. The conventions are identical with Tables 11-14.

| Algorithm | Dataset | ilr | olr | ieps | oeps | o_u | o_d | i_u | i_d |
|---|---|---|---|---|---|---|---|---|---|
| **Avg-SGD** | Shakespeare | 0.012 | - | - | - | - | - | - | - |
| | Philosopher | 0.15 | - | - | - | - | - | - | - |
| **Avg-**$L_2Clip$ | Shakespeare | 0.56 | - | - | - | - | - | 0.55 | 0 |
| | Philosopher | 1 | - | - | - | - | - | 0.41 | 0 |
| **Avg-**$BiClip$ | Shakespeare | 1 | - | - | - | - | - | 0.0001 | 3.3e-5 |
| | Philosopher | 1 | - | - | - | - | - | 0.0001 | 3.3e-5 |
| **RMSProp-**$BiClip$ | Shakespeare | 0.067 | 2.2e-4 | - | 1e-5 | - | - | 0.75 | 1e-7 |
| | Philosopher | 0.067 | 0.0046 | - | 0.001 | - | - | 0.75 | 1e-7 |
| $Bi^2Clip$ | Shakespeare | 1 | 1 | - | - | 1.5 | 1e-7 | 0.0001 | 0.0001 |
| | Philosopher | 1 | 1 | - | - | 1.5 | 1e-7 | 0.0001 | 5e-5 |
| $Adam^2$ | Shakespeare | 1.8e-5 | 0.0056 | 1e-7 | 0.001 | - | - | - | - |
| | Philosopher | 1.8e-5 | 0.0056 | 1e-5 | 1e-5 | - | - | - | - |

