# OpenReview forum: "Efficient Distributed Optimization under Heavy-Tailed Noise"
_ICML.cc/2025/Conference — ICML 2025 poster_

### Official Review · Reviewer_y2B4 · 2025-03-14

**Overall Recommendation:** 3

**Summary:**

The paper proposes a new optimization algorithm BiClip, for heavy-tailed stochastic optimization. Instead of bounding the upper bound of the gradient, the authors also propose to bound the gradient from below. Combining the clipping method with distributed SGD, the authors propose $Bi^2Clip$. The performance of the proposed method is good in the experiments.

**Claims And Evidence:**

Yes

**Essential References Not Discussed:**

No

**Experimental Designs Or Analyses:**

Yes.

1. The experiments on the Transfoemers are for encoder-based one. It would be better to have the experiments on decoder-based transformers.

**Methods And Evaluation Criteria:**

Yes

**Other Comments Or Suggestions:**

The notation $d_t$ for lower bound and notatin $d$ for the dimension of $x$ are confusing.

**Other Strengths And Weaknesses:**

Strength:

1. The proposed method can be easily combined with other optimization algorithms.

2. the authors give the theorectical analysis to show the convergence of proposed method.

Weakness:

1. The algorithm introduce two more schedulers on the upper bound of gradient and lower bound for the gradient.

**Questions For Authors:**

1. Can authors clearly state the assumptions in both theorem and corollaries in Section 5?

2.Is the algorithm hard to tune? Can the author provide sentivity analysis for the $d_t$ and $u_t$.

**Relation To Broader Scientific Literature:**

Powerful distributed algorithms can highly improve the model's ability for all model training in machine learning problems.

**Theoretical Claims:**

Yes, but only for the sketch of the proof.


1. The assumptions used in Theorem 2 is unclear.

2. The proposed algorithm does not seem to have theoretical benifit.

---

> ### Author Rebuttal · Authors · 2025-04-01
>
> We thank reviewer y2B4 for their review. As the reviewer noted, TailOPT can be readily combined with a wide range of optimization strategies, which greatly enhances its practical applicability.  We propose several novel instantiations of TailOPT (such as $Bi^2Clip$) that achieves the strongest empirical performance without the extra memory/compute/communication overhead compared with baselines. Additionally, the TailOPT framework provides theoretical convergence guarantees under heavy-tailed noise with unbounded gradient variance and local updates, and these results have not appeared in prior works.
>
> **[No Decoder-Based Transformers?]** We clarify that our evaluations already include encoder-only (RoBERTa), encoder-decoder (T5), and decoder-only (GPT-2, DistillGPT) transformers. Results on decoder-based architectures (T5, GPT) are included in both Section 6 and Appendix H. For clarity, we summarize key empirical results below:
>
> - Section 6.1 (Synthetic Experiments, Full results in Appendix G.1): We carefully inject heavy-tailed noise into gradients via corrupted labels, simulating both a controlled heavy-tailed and non-heavy-tailed regime. Results show that TailOPT significantly outperforms Adam, with the performance gap widening under heavy-tailed conditions. Notably, we show that heavy-tailed noise destabilizes conventional training methods, a setting in which TailOPT instantions consistently demonstrate optimal performance.
>
> - Section 6.2 (GLUE Benchmark): Table 10 (expanded Table 1) extensively evaluates ~150 optimizer-dataset pairs. $Bi^2Clip$, which applies adaptivity to both inner/outer optimizers, consistently achieves the best performance. We find that the action of simply switching the inner optimizer from SGD to $BiClip$ yields approximately *10% gain on GLUE* ** *average* ** *with identical memory and communication cost*, highlighting the importance of TailOPT and heavy-tailed convergence in training modern transformer models.
>
> - Section 6.3 and Appendix H.2 (Machine Translation/Generation): TailOPT variants outperform $Adam^2$, DiLoCo, and FedAdam across nearly all machine translation tasks, while being more resource-efficient. Table 11 (expanded Table 2) confirms optimal performance of $Bi^2Clip$ on generative tasks via T5, and Appendix H.3 explores robustness under partial participation using GPT-2 and DistillGPT. Across all settings, algorithmic instantiations from TailOPT consistently achieve state-of-the-art performance.
>
> **[Other Questions]** We use $d$ to denote the coordinate dimension, following optimization literature conventions. The variables $u_t$ and $d_t$ represent “up” and “down” clipping thresholds, respectively. We are happy to revise notation to avoid confusion.
>
> Regarding hyperparameter tuning, TailOPT, particularly $Bi^2Clip$, is easy to tune. We use fixed clipping values across all iterations (lines 290-306) and sweep over only *three* grid values each for $u_t$ and $d_t$ for $Bi^2Clip$ (Table 3, Appendix G.5). For example, for MNLI, we sweep $d_t$ over {1e-7, 5e-5, 1e-4}, and the best performing validation accuracies are 0.850, 0.849, and 0.850, with barely any difference. Despite this minimal tuning effort, $Bi^2Clip$ exceeds state-of-the-art performance. In contrast, other baselines (such as Adam) requires careful tuning of more hyperparameters, that is, the learning rate, $\beta_1$, $\beta_2$, and $\varepsilon$. Moreover, $BiClip$ requires only 33% of the memory used by Adam to store gradient statistics and even avoids the need to store or compute auxiliary gradient statistics at all, leading to significant improvements in memory, compute efficiency, and communication cost.
>
> **[Theoretical Results]** Assumptions are listed in Sections 3, 5, lines 145-164, 226-234 (L-smoothness, bounded $\alpha$-moment) for the theorems and corollaries mentioned. We also believe the reviewer may have unintentionally interpreted our setting through the lens of the bounded stochastic gradient literature, where the state-of-the-art convergence rate for non-convex objectives is $\mathcal{O}(T^{-1/2})$. However, these results **do not** generalize to the heavy-tailed regime with local updates (under the **unbounded** stochastic gradient variance assumption), which is the focus of our work. TailOPT is, to our knowledge, **the first general framework** to provably achieve this optimal rate under heavy-tailed noise, a setting that is highly relevant to training modern models such as large language models and large-scale transformers. Due to the unbounded variance in this regime, both the theoretical analysis and algorithmic design are significantly more challenging, which TailOPT addresses theoretically with convergence guarantees which are rigorously validated by empirical results. Our extensive evaluations further support these theoretical advantages, demonstrating consistent improvements over state-of-the-art baselines.
>
> We are happy to answer any further questions.

---

### Official Review · Reviewer_QUNM · 2025-03-14

**Overall Recommendation:** 4

**Summary:**

The paper studies distributed optimization under commonly used schemes of local steps followed by global synchronization, and specifically addresses the issue of heavy-tailed gradient variance in this setting, which is a very relevant problem.
As a solutions clipping techniques are proposed to stabilize both inner and outer optimizers, achieving communication efficient training without the need to exchange gradient statistics.

**Claims And Evidence:**

The paper provides convincing theory and empirical results on relevant language modelling tasks (including finetuning Roberta and T5 respectively). Empirical results show Adam-like performance without the need to exchange gradient statistics, which is a valuable contribution.

**Essential References Not Discussed:**

(other reviewers might comment)

**Experimental Designs Or Analyses:**

The experimental results to me look convincing, and hyperparameter selection schedules in appendix are reasonable

**Methods And Evaluation Criteria:**

yes, on both theory and experimental side

**Other Comments Or Suggestions:**

-

**Other Strengths And Weaknesses:**

-

**Questions For Authors:**

see theory section above

**Relation To Broader Scientific Literature:**

The work cites relevant sources appropriately as far as I can see

**Theoretical Claims:**

The clarity of the theoretical sections should be improved a bit:
- clarifying terminology or "maximal rate at least" (it was more clear for Thm 6)
- more closely putting it in context with the existing results for same algorithms in the case of more basic bounded noise, and for simple clipping instead of bi-clip
- clarify how the bi-clip result recovers (or not) the tail-clip result (as it includes this setting as a special case?)

I was not able to check the proofs, would have appreciated a brief comment on proof intuition/overview and related literature techniques

---

> ### Author Rebuttal · Authors · 2025-04-01
>
> We thank reviewer QUNM for the feedback and for finding our theoretical and empirical results convincing. We fully agree that addressing heavy-tailed gradient variance is a critical challenge, particularly in the context of training large-scale models such as LLMs. As the reviewer notes, the empirical performance of the novel $BiClip$ method (an instantiation of our proposed TailOPT framework) matches or exceeds that of Adam, while avoiding the substantial memory, communication, and compute costs associated with maintaining extra gradient statistics. Moreover, TailOPT is equipped with theoretical guarantees in the heavy-tailed regime with unbounded variance and local updates. We answer the reviewer’s clarification questions below.
>
> **[Clarification of Theory Sections]**
>
> - In response to the reviewer’s suggestions, we will clarify the “maximal rate at least” terminology to align with the formal statement in Theorem 6. Thank you for this feedback.
> - As discussed after Corollary 1, we compare our results with existing convergence rates of other algorithms. We will further expand the discussion comparing TailOPT to existing baselines, such as Avg+SGD, Adam+SGD, and Adagrad+SGD, which are currently known to converge only under bounded gradient noise. Furthermore, we note that the convergence of the advanced $Adam^2$ and DiLoCo algorithms under heavy-tailed noise remain unknown in prior works. Due to space constraints, we moved discussions around convergence of L2 Clip to Appendix C and Appendix D, but we will discuss connections in the main text in the next version.
> - We will also move the discussion from the Appendix (lines 1516-1522) regarding the generalization of BiClip to standard L2 Clipping into the main text, and expand it for clarity. We note that for proving the framework-wide convergence for TailOPT, we prove the bounds separately for various instantiations, including $Bi^2Clip$, RMSProp-$BiClip$, etc. That is, the $Bi^2Clip$ theoretical result does recover other TailOPT instantiations such as Adagrad-$BiClip$, as they are based on different inner and outer optimizers.

---

### Official Review · Reviewer_6e8Q · 2025-03-16

**Overall Recommendation:** 2

**Summary:**

The paper introduces TailOPT, a distributed optimization framework designed to handle heavy-tailed gradient noise in large-scale machine learning models. The authors propose a novel clipping mechanism, \(BiClip\), which performs coordinate-wise clipping to mitigate the effects of heavy-tailed noise without the memory and communication overhead of adaptive optimizers like Adam. The paper provides theoretical convergence guarantees for TailOPT under heavy-tailed noise and demonstrates its empirical effectiveness on several language tasks and models, outperforming state-of-the-art methods.

**Claims And Evidence:**

seems OK

**Essential References Not Discussed:**

Some recent works related to AdaGrad could be discussed in the paper:

Kavis, Ali, Kfir Yehuda Levy, and Volkan Cevher. "High Probability Bounds for a Class of Nonconvex Algorithms with AdaGrad Stepsize." International Conference on Learning Representations.
Hong Y, Lin J. Revisiting convergence of AdaGrad with relaxed assumptions[C]//Proceedings of the Fortieth Conference on Uncertainty in Artificial Intelligence. 2024: 1727-1750.
Faw M, Rout L, Caramanis C, et al. Beyond uniform smoothness: A stopped analysis of adaptive sgd[C]//The Thirty Sixth Annual Conference on Learning Theory. PMLR, 2023: 89-160.

**Experimental Designs Or Analyses:**

It seems OK.

**Methods And Evaluation Criteria:**

seems OK

**Other Comments Or Suggestions:**

Line138Column1, the form $F_{i}(x,\xi) = F_i(x) + \langle \xi,x \rangle$ is a bit strange to me.

Theorem 1, use "Assumptions 1-2", instead of "assumptions 1-2".

**Other Strengths And Weaknesses:**

**Strengths:**
1.  The proposed \(BiClip\) mechanism appears to be a new approach to handling heavy-tailed noise in distributed optimization, offering a memory-efficient alternative to adaptive optimizers.
2. The paper provides rigorous convergence guarantees for TailOPT under heavy-tailed noise.
3.  The authors conduct extensive experiments on synthetic and real-world datasets, demonstrating the practical effectiveness of TailOPT, particularly in large-scale settings.

**Weaknesses:**
1.  The paper is dense and difficult to follow, particularly in the theoretical sections. The presentation of the convergence proofs is overly complex, and the key insights are often buried under heavy notation and technical details. A more intuitive explanation of the main ideas would greatly improve readability.
2. While the empirical results are promising, the paper lacks a thorough ablation study to isolate the impact of different components of TailOPT. Additionally, the comparison with state-of-the-art methods could be more comprehensive, particularly in terms of computational efficiency and scalability.
3.  The paper does not sufficiently address the practical challenges of implementing TailOPT in real-world distributed systems. For example, the impact of network latency, node failures, and heterogeneous hardware on the performance of TailOPT is not discussed.
4.  "All the results implicitly assume that the gradient (or the function value gaps) is bounded, which is a somewhat strong assumption."

**Questions For Authors:**

Does Theorem 1 hold for any $\tilde{\beta_2} \in [0,1)$? Traditional convergence results for RMSProp typically requires that $\beta$ is close to 1.

Is it possible to remove the bounded gradient assumption in all the results?

What is the dependency on $\tau$ in the upper bound of Theorem 6?

**Relation To Broader Scientific Literature:**

The paper introduces TailOPT, a distributed optimization framework designed to handle heavy-tailed gradient noise (rather than the light-tailed gradient noise) in large-scale machine learning models.

**Theoretical Claims:**

I did not check the proof.

---

> ### Author Rebuttal · Authors · 2025-04-01
>
> We thank reviewer 6e8Q for their review. We appreciate the reviewer's acknowledgements about the strengths of our paper, particularly the novelty and efficiency of our proposed algorithm.
>
> **[Proof Challenges]** The challenges in the analysis lie in heavy-tailed noise with unbounded gradient variance and bias introduced by clipping. The core idea is to decompose model updates into descent directions and clipping bias. We carefully design clipping thresholds so that the bias introduced by thresholding diminishes asymptotically. We will incorporate the reviewer's feedback and further clarify key insights.
>
> **[Ablation Studies]** In our submission, we already conducted ablation studies on all components of TailOPT and explored most combinations (e.g., Avg-L2Clip vs Avg-BiClip to explore clipping strategies, Adam-SGD vs Avg-SGD to explore effects of outer optimizers, etc) in Section 6. For instance, Table 1 (or Table 10, Appendix H.1) shows the performance of many methods differing by inner/outer optimizers. We see that (a) our $BiClip$ can mitigate the negative effects of heavy-tailed noise while achieving Adam-like performance, and (b) $BiClip$ applied to both inner/outer optimizers ($Bi^2Clip$) outperforms all baselines.
>
> Additionally, we include a summary table comparing communication/memory requirements for gradient statistics of different algorithms. The communication cost includes model parameters (dimension d). In particular, $Bi^2Clip$ achieves best performance, without incurring extra memory/communication cost compared with baselines.
>
> | Optimizer                  | Inner Memory | Outer Memory | Communication | Avg. GLUE |
> |---------------------------|-----------|-----------|-----------|-----------------|
> | Avg-SGD                   | d         | d         | d         | 61.17           |
> | RMSProp-SGD               | d         | 2d        | d         | 74.73           |
> | RMSProp-BiClip (TailOPT)  | d         | 2d        | d         | 82.99           |
> | $Adam^2$                     | 3d        | 3d        | 2d        | 82.93           |
> | DiLoCo                    | 3d        | 2d         | d         | 83.08           |
> | **$Bi^2Clip$ (TailOPT)**     | d         | d         | d         | **84.52**       |
>
> **[Practical Implementation]**  Our goal is not to develop distributed optimization methods that handle varying network latency or heterogeneous hardware. Instead, we focus on developing, analyzing, and evaluating a communication- and memory-efficient framework (TailOPT) that addresses heavy-tailed gradient noise. That said, we do evaluate TailOPT with node failures theoretically (Appendix D) and empirically (Appendix H.3). We see that under partial participation (e.g., Table 12), algorithms using BiClip achieve the top accuracies. Additionally, the lightweight and modular nature of TailOPT (e.g., table above) ensures ease of TailOPT deployment at scale in practical distributed systems. TailOPT is a stateless algorithm---no algorithm states need be maintained across communication rounds, which also makes it suitable for large-scale potentially resource-constrained settings.
>
> **[Bounded Deterministic Gradients]**
> We assume only the deterministic gradient $\nabla F(x)$ is bounded, while the stochastic gradient $\nabla F(x) + \xi$ is unbounded due to heavy-tailed $\xi$. The former is a common assumption in distributed optimization (e.g., [1-2]), where even the stochastic gradient variance is assumed bounded, a much stronger assumption. Additionally, our assumptions are much weaker than prior works which assume bounded variance on the heavy-tailed stochastic gradient (e.g., see lines 110-124) whereas we allow it to be unbounded.
>
> **[Other Questions & Related Work]**
> - The additive form $F_i(x,\xi) = F_i(x) + \langle \xi, x \rangle$ follows from integrating $\nabla F_i(x,\xi) = \nabla F(x) + \xi$ where $\xi$ models heavy-tailed noise (lines 134-140).
> - Theorem 1 holds for all $\beta_2 \in [0,1)$, thus extending RMSProp-style results as the reviewer noted.
> - We assume a fixed small $\varepsilon$ in the analysis. For readability, we presented only the schedule-dependent terms in the theorem statement. In Theorem 6, setting $\tau = 0$ invalidates convergence by exploding bounds (lines 1681-1694), which is consistent with empirical results highlighting the importance of the adaptivity parameter.
> - We will include discussions on the suggested references, which covers Adagrad-type algorithms. [3] assumes bounded variance and almost surely bounded gradients, a much stronger assumption, and [4] studies affine variance which is different from heavy-tailed.
>
> [1] Zaheer et al., Adaptive methods for nonconvex optimization, 2018
>
> [2] Reddi et al., Adaptive Federated Optimization, 2021
>
> [3] Kavis et al., High Probability Bounds for a Class of Non-Convex Algorithms with Adagrad Stepsize, 2022
>
> [4] Faw et al., Beyond Uniform Smoothness: A Stopped Analysis of Adaptive SGD Affine variance, 2023

---

### Decision · Program_Chairs · 2025-05-01

**Decision:**

Accept (poster)

**Comment:**

This paper introduces TailOPT, a distributed optimization framework for handling heavy-tailed gradient noise, along with a novel clipping mechanism (BiClip) that operates without incurring the memory and communication overhead of adaptive optimizers. The authors provide theoretical convergence guarantees under unbounded noise which is new to the best of my knowledge is new and validate their method on a range of language tasks, showing Adam-like performance with lower resource cost. There are no new hteoretical tools introduced, but the analysis and the presented results are interesting.

The technical contribution is clear and well-motivated, tackling a real problem in large-scale optimization where gradient variance could be heavy-tailed. The analysis is rigorous, though dense at times, and the empirical results are strong, well-structured, and comprehensive — including ablations, scalability analysis, and comparisons against both baselines and strong adaptive methods.
One reviewer questioned clarity and requested ablations, but the rebuttal shows these were already included in the submission.  Other reviewers found the work sound and practically relevant.

I recommend acceptance but encourage the authors to address the minor clarity issues that were raised by the reviewers.